# Canonical neurodevelopmental trajectories of structural and functional manifolds

**Alicja Monaghan**[1,2]*, **Richard AI Bethlehem**[3], **Danyal Akarca**[1,4], **Daniel S Margulies**[5,6], **the CALM Team**[1], **Duncan E Astle**[1,7]

[1]MRC Cognition and Brain Sciences Unit, University of Cambridge, Cambridge, United Kingdom; [2]The Alan Turing Institute, London, United Kingdom; [3]Department of Psychology, University of Cambridge, Cambridge, United Kingdom; [4]Department of Electrical and Electronic Engineering, Imperial College London, London, United Kingdom; [5]Oxford Centre for Integrative Neuroimaging (OxCIN), FMRIB, Nuffield Department of Clinical Neurosciences, University of Oxford, Oxford, United Kingdom; [6]Centre National de la Recherche Scientifique (CNRS), UAR 3129, Paris, France; [7]Department of Psychiatry, University of Cambridge, Cambridge, United Kingdom

**\*For correspondence:**
Alicja.Monaghan@mrc-cbu.cam.
ac.uk

## eLife Assessment

This **important** study provides insights into the neurodevelopmental trajectories of structural and functional connectivity gradients in the human brain and their potential associations with behaviour and psychopathology. The evidence supporting the findings is **solid**. This study will be of interest to neuroscientists interested in understanding functional connectivity across development.

**Abstract** Organisational gradients refer to a continuous low-dimensional embedding of brain regions and can quantify core organisational principles of complex systems like the human brain. Mapping how these organisational principles are altered or refined across development and pheno-types is essential to understanding the relationship between brain and behaviour. Taking a developmental approach and leveraging longitudinal and cross-sectional data from two multi-modal neuroimaging datasets, spanning the full neurotypical-neurodivergent continuum, we charted the organisational variability of structural (610 participants, N=390 with one observation, N=163 with two observations and N=57 with three) and functional (512 participants, N=340 with one observation, N=128 with two observations and N=44 with three). Across datasets, despite differing phenotypes, we observe highly similar structural and functional gradients. These gradients, or organisational principles, are highly stable across development, with the exact same ordering across early childhood into mid-adolescence. However, there is substantial developmental change in the strength of embedding within those gradients: by modelling developmental trajectories as non-linear splines, we show that structural and functional gradients are refined across development. Specifically, structural gradients gradually contract in low-dimensional space as networks become more integrated, whilst the functional manifold expands, indexing functional specialisation. The coupling of these structural and functional gradients follows a unimodal-association axis and varies across individuals, with developmental effects concentrated in the more plastic higher-order networks. Importantly, these developmental effects on coupling, in these higher-order networks, are attenuated in the neurodivergent sample. Finally, we mapped structure-function coupling onto dimensions of psycho-pathology and cognition and demonstrate that dimensions of cognition, such as working memory,

are robust predictors of coupling. In summary, across clinical and community samples, we demonstrate consistent principles of structural and functional brain organisation, with progressive structural integration and functional segregation. These gradients are established early in life, refined through development, and their coupling is predicted by working memory.

## Introduction

What are the core principles of human brain organisation? As we develop, when are those principles established? And how do those principles vary from person to person, according to differences in cognition and behaviour? These are central questions in developmental and systems neuroscience. An emerging approach is to conceptualise brain organisation through *gradients*, revealing latent dimensions (see *Huntenburg et al., 2018*). This provides an empirical extension to seminal work that demonstrated a sensory-fugal axis of structural and functional brain organisation (*Mesulam, 1998*). However, most studies have focussed on gradients of organisation *in adulthood* and at *a group-level*. Our aim was to better understand how those organisational principles emerge throughout childhood and adolescence, and to map variability in that organisation to key individual differences in cognition and psychopathology.

With the advent and rapid proliferation of neuroimaging technologies, it is now possible to image the developing brain at scale. White matter axonal inter-regional projections form a structural connectome, which reconfigures during childhood and adolescence towards a small-world topology (*Khundrakpam et al., 2013*). Highly intercorrelated structural cores provide a structural backbone (*Hagmann et al., 2008*) for functional connectivity (FC). Like SC, FC also reconfigures during childhood and adolescence towards a more distributed topology (*Fair et al., 2009*). In parallel, anatomical constraints on functional networks weaken across development, with strengthening of long-range functional connections between anatomically distant regions, and maintenance of short-range connectivity (*Fair et al., 2009*).

Gradients of organisation may be derived through non-linear dimensionality reduction techniques, such as diffusion map embedding (DME, *Coifman and Lafon, 2006*), providing a summary metric of organisational variability. The result is a set of eigenvectors, within a low-dimensional coordinate system, representing organisational axes upon which inter-regional structural and functional connections sit. Organisational variability may be summarised using different eigen-decomposition methods alongside DME, such as principal component analysis and multi-dimensional scaling. Several studies have now used DME to derive organisational gradients of structural and functional connectivity (reviewed in *Royer et al., 2024*), due to its ability to capture non-linear manifold structures and preserving local geometry. In typical development, structural gradients derived from microstructural profile covariance (*Paquola et al., 2019*; *Valk et al., 2022*), histology (*Paquola et al., 2019*), and probabilistic tractography (*Park et al., 2021*) extend from primary sensorimotor regions towards limbic and paralimbic regions. Functional gradients in childhood are centred in unimodal cortices, but are then anchored in transmodal cortices at the onset of puberty (*Dong et al., 2021*), along which the default-mode network is organised (*Margulies et al., 2016*). Functional constraints on structural manifolds follow an anterior-posterior axis, strongest in highly-differentiated unimodal regions and weakest in undifferentiated transmodal regions (*Paquola et al., 2019*; *Valk et al., 2022*). Structure-function correspondence develops hierarchically across this axis, such that regions with the strongest structural differentiation have the weakest correspondence to FC across childhood and adolescence (*Park et al., 2022*). Using the gradient-based approach as a compression tool, thus forgoing the need to specify singular graph theory metrics a priori, we operationalised individual variability in low-dimensional manifolds as eccentricity (*Gale et al., 2022*; *Park et al., 2021*). Crucially, such gradients appear to be useful predictors of phenotypic variation, exceeding edge-level connectomics. For example, in the case of functional connectivity gradients, their predictive ability for externalising symptoms and general cognition in neurotypical adults surpassed that of edge-level connectome-based predictive modelling (*Hong et al., 2019*), suggesting that capturing low-dimensional manifolds may be particularly powerful biomarkers of psychopathology and cognition.

The increasing availability of large-scale multi-modal neuroimaging databases, in both neurotypical and neurodivergent samples, creates the opportunity to comprehensively quantify canonical trajectories and variability in cortex-wide coordinate systems of structural and functional gradients

(see *Huntenburg et al., 2018*), with sufficient statistical power to make conclusions across development and phenotypes. Applying diffusion-map embedding as an unsupervised machine-learning technique onto matrices of communicability (from streamline SIFT2-weighted fibre bundle capacity) and functional connectivity, we derived gradients of structural and functional brain organisation in children and adolescents, spanning the full neurotypical-neurodivergent range. This allows us to infer the low-dimensional spatial topography of structure-based information transfer and emerging functional connectivity, respectively. Crucially, we employ a mixed cross-sectional and longitudinal design, leveraging structural (610 participants, N=390 with one observation, N=163 with two observations, and N=57 with three) and functional (512 participants, N=340 with one observation, N=128 with two observations, and N=44 with three) scans, across a broad developmental window (6–19 years old), allowing us to robustly chart developmental trajectories. First, we derived structural and functional axes of organisation for each child. We anticipated a primary unimodal-transmodal structural gradient dominant across development, accompanied by a primary unimodal-centred functional gradient in early childhood which is replaced by a unimodal-transmodal functional gradient at the onset of adolescence (*Dong et al., 2021*; *Xia et al., 2022*). Next, as an exploratory analysis, we charted the development of individual variability in these gradients, and statistically quantified periods of significant developmental refinement. Third, we examined the patterning of the intersection of structural and functional gradients, termed structure-function coupling, across the cortex, anticipating a unimodal-transmodal organisational axis (*Baum et al., 2020*; *Suárez et al., 2020*; *Vázquez-Rodríguez et al., 2019*). Fourth, we mapped structure-function coupling with multiple dimensions of cognition and neurodevelopmental symptomatology. Based on evidence suggesting that transmodal association cortices exhibit protracted development (*Sydnor et al., 2021*) and therefore greater developmental sensitivity, with the greatest rate of evolutionary expansion and variability (*Buckner and Krienen, 2013*; *Mueller et al., 2013*), we hypothesised that regions of the cortex with the weakest structure-function coupling, namely higher-order transmodal association cortices, would be the largest source of individual differences in psychopathology and cognition.

## Results

We leveraged diffusion-weighted imaging (DWI) and resting-state functional magnetic resonance imaging (rsfMRI) scans from two mixed-design neurodevelopmental datasets with overlapping age ranges, common cognitive and psychopathology measures, but different recruitment approaches. The first was the Nathan Kline Institute (NKI) Rockland Sample Longitudinal Discovery of Brain Development Trajectories sub-study. This is a community-ascertained sample of neurotypical children aged between 6 and 17 years old, from Rockland County, USA. We processed 447 DWI scans from 258 participants (56.82% male, 28.35% with one scan, 33.48% with two, and 38.17% with three) and 424 rsfMRI scans from 256 participants (54.35% male, 31.29% with one scan, 37.65% with two, and 31.06% with three). The second dataset was the Centre for Attention, Learning, and Memory (CALM), based in Cambridge (UK). This comprised children and adolescents aged between 6 and 19 years old. Unlike the NKI sample, the CALM sample is intentionally designed to capture the breadth of cognitive profiles present in children who present to children's professional clinical and educational services. All children in CALM were referred from educational or clinical professional services because of difficulties with cognitive, learning or behaviour. From CALM we processed 440 DWI scans from 352 participants (70% male, 60% with one scan, 40% with two) and 304 rsfMRI scans from 256 participants (66.45% male, 68.42% with one scan, 31.58% with two). See the corresponding participants section in the Methods for full sample details.

Using the Schaefer 200-node 7-network parcellation, we reconstructed individual-level and group-level structural and functional connectomes (*Figure 1a*), using best practices and several sensitivity analyses to ensure robustness (see MRI and rsfMRI processing sub-sections in the Methods). Structural connectomes were constructed using SIFT2-weighted fibre bundle capacities (FBC) from probabilistic tractography. As diffusion-map embedding requires a smooth fully-connected matrix, and in preparation for further downstream analyses examining coupling of structural and functional gradients, we transformed FBC into communicability matrices. To capture both direct and indirect paths of connectivity and communication, we generated weighted communicability matrices using SIFT2-weighted FBC. These communicability matrices reflect a graph theory measure of information transfer previously shown to maximally predict functional connectivity (*Zamani Esfahlani et al., 2022*; *Seguin*

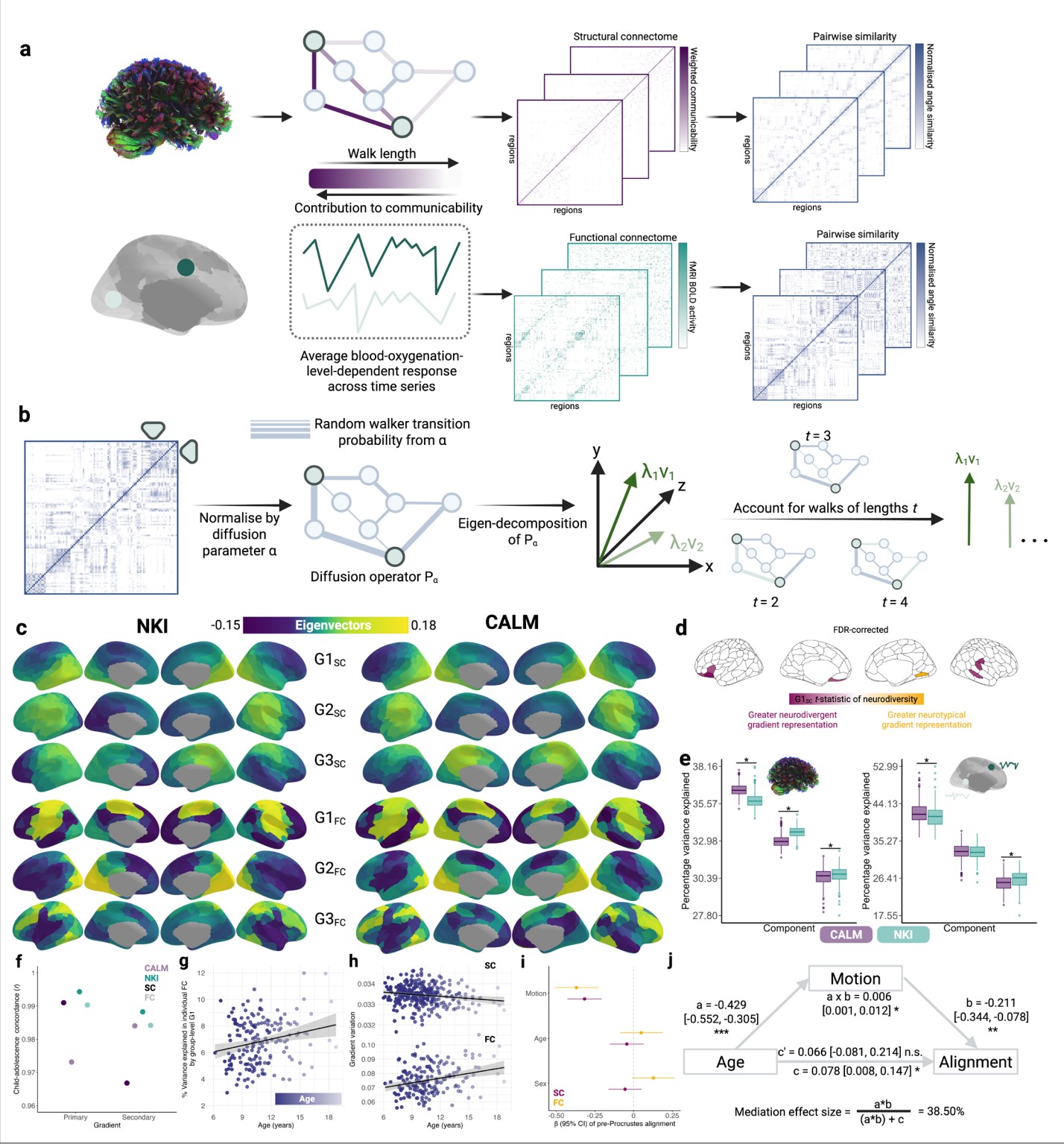

**Figure 1.** Sensitivity of structural and functional gradients to phenotypes and time. (**a**) Using the Schaefer 200-node 7-network parcellation (*Schaefer et al., 2018*), for each participant, we reconstructed structural connectomes from SIFT2-weighted FBC from probabilistic tractography and transformed them into fully connected communicability matrices. The contribution of individual connections to communicability is inversely related to path length. Functional connectomes were blood-oxygenation-level-dependent responses averaged across the time-series. Structural and functional affinity matrices, with the numbers of rows and columns corresponding to the number of cortical regions in the given parcellation, were created by calculating the normalised angle of each connectome. (**b**) Diffusion-map embedding was applied to each affinity matrix. This is normalised by the anisotropic

*Figure 1 continued*

diffusion parameter α and subjected to eigen-decomposition. The eigenvectors are sorted by decreasing amount of variance explained. When the diffusion time *t* is 0, the eigenvalues are divided by 1 minus the eigenvalues. Else, they are raised to the power of *t*. As in prior work, diffusion-map embedding was applied to each hemisphere separately, to avoid detecting a lateralised component as the principal gradient. The left hemisphere was then rotated to the right using a Procrustes rotation. (**c**) The first three structural (G1$_{SC}$ – G3$_{SC}$) and functional (G1$_{FC}$ – G3$_{FC}$) connectivity gradients for the neurotypical (NKI) and neurodivergent (CALM) data sets. (**d**) Effects of neurotypicality on nodal communicability gradient values in the baseline session of CALM, where *t*-statistics were derived from a generalised linear model contrasting referred and control portions of CALM. (**e**) The proportion of variance explained by the first 3 components for individual-level structural connectivity (left) and functional connectivity (right) gradients, respectively, as a function of data set. The middle line in the box plots represents median, flanked by the lower and upper quartiles represented by whiskers, respectively. Dots represent outliers. * indicates a significant effect of dataset (p<0.05) on variance explained within a linear mixed effects model controlling for head motion, sex, and age at scan. (**f**) Group-level gradients derived from children and adolescents are highly similar across datasets and modalities, as shown by Spearman correlation coefficients. The lighter shades reflect functional connectivity, whilst the darker shades reflect communicability gradients. (**g**) Within NKI, the proportion of variance in individual-level functional connectomes explained by the group-level functional connectome increases across development. (**h**) Across development, within baseline referred CALM individuals, the variability (standard deviation) of communicability gradients decreases, whilst variability of functional gradients increases. (**i**) Within the referred baseline portion of CALM, head motion is negatively related to pre-Procrustes alignment of individual-level structural and functional connectomes to the corresponding group gradients, with age and sex as non-significant predictors. Within all linear models, head motion and sex are controlled. (**j**) Through a mediation model, within baseline NKI, the (direct) relationship between age and pre-Procrustes alignment of functional connectomes becomes non-significant after controlling for motion, where increased motion is linked to younger age and weaker alignment.

*et al., 2022*). This also foreshadowed our structure-function coupling analyses, whereby network communication models have been shown to increase coupling strength relative to streamline counts (*Seguin et al., 2020*). Functional connectomes were reconstructed from rsfMRI blood oxygenation-level dependent (BOLD) response amplitude timeseries for each region correlated with that of all other regions. To capture inter-nodal similarity in connectivity, using a normalised angle kernel, we derived individual symmetric affinity matrices from the left and right hemispheres of each communicability and functional connectivity matrix. Varying kernels capture different but highly-related aspects of inter-nodal similarity, such as correlation coefficients, Gaussian kernels, and cosine similarity. Diffusion-map embedding is then applied on the affinity matrices to derive gradients of cortical organisation (*Figure 1b*). This non-linear dimensionality reduction technique obtains eigenvectors ('axes') along which maximal variance in structural and functional organisation, respectively, is accounted for. See 'statistical analyses and data processing' within the Methods section for full details.

## Universal patterning of communicability and functional connectivity gradients across phenotypes

We first examined how group-level gradients of structural communicability and rsfMRI differed between neurotypical (NKI) and neurodivergent (CALM) samples (*Figure 1c*). Averaged across hemispheres, the first three functional connectivity gradients accounted for 42.11%, 36.16%, and 21.73% variance in NKI, respectively, and a similar proportion in CALM (G1$_{FC}$ = 42.32%, G2$_{FC}$ = 32.08%, G3$_{FC}$ = 25.60%). In both NKI and CALM, we replicated canonical functional gradients (*Margulies et al., 2016*). That is, the principal functional gradient (G1$_{FC}$) extended from bilateral somato-motor regions towards the prefrontal cortex. Spin tests preserving spatial autocorrelation revealed that the first functional gradients in CALM and NKI were strongly and significantly correlated ($r_s$ = 0.880, $p_{spin}$ <0.001), with regions of maximal divergence between gradients, quantified as the largest absolute difference in eigenvectors, centred within bilateral somato-motor and visual networks. The second functional gradient (G2$_{FC}$) extended from bilateral visual regions towards the prefrontal cortex and frontal operculum and insula. It displayed high spatial correspondence between CALM and NKI ($r_s$ = 0.666, $p_{spin}$ = 0.001), with the largest between-group divergence centred within bilateral default mode and control networks. Finally, the third functional gradient (G3$_{FC}$) extended from bilateral dorsal attention networks towards somato-motor regions and displayed strong spatial correspondence between groups ($r_s$ = 0.753, $p_{spin}$ <0.001), with the largest between-group divergences centred within the parietal cortex, precuneus, and posterior cingulate cortex divisions of the right default mode network, alongside the right somato-motor network.

Further, just like the functional gradients, gradients of structural communicability were also organised similarly across datasets. Averaged across hemispheres, the first three communicability gradients accounted for 36.21%, 33.40%, and 30.39% variance in NKI, respectively, and a similar proportion in

CALM ($G1_{SC}$ = 36.44%, $G2_{SC}$ = 32.84%, $G3_{SC}$ = 30.72%). In NKI, $G1_{SC}$ was anchored at one end by visual regions in the right hemisphere, and by lateral and dorsal-medial prefrontal cortices at the other end. In CALM, the same gradient was anchored between bilateral prefrontal cortices and right visual networks and displayed strong significant spatial correspondence to NKI ($r_s$ = 0.830, $p_{spin}$ <0.001). The largest between-group divergence occurred within the left orbitofrontal cortex and precuneus divisions of the control network, and within the precuneus and prefrontal cortex divisions of the left default-mode network. The second communicability gradient ($G2_{SC}$) in NKI extended from visual regions and the precuneus in the left hemisphere towards regions in the dorsal attention network and temporo-occipital-parietal lobes in the right hemisphere. The corresponding gradient in CALM displayed high spatial correspondence ($r_s$ = 0.848, $p_{spin}$ <.001), with the largest divergence between gradients centred in the left prefrontal cortex and control networks. Finally, the third communicability gradient ($G3_{SC}$) in NKI spanned prefrontal and somato-motor cortices and was statistically significantly and strongly correlated with the corresponding CALM gradient ($r_s$ = 0.719, $p_{spin}$ <0.001), which was anchored at one end by the right temporal cortex of the default mode network and at the other end by divisions of the left default mode, control, and ventral attention systems. The largest between-group divergence occurred primarily within the parietal and temporal cortex divisions of the left default-mode network. Together, gradients of functional connectivity and communicability displayed similar organisational principles across datasets and phenotypes.

Despite strong spatial correspondence of gradients between datasets, alignment was not perfect.

We investigated the hypothesis that this 'misalignment' was driven by technical site variation or recruitment protocol differences. Helpfully, the CALM study also incorporated a smaller comparison sample, comprising age-matched children recruited from the same schools and neighbourhoods as those referred to the main cohort. This extra non-referred sample provides an opportunity to disentangle these explanations, since they were scanned on the same machine and protocol as the main CALM cohort. We derived group-level communicability and functional connectivity gradients across 91 structural and 79 structural scans, respectively, from CALM comparison children. Gradients in the non-referred sample were almost identical to the referred sample, both in terms of communicability ($G1_{SC}$ $r_s$ = 0.999; $G2_{SC}$ $r_s$ = 0.991; $G3_{SC}$ $r_s$ = 0.995, all $p_{spin}$ <0.001) and functional connectivity ($G1_{FC}$ $r_s$ = 0.983; $G2_{FC}$ $r_s$ = 0.986; $G3_{FC}$ $r_s$ = 0.968, all $p_{spin}$ <0.001). In other words, any small differences in alignment between NKI and CALM are unlikely to be the result of the phenotypic differences between the cohorts. Across datasets with varying phenotypes and scanning protocols, at least at a group-level averaged across development during childhood and adolescence, the same broad organisational principles of communicability and functional connectivity emerge.

This CALM comparison group also allowed us to test whether the principal gradients differed at all between referred and comparison children scanned at the same site. To examine the effect of referral status on participant-level nodal gradient values in CALM, we conducted a series of generalised linear models controlling for head motion, sex, and age at scan (*Figure 1d*). Since the diffusion distances plotted onto the cortical surface represent positions in the 3D coordinate system, gradient scores may reflect the contribution or differentiation of connectivity of that specific region to the low-dimensional manifold. We restricted our analyses to baseline scans to reduce the difference in sample size for the referred (311 communicability and 177 functional gradients, respectively) and control participants (91 communicability and 79 functional gradients, respectively) and to the principal gradients. For communicability, 42 regions showed a significant effect (p<0.05) of neurodivergence before FDR correction, with 9 post FDR correction. 8 of these 9 regions had negative *t*-statistics, suggesting a reduced nodal gradient value and representation in the neurodivergent children, encompassing both lower-order somatosensory cortices alongside higher-order fronto-parietal and default-mode networks. The largest reductions were observed within the prefrontal cortices of the default-mode network (*t*=–3.992, p=6.600 x $10^{-5}$, $p_{FDR}$ = 0.013, Cohen's *d*=–0.476), the left orbitofrontal cortex of the limbic network (*t*=–3.710, p=2.070 x $10^{-4}$, $p_{FDR}$ = 0.020, Cohen's *d*=–0.442), and right somato-motor cortex (*t*=–3.612, p=3.040 x $10^{-4}$, $p_{FDR}$ = 0.020, Cohen's *d*=–0.431). The right visual cortex was the only exception, with stronger gradient representation within the neurotypical cohort (*t*=3.071, p=0.002, $p_{FDR}$ = 0.048, Cohen's *d*=0.366). For functional connectivity, comparatively fewer regions exhibited a significant effect (p<0.05) of neurotypicality, with 34 regions prior to FDR correction and 1 post. Significantly stronger gradient representation was observed in neurotypical children within the right precentral ventral division of the default-mode network (*t*=3.930, p=8.500 x $10^{-5}$, $p_{FDR}$ = 0.017,

Cohen's $d$=0.532). Together, this suggests that the strongest and most robust effects of neurodivergence are observed within gradients of communicability, rather than functional connectivity, where alterations in both affect higher-order associative regions.

To assess the effect of sparsity on the derived gradients, we examined group-level structural (N=222) and functional (N=213) connectomes from the baseline session of NKI. The first three functional connectivity gradients derived using the full connectivity matrix (density = 92%) were highly consistent with those obtained from retaining the strongest 10% of connections in each row ($r_1$=0.999, $r_2$=0.998, $r_3$ <0.999, all p<0.001). Likewise, the first three communicability gradients derived from retaining all counts from SIFT2-weighted FBC (density = 83%) were almost identical to those obtained from 10% row-wise thresholding ($r_1$=0.994, $r_2$=0.963, $r_3$=0.955, all p<0.001). This suggests that the reported gradients are driven by the strongest or most consistent connections within the connectomes, with minimal additional information provided by weaker connections. In terms of functional connectivity, such consistency reinforces past work demonstrating that the sensorimotor-to-association axis, the major axis within the principal functional connectivity gradient, emerges across both the top- and bottom-ranked functional connections (*Nenning et al., 2023*).

Finally, as a sensitivity analysis, to determine the effect of communicability on the gradients, we derived affinity matrices for both datasets using a simpler measure: the log of raw FBC. The first three FBC-derived components compared to communicability were highly consistent across both NKI ($r_s$ = 0.791, $r_s$ = 0.866, $r_s$ = 0.761) and the referred subset of CALM ($r_s$ = 0.951, $r_s$ = 0.809, $r_s$ = 0.861), suggesting that in practice the organisational gradients are highly similar regardless of the SC metric used to construct the affinity matrices.

## Stability of individual-level gradients across developmental time

After establishing group-level gradients, we applied diffusion-map embedding to 887 individual structural and 728 functional connectomes, across datasets and timepoints. Within modalities and components, gradients explained a similar proportion of variance across datasets. The median (range) variance explained by the first three structural components were 35.75% (34.55–38%), 33.61% (32.44–35%), and 30.67% (27.8–32%), respectively, in NKI, and 36.50% (35.22–38%), 32.94% (31.83–35%), and 30.52% (28.03%–32%) in CALM. For the functional gradients, the median (range) variance explained by the first three components were 41.12% (35.61–53%), 33.61% (32.44–35%), and 30.67% (27.8–32%), respectively, in NKI, and 41.60% (36.11–51%), 32.77% (25.95–38%), and 25.40% (20.07–31%) in CALM. In other words, across the two datasets, there is a remarkable consistency in the ordering and the variance explained at an individual level, for both structural and functional gradients.

But does the ordering of these gradients change with developmental time? *Dong et al., 2021* demonstrated that the primary (unimodal) functional gradient accounted for 38% variance in FC in childhood but only 11% in adolescence, whilst the primary (association) functional gradient accounted for 38% variance in adolescence but only 12% in childhood. We were unable to replicate this effect (*Figure 1e*). Instead, across datasets, we observed that the proportion of variance explained by the principal structural and functional gradients, respectively, remained highly stable over time, without any sudden change in variance explained at the onset of adolescence. This was reflected by the relatively small range in variance explained, rather than the extensive range expected by a sudden shift in gradient dominance across development and datasets. Put simply, not only is the ordering and the patterning of the gradients highly similar across both datasets, the structural and functional gradients of brain organisation were established early in life and are highly stable across childhood and adolescence.

One possibility is that our observation of gradient stability – rather than a swapping of the order for the first two gradients (*Dong et al., 2021*) – is because we calculated them at an individual level. To test this, we created subgroups and contrasted the first two group-level structural and functional gradients derived from children (younger than 12 years old) versus those from adolescents (12 years old and above), using the same age groupings as prior work (*Dong et al., 2021*). If our use of individually calculated gradients produces the stability, then we should observe the swapping of gradients in this sensitivity analysis (*Figure 1f*). Using baseline scans from NKI, the primary structural gradient in childhood (N=99) was highly correlated ($r_s$ = 0.995) with those derived from adolescents (N=123). Likewise, the secondary structural gradient in childhood was highly consistent in adolescence ($r_s$ = 0.988). In terms of functional connectivity, the principal gradient in childhood (N=88) was highly consistent

in adolescence ($r_s$ = 0.990, N=125). The secondary gradient in childhood was again highly similar in adolescence ($r_s$ = 0.984). The same result occurred in the CALM dataset: In the baseline referred subset of CALM, the primary and secondary communicability gradients derived from children (N=258) and adolescents (N=53) were near-identical ($r_s$ = 0.991 and $r_s$ = 0.967, respectively). Alignment for the primary and secondary functional gradients derived from children (N=130) and adolescents (N=43) were also near-identical ($r_s$ = 0.972 and $r_s$ = 0.983, respectively). These consistencies across development suggest that gradients of communicability and functional connectivity established in childhood are the same as those in adolescence, irrespective of group-level or individual-level analysis. Put simply, our failure to replicate the swapping of gradient order in *Dong et al., 2021* is not the result of calculating gradients at the level of individual participants.

To evaluate the consistency of gradients across development, across baseline participants with functional connectomes from the referred CALM cohort (N=177), we calculated the proportion of variance in individual-level connectomes accounted for by group-level functional gradients. Specifically, we calculated the proportion of variance in an adjacency matrix $A$ accounted for by the vector $v_i$ as the fraction of the square of the scalar projection of $v_i$ onto $A$, over the Frobenius norm of $A$. Using a generalised linear model, we then tested whether the proportion of variance explained varies systematically with age, controlling for sex and head-motion. The variance in individual-level functional connectomes accounted for by the group-level principal functional gradient gradually increased with development ($\beta$=0.111, 95% CI = [0.022, 0.199], p=1.452 x $10^{-2}$, Cohen's $d$=0.367), as shown in *Figure 1g*, and decreased with higher head motion ($\beta$=−10.041, 95% CI = [-12.379,−7.702], p=3.900 x $10^{-17}$), with no effect of sex ($\beta$=0.071, 95% CI = [−0.380, 0.523], p=0.757). We observed no developmental effects on the variance explained by the second ($r_s$ = 0.112, p=0.139) or third ($r_s$ = 0.053, p=0.482) group-level functional gradient. When repeated with the baseline functional connectivity for NKI (N=213), we observed no developmental effects ($\beta$=0.097, 95% CI = [−0.035, 0.228], p=0.150) on the variance explained by the principal functional gradient after accounting for motion ($\beta$=−3.376, 95% CI = [−8.281, 1.528], p=0.177) and sex ($\beta$=−0.368, 95% CI = [−1.078, 0.342], p=0.309). However, we observed significant developmental correlations between age and variance ($r_s$ = 0.137, p=0.046) explained *before* accounting for head motion and sex. We observed no developmental effects on the variance explained by the second functional gradient ($r_s$ = −0.066, p=0.338), but a weak negative developmental effect on the variance explained by the third functional gradient ($r_s$ = −0.189, p=0.006). Note, however, the magnitude of the variance accounted for by the third functional gradient was very small (all <1%). When applied to communicability matrices in CALM, the proportion of variance accounted for by the group-level communicability gradient was negligible (all <1%), precluding analysis of developmental change.

To further probe the consistency of gradients across development, we examined developmental changes in the standard deviation of gradient values, corresponding to heterogeneity, following prior work examining morphological (*He et al., 2025*) and functional connectivity gradients (*Xia et al., 2022*). Using a series of generalised linear models within the baseline referred subset of CALM, correcting for head motion and sex, we found that gradient variation for the principal functional gradient increased across development ($\beta$=0.219, 95% CI = [0.091, 0.347], p=0.001, Cohen's $d$=0.504), indicating greater heterogeneity (*Figure 1h*), whilst gradient variation for the principal communicability gradient decreased across development ($\beta$=−0.154, 95% CI = [-0.267,−0.040], p=0.008, Cohen's $d$=−0.301), indicating greater homogeneity (*Figure 1h*). Note, a paired *t*-test on the 173 common participants demonstrated a significant effect of modality on gradient variability ($t$(172) = −56.639, p=3.663 x $10^{-113}$), such that the mean variability of communicability gradients ($M$=0.033, SD = 0.001) was less than half that of functional connectivity ($M$=0.076, SD = 0.010). Together, this suggests that principal functional connectivity and communicability gradients are established early in childhood and display age-related *refinement*, but not replacement.

A second possible discrepancy between our results and that of prior work examining developmental change in group-level functional gradients (*Dong et al., 2021*) was the use of Procrustes alignment. Such alignment of individual-level gradients to group-level templates is a necessary step to ensure valid comparisons between corresponding gradients across individuals and has been implemented in sliding-window developmental work tracking functional gradient development (*Xia et al., 2022*). Nonetheless, we tested whether our observation of stable principal functional and communicability gradients may be an artefact of the Procrustes rotation. We did this by modelling how individual-level

alignment without Procrustes rotation to the group-level templates varies with age, head motion, and sex, as a series of generalised linear models. We included head motion as the magnitude of the Procrustes rotation has been shown to be positively correlated with mean framewise displacement (*Sasse et al., 2024*) and prior group-level work (*Dong et al., 2021*) included an absolute motion threshold rather than continuous motion estimates. As shown in *Figure 1i*, using the baseline referred CALM sample, there was no significant relationship between alignment and age ($\beta$=–0.044, 95% CI = [–0.154, 0.066], p=0.432) after accounting for head motion and sex. Interestingly, however, head motion was significantly associated with alignment ($\beta$=–0.318, 95% CI = [–0.428, –0.207], p=1.731 x $10^{-8}$), such that greater head motion was linked to weaker alignment. Note that older children tended to have exhibit less motion for their structural scans ($r_s$ = –0.335, p<0.001). We observed similar trends in functional alignment, whereby tighter alignment was significantly predicted by lower head motion ($\beta$=–0.370, 95% CI = [-0.509, –0.231], p=1.857 x $10^{-7}$), but not by age ($\beta$=0.049, 95% CI = [–0.090, 0.187], p=0.490). Note that age and head motion for functional scans were not significantly related ($r_s$ = –0.112, p=0.137). When repeated for the baseline scans of NKI, alignment with the principal structural gradient was not significantly predicted by either scan age ($\beta$=0.019, 95% CI = [–0.124, 0.163], p=0.792) or head motion ($\beta$=–0.133, 95% CI = [–0.175, 0.009], p=0.067) together in a single model, where age and motion were negatively correlated ($r_s$ = –0.355, p<0.001). Alignment with the principal functional gradient was significantly predicted by head motion ($\beta$=–0.183, 95% CI = [-0.329, –0.036], p=0.014) but not by age ($\beta$=0.066, 95% CI = [–0.081, 0.213], p=0.377), where age and motion were also negatively correlated ($r_s$ = –0.412, p<0.001). Across modalities and datasets, alignment with the principal functional gradient in NKI was the only example in which there was a significant correlation between alignment and age ($r_s$ = 0.164, p=0.017) before accounting for head motion and sex. This suggests that apparent developmental effects on alignment are minimal, and where they do exist they are removed after accounting for head motion. Put together, this suggests that the lack of order swapping for the first two gradients is not the result of the Procrustes rotation – even without the rotation there is no evidence for swapping.

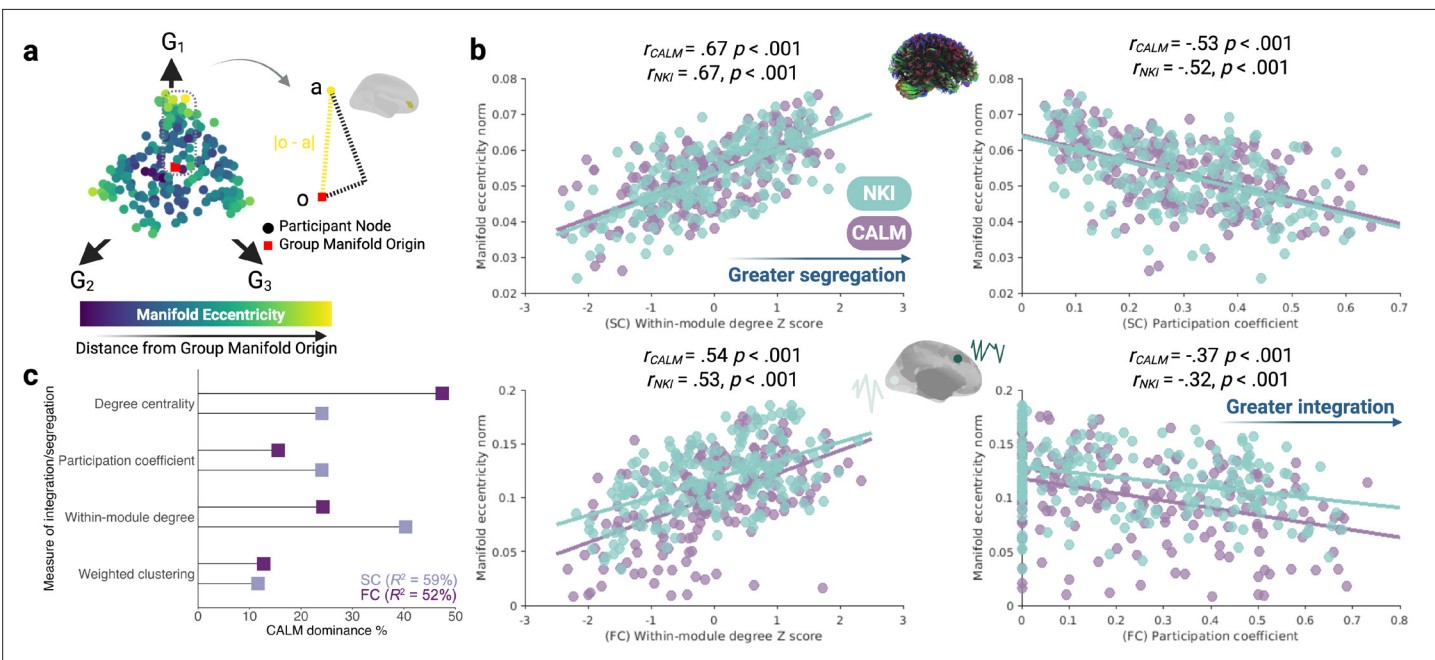

**Figure 2.** Validating manifold eccentricity as a measure of variability in structural and functional axis organisation. (**a**) Following prior work (*Park et al., 2021*), we defined manifold eccentricity as the Euclidean distance between each node's three-dimensional embedding in manifold space and the group manifold origin, for a given participant and modality. The structural embedding, across the first three structural gradients ($G_1 – G_3$) for a representative NKI participant is shown. (**b**) Across datasets and modalities, increased manifold eccentricity is associated with increased network segregation. Each point represents a node, with CALM visualised in purple and NKI in green. (**c**) A dominance analysis for the group-level structural and functional connectomes in CALM reveals distinct and modality-dependent contributions of measures of integration to manifold eccentricity.

To emphasise the importance of head motion in the appearance of developmental change in alignment, we examined whether accounting for head motion removes any apparent developmental change within NKI. Specifically, we tested whether head motion mediates the relationship between age and alignment (*Figure 1j*), controlling for sex, given that higher motion is associated with younger children ($\beta$=–0.429, 95% CI = [-0.552,–0.305], p=7.957 x $10^{-11}$), and stronger alignment is associated with reduced motion ($\beta$=–0.211, 95% CI = [-0.344, –0.078], p=2.017 x $10^{-3}$). Motion indirectly mediated the relationship between age and alignment ($\beta$=0.078, 95% CI = [0.006, 0.146], p=1.200 x $10^{-2}$), accounting for 38.5% variance in the age-alignment relationship, such that the link between age and alignment became non-significant after accounting for motion ($\beta$=0.066, 95% CI = [–0.081, 0.214], p=0.378). This firstly confirms our GLM analyses, where we control for motion and find no age associations. Moreover, this suggests that caution is required when associations between age and gradients are observed. In our analyses, because we calculate individual gradients, we can correct for individual differences in head motion in all our analyses.

## Manifold eccentricity as a measure of individual variability in gradient organisation

To examine variability in axis organisation across time, we used manifold eccentricity as a dependent variable (*Figure 2a*). For each modality (two levels: SC and FC) and dataset (three levels: neurotypical CALM, neurodivergent CALM, and NKI), we computed the group manifold origin as the mean of their respective first three gradients. Because of the normal nature of the manifolds this necessarily means that these origin points will be very near-zero, but we include the exact values in the 'Manifold Eccentricity' methodology sub-section. For each participant's 3D embedding in manifold space, we calculated the Euclidean distance between each node and the group manifold origin (the intersection between the three group-level gradients for a given modality). This measure has previously been reported in structural connectivity (*Park et al., 2021*), and a similar measure in functional connectivity (*Bethlehem et al., 2020*).

To better understand what manifold eccentricity measures in terms of connectome organisation, we investigated its relationship with nodal graph metrics (*Figure 2b*). We included two graph theory metrics previously associated with structural manifold eccentricity in a neurotypical sample of children and adolescents: participation coefficient, describing the uniformity of dispersion of edges amongst different modules (*Guimerà and Nunes Amaral, 2005*), and within-module degree Z-score, a standardised measure of the number of intra-modular connections. Across datasets and modalities, higher manifold eccentricity was significantly positively correlated with within-module degree *Z*-scores but negatively correlated with participation coefficient. The consistency across datasets and modalities suggests that higher manifold eccentricity corresponds to greater segregation. To further contextualise manifold eccentricity in terms of integration and segregation beyond simple correlations, we conducted a multivariate dominance analysis (*Budescu, 1993*) of four graph theory metrics of segregation as predictors of nodal normalised manifold eccentricity within the group-level referred CALM structural and functional connectomes (*Figure 2c*). A dominance analysis assesses the relative importance of each predictor in a multilinear regression framework by fitting $2^n - 1$ models (where n is the number of predictors) and calculating the relative increase in adjusted R2 caused by adding each predictor to the model across both main effects and interactions. A multilinear regression model including weighted clustering coefficient, within-module degree Z-score, participation coefficient and normalised degree centrality accounted for 59% of variance in nodal manifold eccentricity in the group-level CALM structural connectome. Within-module degree Z score was the most important predictor (40.31% dominance), almost twice that of the participation coefficient (24.03% dominance) and normalised degree centrality (24.05% dominance) which made roughly equal contributions. The least important predictor was the weighted clustering coefficient (11.62% dominance). When the same approach was applied for the group-level referred CALM functional connectome, the four predictors accounted for 52% variability. However, in contrast to the structural connectome, functional manifold eccentricity seemed to incorporate the same graph theory metrics in different proportions. Normalised degree centrality was the most important predictor (47.41% dominance), followed by within-module degree Z-score (24.27%), and then the participation coefficient (15.57%) and weighted clustering coefficient (12.76%) which made approximately equal contributions. Thus, whilst structural manifold eccentricity was dominated most by within-module degree Z-score and

least by the weighted clustering coefficient, functional manifold eccentricity was dominated most by normalised degree centrality and least by the weighted clustering coefficient. This suggests that manifold mapping techniques incorporate different aspects of integration dependent on modality. Together, manifold eccentricity acts as a composite measure of segregation, being differentially sensitive to different aspects of segregation, without necessitating a priori specification of graph theory metrics. Further discussion of the value of gradient-based metrics in developmental contexts and as a supplement to traditional graph theory analyses is provided in the 'Manifold Eccentricity' methodology sub-section.

## Developmental trajectories of structural manifold contraction and functional manifold expansion

Next, we charted age-related changes in organisation of structural and functional manifolds, operationalised as manifold eccentricity, using generalised additive mixed models (GAMMs). We did this globally as well as for seven intrinsic connectivity networks (ICNs; *Yeo et al., 2011*). Within each GAMM, age and an age-dataset interaction were smooth covariates, whilst mean framewise displacement, sex, and dataset were parametric linear covariates. We included a random effect for each participant to account for longitudinal data. Dataset was significantly associated with structural manifold eccentricity across all levels of analysis (*Figure 3a*), with the largest cohort effects globally ($t=-74.351$, $p_{FDR} <0.001$) and within the default-mode network ($t=-42.476$, $p_{FDR} <0.001$), driven by *higher* manifold eccentricity in CALM than NKI. Further, remodelling of structural manifolds was consistently and significantly associated with development globally and across all networks, except the somato-motor network. Structural manifold eccentricity decreased with age, indicating greater integration. The strongest developmental effects were observed globally ($F=52.644$, $p_{FDR} <0.001$), alongside limbic ($F=29.486$, $p_{FDR} <0.001$) and dorsal attention ($F=27.148$, $p_{FDR} <0.001$) networks. The non-linear GAMM procedure also produces a measure of the non-linearity of an effect, or its effective degrees of freedom (EDF), whereby an EDF of 1 indicates a linear effect, with larger deviations from 1 indicating greater non-linearity. This metric revealed considerable differences in the nature of the developmental effect on structural manifold eccentricity, ranging from linear globally and within the frontal-parietal network (EDF = 1.000), to non-linear within the visual (EDF = 2.434) and dorsal attention (EDF = 2.056) networks.

In line with previous work (*Sydnor et al., 2023*; *Tervo-Clemmens et al., 2023*; *Tooley et al., 2023*), but not yet explored in terms of multi-modal developmental neuroimaging data, we precisely quantified significant periods of developmental refinement by calculating the first derivative of the smooth age term for each dataset separately, based on the original GAMM specification and data used for model fitting subset by dataset. We then extracted developmental periods in which the simultaneous confidence intervals excluded zero ($p<0.05$, two-sided), computed for each unique age in each modality and dataset. Developmental effects on structural manifold eccentricity were statistically significant across the entire developmental age range for both CALM and NKI globally, alongside within dorsal attention, ventral attention, and limbic networks. Within the visual network, developmental effects were statistically significant until mid-adolescence, in both CALM (6.00–14.58 years) and NKI (6.68–15.27 years). Within the frontoparietal network, age-related change was significant across the entire developmental period for CALM (6.00–19.17 years), but not statistically significant for NKI. A similar picture emerged for the default-mode network, where age-related change was statistically significant for the entire developmental range in CALM (6.00–19.17 years), but only for a brief adolescent period within NKI (14.04–14.59 years). This suggests that within higher-order frontoparietal and default-mode networks, structural manifold remodelling was reliably associated with age within children referred for problems with attention, learning, or memory, and this was distinct from the developmental trajectory followed by children within a broader community sample. Finally, we examined interactions between age and dataset. Such interactions were significantly associated with structural manifold eccentricity globally ($F=11.665$, $p_{FDR} = 0.001$), alongside default-mode ($F=30.074$, $p_{FDR} <0.001$) and limbic ($F=1.504$, $p_{FDR} = 0.046$) networks. Across all three levels of analysis, effects were driven by a stronger age effect within CALM ($F_{Global} = 37.589$, $p<0.001$; $F_{Default} = 27.655$, $p<0.001$; $F_{Limbic} = 34.867$, $p<0.001$) than NKI ($F_{Global} = 18.045$, $p<0.001$; $F_{Default} = 1.305$, $p=0.222$; $F_{Limbic} = 14.863$, $p<0.001$). In summary, the structural manifold is gradually contracting with developmental time, most markedly in the limbic and dorsal attention systems. Manifold eccentricity is higher in neurodivergent

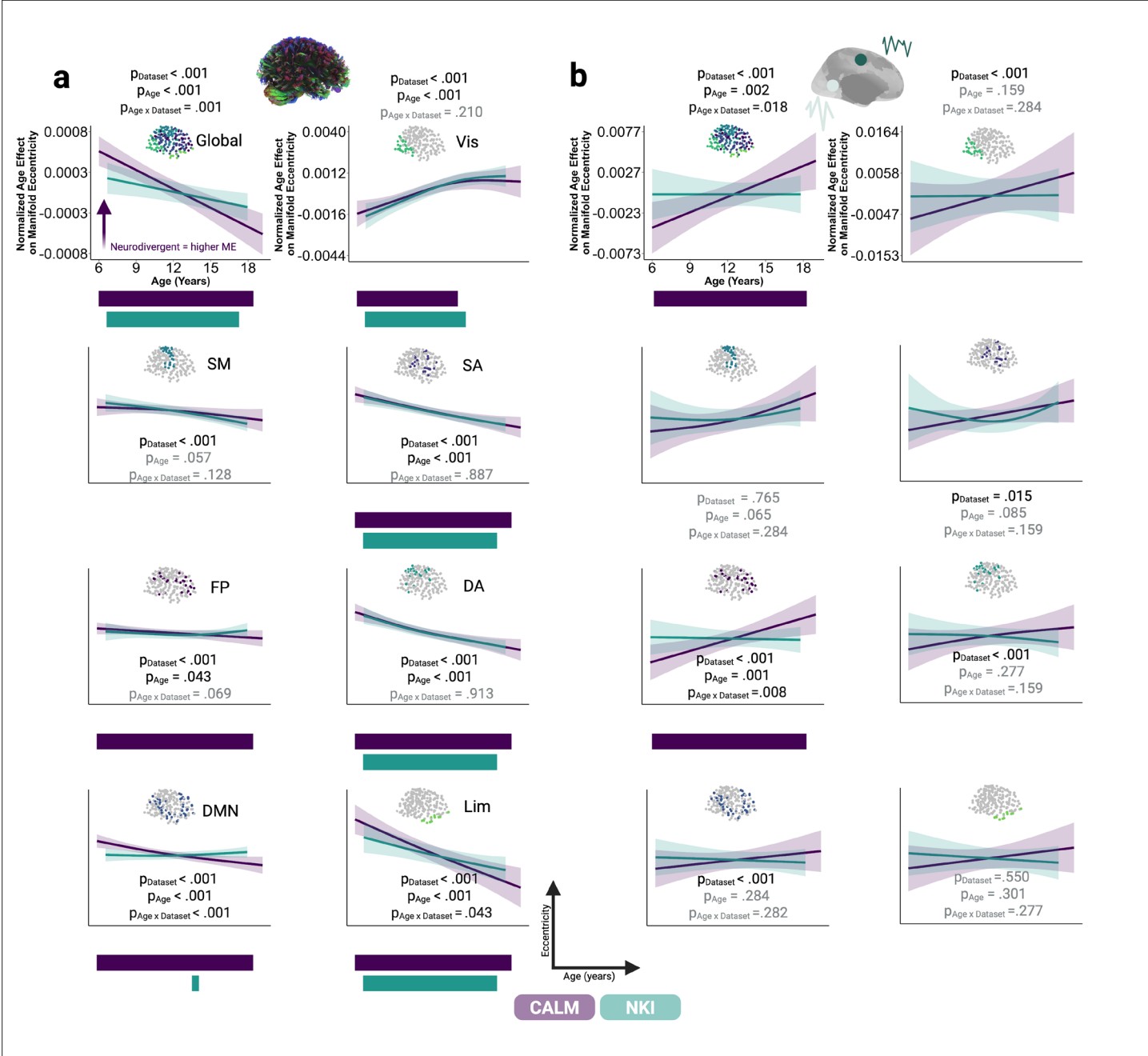

**Figure 3.** Sensitivity of structural and functional manifold organisation to time and phenotype. Normalised factor-smooth interaction between dataset (CALM in purple, NKI in green) and age when predicting (**a**) structural or (**b**) functional manifold eccentricity, at global and 7 intrinsic connectivity network levels (*Yeo et al., 2011*), respectively, within generalised additive mixed models (GAMM). Within each GAMM, age was a smooth covariate, whilst mean framewise displacement, sex, and dataset were parametric covariates. Age-dataset interactions are plotted to control for the main effect of cohort. 95% credible intervals were extracted by sampling the posterior distribution of the age-dataset interaction 10,000 times. Horizontal bars within each sub-plot represent developmental periods in which the first derivative of the age-dataset interaction was statistically significant, that is, when the associated simultaneous confidence intervals did not include zero (p<0.05, two-tailed). The direction and colour of the arrow for structural global manifold eccentricity indicates a significant main effect of neurodiversity, such that structural global manifold eccentricity was significantly higher within the referred portion of CALM at baseline (N=313) than within neurotypical participants (N=222 from NKI, and N=91 from control portion of CALM). First derivatives for the smooth age term were plotted for GAMMs with a significant age-dataset interaction or age effect. The y-axis scale for the visual manifold eccentricity GAMM applies to all other intrinsic connectivity network plots. All p-values were corrected for multiple comparisons, within modalities, by controlling false discovery rates. Grey indicates effects not statistically significant at p≤0.05. Vis, visual; SM, somato-motor; SA, salient/ventral attention; FP, fronto-parietal; DA, dorsal attention; DMN, default-mode network; Lim, Limbic.

than neurotypical participants, but differential age effects suggest that the ongoing contraction is more marked in neurodivergent participants.

Having demonstrated significant contraction of structural manifolds and sensitivity to development and dataset, we repeated the same modelling procedure with functional manifold eccentricity as the outcome (*Figure 3b*). Dataset was significantly associated with functional manifold contraction globally ($t=-6.975$, $p_{FDR}$ <0.001, Mean$_{CALM}$ = 0.108, Mean$_{NKI}$ = 0.104), and in visual ($t=-4.066$, $p_{FDR}$ <0.001, Mean$_{CALM}$ = 0.127, Mean$_{NKI}$ = 0.120), dorsal attention ($t=-4.903$, $p_{FDR}$ <0.001, Mean$_{CALM}$ = 0.103, Mean$_{NKI}$ = 0.097), ventral attention ($t=-2.906$, $p_{FDR}$ = 0.015, Mean$_{CALM}$ = 0.100, Mean$_{NKI}$ = 0.098), fronto-parietal ($t=-4.712$, $p_{FDR}$ <0.001, Mean$_{CALM}$ = 0.100, Mean$_{NKI}$ = 0.096), and default-mode ($t=-8.885$, $p_{FDR}$ <0.001, Mean$_{CALM}$ = 0.111, Mean$_{NKI}$ = 0.101) networks. Across these partitions, CALM had significantly higher functional manifold eccentricity, albeit to a small degree, with the largest cohort effects globally and within the default-mode network, indicating greater segregation. Whilst structural manifolds were sensitive to development across almost all levels of analysis, we only observed significant age-related functional manifold expansion globally ($F=13.369$, $p_{FDR}$ = 0.002) and within the fronto-parietal network ($F=14.627$, $p_{FDR}$ = 0.001). Within both levels of analysis, statistically significant first derivatives of age revealed significant age-related change across the whole developmental period in CALM (6.17–19.17 years), but no significant change in NKI. Finally, we observed significant age-dataset interactions globally ($F=7.971$, $p_{FDR}$ = 0.018) and within the fronto-parietal network ($F=9.816$, $p_{FDR}$ = 0.008). In both cases, CALM exhibited larger age effects ($F_{Global}$ = 6.276, p=0.013; $F_{Fronto-parietal}$=8.567, p=0.004) than NKI ($F_{Global}$ = 2.934, p=0.087; $F_{Fronto-parietal}$=2.122, p=0.202). Together, we demonstrate that structural manifolds *contract* across childhood and adolescence, indicating greater integration whilst, in parallel, functional manifolds *expand*, indicating greater segregation. Further, structural manifold contraction shows more widespread effects of development and cohort than functional manifold expansion.

Much as with the gradients themselves, we suspected that much of the simple main effect of dataset could reflect the scanner/site, rather than the difference in phenotype. Again, we drew upon the CALM comparison children to help us disentangle these two explanations. As a sensitivity analysis to parse effects of neurotypicality and dataset on manifold eccentricity, we conducted a series of generalised linear models with mean global and network-level manifold eccentricity as the outcome measure, for each modality. We did this across all the baseline data (i.e. including the neurotypical comparison sample for CALM) using neurotypicality (two levels: neurodivergent or neurotypical), site (two levels: CALM or NKI), sex, head motion, and age at scan (*Figure 3*). We restricted our analysis to baseline scans to create more equally balanced groups. In terms of structural manifold eccentricity (N=313 neurotypical including 91 control CALM participants and 222 NKI participants, N=311 neurodivergent), we observed higher manifold eccentricity in the neurodivergent participants at a global level ($\beta$=0.090, p=0.019, Cohen's $d$=0.188) but the individual network level effects did not survive the multiple comparison correction necessary for looking across all seven networks, with the default-mode network being the strongest ($\beta$=0.135, p=0.027, $p_{FDR}$ = 0.109, Cohen's $d$=0.177). There was no significant effect of neurodiversity on functional manifold eccentricity (N=292 neurotypical including 79 control CALM participants and 213 NKI participants, N=177 neurodivergent). This suggests that neurodiversity is significantly associated with structural manifold eccentricity, over and above differences in site, but we cannot distinguish these effects reliably in the functional manifold data.

## Structure-function coupling follows a unimodal-transmodal axis, with developmental effects concentrated within higher-order association networks

Thus far, we tracked the organisation of structural and functional manifolds separately. However, extensive evidence points towards structural connectivity providing a backbone for functional brain organisation (*Hagmann et al., 2008*; *Honey et al., 2007*; *Honey et al., 2009*). To examine the relationship between each node's relative position in structural and functional manifold space, we turned our attention to structure-function coupling. Whilst prior work typically computed coupling using raw streamline counts and functional connectivity matrices, either as a correlation (*Baum et al., 2020*) or through a multiple linear regression framework (*Vázquez-Rodríguez et al., 2019*), we opted to directly incorporate low-dimensional embeddings within our coupling framework. Specifically, as opposed to correlating row-wise raw functional connectivity with structural connectivity eigenvectors (*Park*

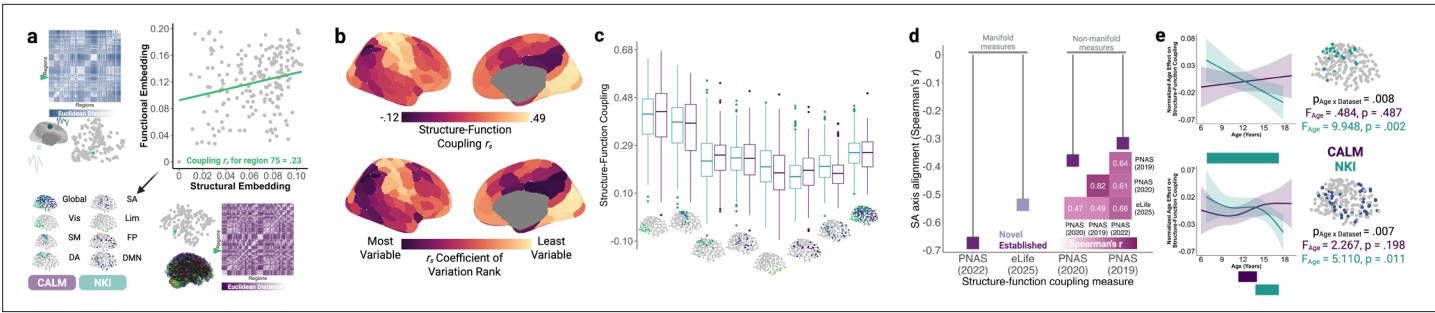

**Figure 4.** Magnitude and variability of structure-function coupling are spatially patterned along a unimodal-transmodal axis, are sensitive to both development and dataset, and have statistically significant developmental trajectories centred within higher-order association networks. (**a**) To derive a nodal measure of structure-function coupling, for each region (green) we calculated the Euclidean distance with all other regions, within structural and functional manifolds separately, producing two 1x200 vectors, using the Schaefer 200-node 7-network parcellation (*Schaefer et al., 2018*). The Spearman rank correlation coefficient ($r_s$) between these two vectors was the structure-function coupling measure. The larger the coupling coefficient, the more similar the embedding of each node in structural and functional manifold space, relative to all other networks. Note that a negative coupling value indicates anticorrelation. Structural and functional embeddings for a representative NKI participant are shown. Coupling was calculated at 8 levels of analysis: globally, and for 7 intrinsic connectivity networks (*Yeo et al., 2011*). (**b**) Both structure-function coupling magnitude (top) and inter-individual variability (bottom) are patterned along a unimodal-transmodal axis, where the largest and least variable coupling is at the unimodal anchor, whilst the smallest and most variable coupling is at the transmodal anchor. Note that regions were ranked according to absolute coefficient of variation. (**c**) Globally and within networks, structure-function coupling is consistent across datasets. Within each box plot, the box represents the lower quartile, median, and upper quartile, respectively. Circular points represent outliers. (**d**) Alignment of novel and established absolute structure-function coupling metrics with the sensorimotor-association (SA) axis are visualised. The inset Spearman correlation plot of the four coupling measures shows moderate-to-strong correlations ($p_{spin}$ <0.005 for all spatial correlations). The accompanying lollypop plot shows the alignment between the sensorimotor-to-association axis and each of the four coupling measures, with the novel measure coloured in light purple ($p_{spin}$ <0.007 for all spatial correlations). (**e**) Within a GAMM of global or network-level structure-function coupling as the outcome and using age and an age-dataset interaction as smooth covariates, alongside framewise displacement (averaged across modalities), sex, and dataset as parametric covariates, the age-dataset interaction was significantly linked to coupling in dorsal attention ($p_{FDR}$ = 0.008) and default-mode ($p_{FDR}$ = 0.007) networks. Age-dataset interactions are plotted to control for the main effect of cohort. Horizontal bars represent periods of significant developmental refinement of structure-function coupling, calculated as statistically significant (non-zero confidence intervals) simultaneous first derivatives of age effects within GAMMs conducted separately for each dataset. Within both network partitions, age effects were stronger in NKI than CALM. Across all sub-plots, CALM is visualised in purple, and NKI in green. p-values were corrected for multiple comparisons by controlling false discovery rates, except for the main effects of age within separate dataset-specific GAMMs. Vis, visual; SM, somato-motor; SA, salient/ventral attention; FP, fronto-parietal; DA, dorsal attention; DMN, default-mode network; Lim, Limbic.

*et al., 2022*), our metric directly incorporates the relative position of each node in low-dimensional structural and functional manifold spaces. Each node was situated in a low-dimensional 3D space, the axes of which were each participant's gradients, specific to each modality. For each participant and each node, we computed the Euclidean distance with all other nodes within structural and functional manifolds separately, producing a vector of size 200x1 per modality. The nodal coupling coefficient was the Spearman correlation between each node's Euclidean distance to all other nodes in structural manifold space, and that in functional manifold space (*Figure 4a*). Put simply, a strong nodal coupling coefficient suggests that that node occupies a similar location in structural space, relative to all other nodes, as it does in functional space. Further, a negative coupling value suggests that a node's position in structural manifold space, relative to all other nodes, is anticorrelated to its relative position in functional manifold space. 199 CALM participants had coupling values for one timepoint, with 84 additional coupling values for a second timepoint. 121 NKI participants had a baseline coupling value, with 126 and 99 additional points for a second and third timepoint, respectively. In line with prior studies (*Baum et al., 2020*; *Suárez et al., 2020*; *Vázquez-Rodríguez et al., 2019*), across data sets, we observed a unimodal-transmodal axis of structure-function coupling magnitude and inter-individual variability (*Figure 4b*). The largest and least variable coupling was centred around the unimodal anchor, including bilateral somato-motor and visual networks, whilst the smallest and most variable coupling was centred around the transmodal anchor, including bilateral default-mode and limbic networks. The distribution of cortical coefficients of variation (CV) varied considerably, with the largest CV (in the parietal division of the left default-mode network) being over 400 times that of the smallest (in the right visual network). The distribution of absolute CVs was positively skewed, with a Fisher skewness coefficient $g_1$ of 7.172, meaning relatively few regions had particularly high

inter-individual variability, and highly peaked, with a kurtosis of 54.883, where a normal distribution has a skewness coefficient of 0 and a kurtosis of 3. Further, stratifying structure-function coupling by network and dataset (*Figure 4c*) revealed consistent coupling across datasets. To parse effects of neurotypicality and dataset on structure-function coupling, we conducted a series of generalised linear models with mean global and network-level coupling as the outcomes, using neurotypicality, site, sex, head motion, and age at scan, at baseline (N=77 CALM neurotypical, N=173 CALM neurodivergent, and N=170 NKI). However, we found no significant effects of neurotypicality on structure-function coupling.

To evaluate our novel coupling metric, we compared its cortical spatial distribution to three others (*Baum et al., 2020*; *Park et al., 2022*; *Vázquez-Rodríguez et al., 2019*), using the group-level thresholded structural and functional connectomes from the referred CALM cohort. As shown in *Figure 4d*, our novel metric was moderately positively correlated to that of a multi-linear regression framework ($r_s$ = 0.494, $p_{spin}$ = 0.004; *Vázquez-Rodríguez et al., 2019*) and nodal correlations of fibre bundle capacity and functional connectivity ($r_s$ = 0.470, $p_{spin}$ = 0.005; *Baum et al., 2020*). As expected, our novel metric was strongly positively correlated to the manifold-derived coupling measure ($r_s$ = 0.661, $p_{spin}$ <0.001; *Park et al., 2022*), more so than the first ($Z$(198) = 3.669, p<0.001) and second measure ($Z$(198) = 4.012, p<0.001). Structure-function coupling is thought to be patterned along a sensorimotor-association axis (*Sydnor et al., 2021*): all four metrics displayed weak-to-moderate alignment (*Figure 4d*). Interestingly, the manifold-based measures appeared most strongly aligned with the sensorimotor-association axis: the novel metric was more strongly aligned than the multi-linear regression framework ($Z$(198) = −11.564, p<0.001) and the raw connectomic nodal correlation approach ($Z$(198) = −10.724, p<0.001), but the previously implemented structural manifold approach was more strongly aligned than the novel metric ($Z$(198) = −12.242, p<0.001). This suggests that our novel metric exhibits the expected spatial distribution of structure-function coupling, and the manifold approach more accurately recapitulates the sensorimotor-association axis than approaches based on raw connectomic measures.

Next, we examined the developmental trajectories of structure-function coupling within the GAMM framework, with the same model specification as before (*Figure 4e*). Whilst age was not statistically significant linked to coupling globally or across any network partition, we did observe that an interaction between age and dataset was a significant linked to coupling within dorsal attention ($F$=9.579, $p_{FDR}$ = 0.008) and default-mode ($F$=7.245, $p_{FDR}$ = 0.007) networks. Within the dorsal attention network, both datasets exhibited a linear age effect (EDF = 1), with a stronger age effect from NKI ($F$=9.948, p=0.002) than CALM ($F$=0.484, p=0.487), and spanning the entire developmental period examined (6.90–17.96 years). Similarly, within the default-mode network, the age effect in NKI ($F$=5.110, p=0.011) was stronger than CALM ($F$=2.267, p=0.198). The age effect in NKI was highly non-linear (EDF = 2.341), with a later onset period of statistically significant development (14.39–17.96 years) than CALM (11.75–14.58 years). Together, these effects demonstrate that whilst the magnitude of structure-function coupling appears not to be sensitive to neurodevelopmental phenotype, its development with age *is*, particularly in higher-order association networks, with developmental change being reduced in the neurodivergent sample.

To further examine whether a closer correspondence of structure-function coupling with age is associated with neurotypicality, we conducted a follow-up analysis using the additional age-matched neurotypical portion of CALM (N=77). Given the widespread developmental effects on coupling within the neurotypical NKI sample, compared to the absent effects in the neurodivergent portion of CALM, we would expect strong relationships between age and structure-function coupling with the neurotypical portion of CALM. This is indeed what we found: structure-function coupling showed a linear negative relationship with age globally ($F$=16.76, $p_{FDR}$ <0.001, adjusted $R^2$=26.44%), alongside fronto-parietal ($F$=9.24, $p_{FDR}$ = 0.004, adjusted $R^2$=19.24%), dorsal-attention ($F$=13.162, $p_{FDR}$ = 0.001, adjusted $R^2$=18.14%), ventral attention ($F$=11.47, $p_{FDR}$ = 0.002, adjusted $R^2$=22.78), somato-motor ($F$=17.37, $p_{FDR}$ <0.001, adjusted $R^2$=21.92%), and visual ($F$=11.79, $p_{FDR}$ = 0.002, adjusted $R^2$=20.81%) networks. Together, this supports our hypothesis that within neurotypical children and adolescents, structure-function coupling decreases with age, showing a stronger effect compared to their neurodivergent counterparts, in tandem with the emergence of higher-order cognition. Thus, whilst the magnitude of structure-function coupling across development appeared insensitive to neurotypicality, its maturation *is* sensitive. Tentatively, this suggests that neurotypicality is linked to stronger and more

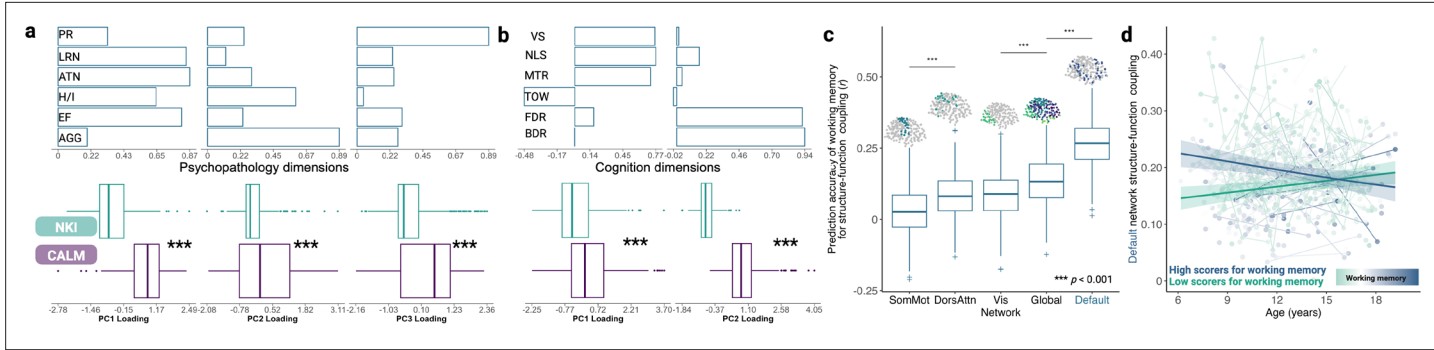

**Figure 5.** Dimensions of cognition, rather than psychopathology, are developmentally-sensitive predictors of structure-function coupling. We applied principal component analysis with varimax rotation to obtain orthogonal dimensions of psychopathology and cognition, across all which CALM loaded more strongly onto than NKI. Within each box plot, the box represents the lower quartile, median, and upper quartile, respectively. Circular points represent outliers. (**a**) The first dimension of psychopathology captures learning problems ('LRN'), inattention ('ATN') and executive functioning ('EF'). The second dimension captures aggression ('AGG') and hyperactivity/inattention ('H/I'), whilst the third dimension captures peer relation difficulties ('PR'). (**b**) Visual search ('VS'), number-letter switching ('NLS'), motor speed ('MTR'), and the Tower ('TOW') task loaded strongly onto the first cognitive dimension, reflecting executive functioning. Forward- and backward-digit spans ('FDR' and 'BDR') loaded strongly onto the second cognitive dimension, reflecting working memory. (**c**) Cross-validated predictive accuracy of working memory for structure-function coupling of five networks within which we observed a significant interaction between age and working memory, namely somato-motor ('SomMot'), dorsal attention ('DorsAttn'), visual ('Vis'), default-mode ('Default') and globally. The default-mode network is highlighted as having the strongest developmentally sensitive relationship with working memory, such that the relationship between default-mode coupling and working memory was dependent on age, with the two groups gradually closing the developmental gap. Pairwise comparisons were made between consecutively ordered networks, to reduce multiple comparisons. Predictive accuracy across all visualised networks was significantly greater than 0, as determined through a one-tailed *t*-test (all p<0.001). (**d**) Main effects of age are visualised within participants with low (n=314, visualised in green) or high (n=315, visualised in blue) scores on the second cognitive dimension, as determined by a median split, for structure-function coupling in the default-mode network. Parametric main effects of mean framewise displacement, sex, principal component 1 scores, and an interaction between age and factor 1 scores were included as covariates. 95% simultaneous confidence intervals derived from using the group-specific (high or low scores) generalised additive mixed model to predict structure-function coupling at equal intervals across the entire developmental range examined (6.17–19.17 years old).

consistent maturational development of structure-function coupling, whereby the tethering of functional connectivity to structure across development is adaptive.

## Dimensions of cognition, but not psychopathology, are developmentally sensitive predictors of structure-function coupling

Thus far, we have charted the canonical trajectories of structural and functional manifold reorganisation across childhood and adolescence and examined structure-function relationships. However, what is the behavioural significance, if any, of differences in structure-function coupling? The final step of our analysis related structure-function coupling to dimensions of psychopathology and cognition. To this end, we took advantage of the full range of behavioural and cognitive heterogeneity offered by combining NKI and CALM. To derive orthogonal dimensions of psychopathology (*Figure 5a*), we conducted a principal component analysis (PCA) with varimax rotation on 6 age-standardised *t*-score measures from the Conners III Questionnaire (*Conners et al., 2011*), measuring inattention, hyperactivity/impulsivity, learning problems, executive functioning, aggression, and peer relations. Component 1 explained 44.73% variance, and captured inattention, learning problems, and executive functioning. Component 2 explained 23.09% variance and was dominated by aggression. Component 3 explained 18.23% variance and was dominated by peer relationships. A series of two-tailed continuity-corrected Mann-Whitney U tests revealed significantly larger loadings across CALM than NKI for psychopathology component 1 (*U*=87,320, p<0.001), component 2 (*U*=58,213, p<0.001), and component 3 (*U*=63,832, p<0.001). Two dimensions of cognition were derived using the same method (*Figure 5b*), based on 6 z-score age-standardised broad measures of cognition: forward and backward digit recall from the Automated Working Memory Assessment (*Alloway, 2007*), alongside Tower achievement score, visual scanning speed, number-letter switching, and motor speed from the Delis-Kaplan Executive Function System (*Delis et al., 2001*). The first component explained 32.30% of variance in the cognitive measures. Of the four measures which most strongly loaded onto the first component, visual scanning, number-letter switching, and motor speed loaded

positively, whilst the Tower task loaded negatively. The second component explained 29.63% of variance, onto which forward and backward digit recall loaded most strongly, suggesting that this component captures working memory. Like psychopathology, the difference between median loadings for participants from CALM and NKI for cognitive components 1 ($U$=65,086, p<0.001) and 2 ($U$=96,732, p<0.001) were statistically significant. Together, this suggests that CALM and NKI differed significantly in terms of psychopathology and cognitive profiles.

We then examined the extent to which dimensions of psychopathology were linked to structure-function coupling, using the GAMM framework. Alongside sex, mean framewise displacement and age as covariates, we additionally included main effects of each psychopathology dimension and interactions with age to parse development-specific effects. Neither the main effects of psychopathology dimension or their interaction with age were significantly linked to structure-function coupling globally or across any network partition following multiple comparison correction. However, when we applied the same modelling framework to two dimensions of cognition, we found that higher loadings on the second cognitive dimension, capturing working memory, significantly and linearly predicted stronger structure-function coupling within the default-mode network (EDF = 1.000, $F$=13.714, $p_{FDR}$ = 0.006). Furthermore, we found several significant interactions between age and this second cognitive factor (working memory). This interaction was significantly linked to structure-function coupling globally ($F$=9.347, $p_{FDR}$ = 0.016), as well as within visual ($F$=7.386, $p_{FDR}$ = 0.032), somato-motor ($F$=7.841, $p_{FDR}$ = 0.027), dorsal attention ($F$=8.002, $p_{FDR}$ = 0.027) and default-mode networks ($F$=12.446, $p_{FDR}$ = 0.006).

To examine the predictive value of the second cognitive factor for global and network-level structure-function coupling, operationalised as a Spearman rank correlation coefficient, we implemented a stratified fivefold cross-validation framework and predictive accuracy compared with that of a null data frame with cognitive factor 2 scores permuted across participant blocks (see 'GAMM cross-validation' in the Methods). This procedure was repeated 100 times to account for randomness in the train-test splits, using the same model specification as above. Therefore, for each of the 5 network partitions in which an interaction between the second cognitive factor and age was a significant predictor of structure-function coupling (global, visual, somato-motor, dorsal attention and default-mode), we conducted a Welch's independent-sample $t$-test to compare 500 empirical prediction accuracies with 500 null prediction accuracies. Across all five network partitions, predictive accuracy of coupling was significantly higher than that of models trained on permuted cognitive factor 2 scores (all p<0.001). We observed the largest difference between empirical ($M$=0.029, SD = 0.076) and null ($M$=–0.052, SD = 0.087) prediction accuracy in the somato-motor network [$t$(980.791)=15.748, p<0.001, Cohen's $d$=0.996], and the smallest difference between empirical ($M$=0.080, SD = 0.082) and null ($M$=0.047, SD = 0.081) prediction accuracy in the dorsal attention network [$t$(997.720)=6.378, p<0.001, Cohen's $d$=0.403]. To compare relative prediction accuracies, we ordered networks by descending mean accuracy and conducted a series of Welch's independent sample $t$-tests, followed by FDR correction (*Figure 5c*). Prediction accuracy was highest in the default-mode network ($M$=0.265, SD = 0.085), twofold that of global coupling ($t$(992.824)=25.777, $p_{FDR}$ = 5.457 x 10$^{-112}$, Cohen's $d$=1.630, $M$=0.131, SD = 0.079). Global prediction accuracy was significantly higher than the visual network ($t$(992.644)=9.273, $p_{FDR}$ = 1.462 x 10$^{-19}$, Cohen's $d$=0.586, $M$=0.083, SD = 0.085), but visual prediction accuracy was not significantly higher than within the dorsal attention network ($t$(997.064)=0.554, $p_{FDR}$ = 0.580, Cohen's $d$=0.035, $M$=0.080, SD = 0.082). Finally, prediction accuracy within the dorsal attention network was significantly stronger than that of the somato-motor network [$t$(991.566)=10.158, $p_{FDR}$ = 7.879 x 10$^{-23}$, Cohen's $d$=0.642, $M$=0.029, SD = 0.076]. Together, this suggests that out-of-sample developmental predictive accuracy for structure-function coupling, using the second cognitive factor, is strongest in the higher-order default-mode network and lowest in the lower-order somatosensory network.

To parse these cognition-age interactions, we ran a median split by cognitive ability and repeated the GAMM procedure within each half. For global structure-function coupling, we observed a stronger age effect in participants with lower ($F$=9.449, p=0.002), rather than higher ($F$=1.908, p=0.168), cognitive dimension scores. For visual structure-function coupling, we also observed a stronger age effect in participants with lower ($F$=6.024, p=0.015) than higher ($F$=2.178, p=0.226) working memory scores. The median split yielded non-significant age effects within the somato-motor network, possibly due to a lack of power, with marginally stronger developmental effects in participants with higher working

memory scores ($F$=3.777, p=0.053). The difference in the developmental association of working memory loadings with structure-function coupling was perhaps most pronounced within the dorsal attention network, with the age effect on structure-function coupling within the low ability group was over tenfold ($F$=13.875, p=0.001) that of the high ability group ($F$=0.009, p=0.924). Finally, whilst age was significantly linked to coupling globally, as well as within the visual, dorsal attention, and somato-motor network within participants with either high or low scores on the second cognitive dimension, developmental effects were statistically significant for both low ($F$=8.779, p=0.003) and high ($F$=7.966, p=0.005) loading groups when predicting default-mode network coupling (**Figure 5d**). In summary, cognitive dimensions, specifically working memory, significantly predicted stronger structure-function coupling across several network partitions in a developmentally specific manner, with larger between-dataset differences *earlier* in development. Those with the highest cognitive scores exhibited a gradual reduction in coupling, whilst those with the lowest cognitive scores exhibited a gradual increase in coupling. Further, those with higher working memory scores tend to show the strongest developmental changes in coupling, with higher coupling earlier in development, but weaker towards the end of adolescence, and more marked in the dorsal attention network.

## Discussion

Using diffusion-map embedding we established highly stable principles of structural and functional brain organisation, with the ordering of key gradients being entirely insensitive to age. Furthermore, group-level structural and functional gradients were highly consistent across datasets, whilst differences between datasets were emphasised when taking development into account, through differing rates of structural manifold contraction and functional manifold expansion, respectively, alongside maturation of structure-function coupling. Similarities in group-level gradients and their developmental trajectories suggest that such gradients are established early in life, refined through development, and may be examples of universal organisational principles of cortical development. By capitalising on manifold eccentricity as a composite measure of segregation across development, we build upon an emerging literature pioneering gradients as a method to establish underlying principles of structural (**Paquola et al., 2020**; **Park et al., 2021**) and functional (**Dong et al., 2021**; **Margulies et al., 2016**; **Xia et al., 2022**) brain development without a priori specification of specific graph theory metrics of interest. Finally, we quantified the intersection between these two forms of low-dimensional embedding, generating a metric of the coupling between structural and functional manifolds. This coupling displayed a characteristic unimodal-transmodal organisational axis and was highly consistent across datasets. Interactions between age and the second cognitive dimension, capturing working memory, significantly predicted structure-function coupling globally, alongside within somato-motor, dorsal attention, and default-mode networks, with the largest and most consistent age effects concentrated within higher-order late-developing association cortices.

First, we demonstrated the stable ordering of structural and functional gradients across childhood and adolescence, and across samples making up a neurodivergent-neurotypical continuum. That this consistent pattern emerges despite considerable differences in phenotype suggests that there is a degree of species universality in these organisational principles. The stable functional gradient ordering is in contrast to a small body of developmental work suggesting that the primary functional gradient in childhood differentiates between the somato-motor and auditory cortices from the visual cortex, and is replaced by a gradient anchored in unimodal and heteromodal association cortices during late childhood (**Xia et al., 2022**) and early adolescence (**Dong et al., 2021**). Despite the growing evidence in support of different primary gradients for childhood and adolescence, inconsistencies exist in the literature. For example, **Dong et al., 2021** reported that the principal functional gradient in childhood accounted for 38% variance, with the second accounting for 11%. Such a stark difference between the principal and secondary functional gradient is in contrast to both the current study and prior work (**Xia et al., 2022**). We consider the role of thresholding, cortical resolution, and head motion as avenues to reconcile the present results with select reports in the literature (**Dong et al., 2021**; **Xia et al., 2022**). That is, whilst we observed developmental *refinement* of gradients, in terms of manifold eccentricity, standard deviation and variance explained, we did not observe *replacement*. Note, as opposed to calculating gradients based on group data, such as a sliding window approach, which may artificially smooth developmental trends and summarise them to the mean, we used participant-level data throughout. We would suggest that thresholding has a greater

effect on vertex-level data, rather than parcel-level. For example, a recent study revealed that the emergence of principal vertex-level functional connectivity gradients in childhood and adolescence is indeed threshold-dependent (**Dong et al., 2024**). Specifically, the characteristic unimodal organisation for children and transmodal organisation for adolescents only emerged at the 90% threshold: a 95% threshold produced a unimodal organisation in both groups, whilst an 85% threshold produced a transmodal organisation in both groups. Put simply, the 'swapping' of gradient orders only occurs at certain thresholds.

Furthermore, our results are not necessarily contradictory to this prior report (**Dong et al., 2021**): developmental changes in high-resolution gradients may be supported by a stable low-dimensional coarse manifold. Indeed, our decision to use parcellated connectomes was partly driven by recent work which demonstrated that vertex-level functional gradients may be derived using biologically plausible but random data with sufficient spatial smoothing, whilst this effect is minimal at coarser resolutions (**Watson and Andrews, 2023**). We observed a gradual increase in the variance of individual connectomes accounted for by the principal functional connectivity gradient in the referred subset of CALM, in line with prior vertex-level work demonstrating a gradual emergence of the sensorimotor-association axis as the principal axis of connectivity (**Xia et al., 2022**), as opposed to a sudden shift. It is also possible that vertex-level data is more prone to motion artefacts in the context of developmental work. Transitioning from vertex-level to parcel-level data involves smoothing over short-range connectivity, thus greater variability in short-range connectivity can be observed in vertex-level data. However, motion artefacts are known to increase short-range connectivity and decrease long-range connectivity, mimicking developmental changes (**Satterthwaite et al., 2013**). Thus, whilst vertex-level data offers greater spatial resolution in representation of short-range connectivity relative to parcel-level data, it is possible that this may come at the cost of making our estimates of the gradients more prone to motion. Given the growing application of gradient-based analyses in modelling structural (**He et al., 2025**; **Li et al., 2024**) and functional (**Dong et al., 2021**; **Xia et al., 2022**) brain development, we hope to provide a blueprint of factors which may affect developmental conclusions drawn from gradient-based frameworks.

The gradual functional manifold expansion we observed, indicating functional segregation, builds upon previous reports of cortex-wide functional segregation across youth (**Tooley et al., 2022**), but at the level of overarching organisational gradients, rather than individual graph theory measures, providing a more unifying account of cortical reorganisation. In parallel, the gradual structural manifold contraction, indicating structural integration, supports existing generative models of structural brain development predicting network integration across youth (**Akarca et al., 2023**). However, the developmental changes in structural manifold eccentricity are not uniformly distributed across cortex or participants; the developmental effects on structural manifold contraction were concentrated in higher-order association regions, with significantly higher manifold eccentricity globally in neurodivergent participants. We demonstrated increased consistency of developmental effects in structural manifold integration and greater sensitivity to neurodiversity, relative to functional manifolds. This may reflect greater variability in functional, rather than structural, brain development, and therefore reorganisation of the low-dimensional *structural* manifold space may be a more consistent and reliable marker of development and phenotype.

We consider two possible factors to explain the greater sensitivity of neurodivergence to gradients of communicability, rather than functional connectivity. First, functional connectivity is likely more sensitive to head motion than structural-based communicability and suffers from reduced statistical power due to stricter head motion thresholds, alongside greater inter-individual variability. Second, whilst prior work contrasting functional connectivity gradients from neurotypical adults with those with confirmed ASD diagnoses demonstrated vertex-level reductions in the default-mode network in ASD and marginal increases in sensory-motor communities (**Hong et al., 2019**), indicating a sensitivity of functional connectivity to neurodivergence, important differences remain. Specifically, whilst the vertex-level group-level differences were modest, in line with our work, greater differences emerged when considering step-wise functional connectivity (SFC); in other words, when considering the *dynamic transitions of* or *information flow through* the functional hierarchy underlying the static functional connectomes, such that ASD was characterised by initial faster SFC within the unimodal cortices followed by a lack of convergence within the default-mode network (**Hong et al., 2019**). This emphasis on information flow and dynamic underlying states may point towards greater sensitivity of

neurodivergence to structural communicability – a measure directly capturing information flow – than static functional connectivity.

In line with prior work (*Baum et al., 2020*; *Gu et al., 2021*; *Park et al., 2022*; *Preti and Van De Ville, 2019*; *Suárez et al., 2020*), an axis of structure-function coupling emerged across datasets, strongest in unimodal lower-order regions and weakest in heteromodal higher-order regions. This sensorimotor-associative axis of structure-function coupling follows patterns of evolutionary expansion in the cortex, such that regions with the smallest evolutionary expansion and highest cross-species similarity have functionally-specific roles, anchored in unimodal regions, whilst regions with the largest evolutionary expansion and smallest cross-species similarity have functionally-diverse roles, anchored in association cortices (*Sydnor et al., 2021*; *Xu et al., 2020*). Cross-species tritiated thymidine experiments *Finlay and Darlington, 1995* have demonstrated that brain regions with disproportionately large volumes, such as association cortices, experience protracted neurogenesis. Speculatively, this may be associated with greater sensitivity to age, which we demonstrated, relative to lower-order networks, as significant interactions between age and data set for structure-function coupling in higher-order networks, particularly those incorporating the frontal lobes. Further, whilst the neurodivergent dataset displayed stronger developmental effects in structural manifold remodelling, we observed stronger developmental effects in structure-function coupling within the neurotypical sample, concentrated in higher-order networks. Speculatively, this may point towards protracted structural manifold development underpinning more flexible structure-function coupling, which is more sensitive to age and associated with neurotypicality.

Whilst the spatial patterning of structure-function coupling across the cortex has been extensively documented, as explained above, less is known about developmental trajectories of structure-function coupling, or how such trajectories may be altered in those with neurodevelopmental conditions. To our knowledge, only one prior study has examined differences in developmental trajectories of (non-manifold) structure-function coupling in typically developing children and those with attention-deficit hyperactivity disorder (*Soman et al., 2023*), one of the most common conditions in the neurodivergent portion of CALM. Namely, using cross-sectional and longitudinal data from children aged between 9 and 14 years old, they demonstrated increased coupling across development in higher-order regions overlapping with the default-mode, salience, and dorsal attention networks, in children with ADHD, with no significant developmental change in controls, thus encompassing an ectopic developmental trajectory (*Di Martino et al., 2014*; *Soman et al., 2023*). Whilst the current work does not focus on any condition, rather the broad mixed population of young people with neurodevelopmental symptoms (including those with and without diagnoses), there are meaningful individual and developmental differences in structure-coupling. Crucially, it is not the case that simply having stronger coupling is desirable. The current work reveals that there are important developmental trajectories in structure-function coupling, suggesting that it undergoes considerable refinement with age. Note that whilst the magnitude of structure-function coupling across development did not differ significantly as a function of neurodivergence, *its relationship to age did*. Our working hypothesis is that structural connections allow for the ordered integration of functional areas, and the gradual functional modularisation of the developing brain. For instance, those with higher cognitive ability show a stronger refinement of structure-function coupling across development. Future work in this space needs to better understand not just how structural *or* functional organisation change with time, but rather how one supports the other.

Additionally, dimensions of cognitive ability, specifically working memory, were robust predictors of stronger structure-function coupling within higher-order association regions. This supports the tethering hypothesis of brain development (*Buckner and Krienen, 2013*), which posits that association cortices are *untethered* by, and therefore less susceptible to, the intrinsic morphogenic signalling patterns and extrinsic activity from sensory systems that otherwise constrain development of evolutionarily ancient regions, such as sensory cortices. This also supports existing structure-function coupling studies which demonstrated that variability in structure-function coupling derived from an N-back working memory task follows a unimodal-transmodal axis, and exhibits developmental effects principally concentrated within higher-order association areas (*Baum et al., 2020*). The present work expands these findings by testing associations between structure-function coupling and broad dimensions of psychopathology and cognition in neurotypical and neurodivergent participants, across datasets. Across development, we also observed that stronger structure-function coupling within the

default-mode network was associated with significantly higher loadings for the second cognitive component, but that this effect was development-dependent. This might suggest that better performance on working memory tasks is supported by stronger alignment between low-dimensional structural and functional axes: this effect is strongest early in development for high-achievers and strongest in later adolescent development for low-achievers.

An emerging theme from our work is the importance of the default-mode network as a region in which structure-function coupling is reliably predicted by working memory across neurodevelopmental phenotypes and datasets during childhood and adolescence. Recent neurotypical adult investigations combining high-resolution post-mortem histology, in vivo neuroimaging, and graph-theory analyses have revealed how the underlying neuroanatomy of the default-mode network may support diverse functions (*Paquola et al., 2025*), and thus exhibit lower structure-function coupling compared to unimodal regions. The default-mode network has distinct neuroanatomy compared to the remaining 6 intrinsic resting-state functional networks (*Yeo et al., 2011*), containing a distinctive combination of 5 of the 6 von Economo and Koskinas cell types (*Economo and Koskinas, 1925*), with an over-representation of heteromodal cortex and uniquely balancing output across all cortical types. A primary cytoarchitectural axis emerges, beyond which are mosaic-like spatial topographies. The duality of the default-mode network, in terms of its ability to both integrate and be insulated from sensory information, is facilitated by two microarchitecturally distinct subunits anchored at either end of the cytoarchitectural axis (*Paquola et al., 2025*). Whilst beyond the scope of the current work, structure-function coupling and their predictive value for cognition may also differ across divisions *within* the default-mode network, particularly given variability in the smoothness and compressibility of cytoarchitectural landscapes across subregions (*Paquola et al., 2025*).

In an effort to establish more reliable brain-behaviour relationships despite not having the statistical power afforded by large-scale, typically normative, consortia (*Rosenberg and Finn, 2022*), we demonstrated the development-dependent link between default-mode structure-function coupling and working memory generalised across clinical (CALM) and normative (NKI) samples, across varying MRI acquisition parameters and harnessing within- and across-participant variation. Such multivariate associations are likely more reliable than their univariate counterparts (*Marek et al., 2022*), but can be further optimised using task-related fMRI (*Rosenberg and Finn, 2022*). The consistency, or lack of, of developmental effects across datasets emphasises the importance of validating brain-behaviour relationships in highly diverse samples. Particularly evident in the case of structure-function coupling development, through our use of contrasting samples, is *equifinality* (*Cicchetti and Rogosch, 1996*), a key concept in developmental neuroscience: namely, similar 'endpoints' of structure-function coupling may be achieved through different initialisations dependent on working memory.

Nonetheless, our results have several methodological limitations. First, longitudinal data in the current study forms a small proportion of the overall sample, resulting in statistical power insufficient to detect smaller effects. This may be particularly evident in the case of functional manifold reorganisation, which has greater variability than structural manifold reorganisation. Secondly, a potential methodological limitation in the construction of structural connectomes was the 30 mm tract length threshold which, despite being the QSIprep reconstruction default (*Cieslak et al., 2021*), may have potentially excluded short-range association fibres. This is pertinent as tracts of different lengths exhibit unique distributions across the cortex and functional roles (*Bajada et al., 2019*): short-range connections occur throughout the cortex but peak within primary areas, including the primary visual, somato-motor, auditory and para-hippocampal cortices, and are thought to dominate lower-order sensorimotor functional resting-state networks, whilst long-range connections are most abundant in tertiary association areas and are recruited alongside tracts of varying lengths within higher-order functional resting-state networks. Therefore, inclusion of short-range association fibres may have resulted in a relative increase in representation of lower-order primary areas and functional networks. On the other hand, we also note the potential misinterpretation of short-range fibres: they may be unreliably distinguished from null models in which tractography is restricted by cortical gyri only (*Bajada et al., 2019*). Further, prior (neonatal) work has demonstrated that the order of connectivity of regions and topological fingerprints are consistent across varying streamline thresholds (*Mousley et al., 2025*), suggesting minimal impact. Thirdly, given the spatial smoothing applied to the functional connectivity data, and examining its correspondence to streamline-count connectomes through structure-function coupling, applying the equivalent smoothing to structural connectomes may improve the

reliability of inference and subsequent sensitivity to cognition and psychopathology. Connectome spatial smoothing involves applying a smoothing kernel to the two streamline endpoints, whereby variations in smoothing kernels are selected to optimise the trade-off between subject-level reliability and identifiability, thus increasing the signal-to-noise ratio and the reliability of statistical inferences of brain-behaviour relationships (*Mansour L et al., 2022*). However, we note that such smoothing is more effective for high-resolution connectomes, rather than parcel-level, and so have only made a modest improvement (*Mansour L et al., 2022*). Fourthly, a potential limitation of our study was the exclusion of subcortical regions. However, prior work has shed light on the role of subcortical connectivity in structural and functional gradients, respectively, of neurotypical populations of children and adolescents (*Park et al., 2021*; *Xia et al., 2022*). For example, in the context of the primary-to-transmodal and sensorimotor-to-visual functional connectivity gradients, the mean gradient scores within subcortical networks were demonstrated to be relatively stable across childhood and adolescence (*Xia et al., 2022*). In the context of structural connectivity gradients derived from SIFT2-weighted FBC, which we demonstrated were highly consistent with those derived from communicability, subcortical structural manifolds weighted by their cortical connectivity were anchored by the caudate and thalamus at one pole, and by the hippocampus and nucleus accumbens at the opposite pole, with significant age-related manifold expansion within the caudate and thalamus (*Park et al., 2021*). A final limitation relates to identification of sensitive periods in gradient development: these are characterised by *Fuhrmann et al., 2015* as periods of heightened environmental sensitivity and neuroplasticity, resulting in increased developmental change but also vulnerability, typically occurring in adolescence. This is driven by experience-dependent plasticity over and beyond baseline experience-independent plasticity, with individual differences in its onset and offset. Whilst we identified periods of significant manifold developmental change spanning childhood and adolescence, these are not necessarily *sensitive*. To test this empirically, future work could evaluate the effects of modifiable environmental factors, such as early-life adversity, on manifold change above and beyond age-related effects, ideally within a longitudinal design. Specifically, we would assess whether such effects are heightened within adolescence, significantly greater than that of childhood and early adulthood, indicating a possible sensitive period.

In summary, by triangulating clinical and community developmental samples, we robustly charted the development of structural and functional manifold reorganisation across childhood and adolescence, utilising 887 structural and 728 functional scans, respectively, by modelling the linear and non-linear functional forms of age. We suggest that childhood and adolescence are characterised by consistently ordered structural and functional gradients established early in life, refined through development with the greatest change in higher-order association networks, and whose coupling reflects dimensions of cognition, specifically working memory. Our findings also highlight the importance of studying *development* in examining individual differences in brain organisation across time, particularly during youth, and highlight equifinality. That is, whilst group-level trends were very similar across phenotypes, the trajectories to reach such organisation varied.

## Methods
### Participants

To evaluate how gradients varied across time and neurodevelopmental profiles, we used two mixed-design samples with overlapping age ranges. The first recruited neurodivergent children, and the second recruited neurotypical children (6.4–19 years old). The first data set was the Centre for Attention, Learning, and Memory (CALM). Native English-speaking children, referred by educational and clinical professionals, with difficulties in attention, language, and/or memory were included regardless of diagnosis, medication, and health conditions, except with uncorrected sensory impairments or health conditions affecting cognition (see *Holmes et al., 2019* for a detailed protocol). Our final sample consisted of participants referred to the service due to difficulties with attention, learning, or memory, and with T1w and rsfMRI or DTI data across at least one time point, excluding repeated scans. Thus, we processed 440 DWI scans from CALM for 352 participants (70% male), where 60% had one scan ($Mean_{Age}$ = 10.72 ± 1.74 years), and the remainder had two ($Mean_{Age}$ = 13.76 ± 2.26 years). Further, we processed 304 rsfMRI scans from 256 participants (66.45% male), where 68.42% had one scan ($Mean_{Age}$ = 10.72 ± 1.81 years) and the remainder had two ($Mean_{Age}$ = 13.84 ± 2.24 years).

The second sample was the Longitudinal Discovery of Brain Development Trajectories sub-study of the NKI-Rockland Sample (see *Tobe et al., 2022* for detailed protocols). Rather than recruiting neuro-divergent children, this study contained a community-ascertained neurotypical sample of 339 children (55.16% male) aged between 6 and 17 years old at recruitment from Rockland, Orange, Bergen, and Westchester Counties in the United States. Baseline recruitment started in December 2013, with follow-ups at 12- or 15-month intervals. 136 children had a single session, 76 had one follow-up, and 127 had two follow-ups. In contrast to referral by health and education professionals in CALM, NKI children were referred to the service through word of mouth, flyers, prior engagement with NKI events and education days. Exclusion criteria included a history of autism spectrum disorder, psychiatric hospitalisations, an intelligence quotient below 70, and prior treatment with antidepressants, neuroleptics, and mood stabilisers (see *Tobe et al., 2022* for further details). Across two 6 hr days, children completed a broad cognitive battery, provided biological samples for genetic analysis, and multi-modal neuroimaging. After September 2015, this protocol was amended to one 8 hr day. The study was approved by the NKI for Psychiatric Research Institutional Review Board. Written consent was obtained by the caregiver, and then the participant if they returned for follow-up after their 18th birthday. We processed 448 DWI scans from 259 NKI participants (56.92% male), where 28.35% had one scan (Mean$_{Age}$ = 11.94 ± 3.04 years), 33.48% had two (Mean$_{Age}$ = 12.20 ± 2.69 years), and the remainder had three (Mean$_{Age}$ = 12.46 ± 2.52 years). Further, we processed 425 rsfMRI scans from 257 NKI participants (54.35% male), where 31.29% had one scan (Mean$_{Age}$ = 12.28 ± 2.97 years), 37.65% had two (Mean$_{Age}$ = 12.34 ± 2.82 years), and 31.06% had three (Mean$_{Age}$ = 12.94 ± 2.76 years).

## Pre-processing of behavioural and cognitive data

We pre-processed behavioural and cognitive data in R v4.2.2 using RStudio. Using predictive mean matching in the mice v3.16 package (*Buuren and Groothuis-Oudshoorn, 2011*), for baseline CALM with neuroimaging data (N=443), we imputed missing gender (N=2) and age-standardised scores for object counting (N=7) from the Phonological Assessment Battery (PhAB; *Frederickson et al., 1997*), alongside digit recall (N=4) and backward digit span (N=5) from the Automated Working Memory Assessment (AWMA; *Alloway, 2007*). For longitudinal CALM with neuroimaging data (N=151), we imputed age standardised PhAB alliteration (N=2) and object counting (N=2) scores, alongside AWMA digit recall (N=3) scores.

## MRI data acquisition

### CALM

Diffusion tensor imaging (DTI) and rsfMRI data were used in the present study, with T1-weighted (T1w) and T2-weighted (T2w) scans for registration. All modalities were collected using a 3T Siemens Prisma scanner with 32-channel head coil, within a 1 hr session at the MRC Cognition and Brain Sciences Unit, Cambridge, United Kingdom (see *Holmes et al., 2019* for additional details). Baseline scans had all four scan types, whilst T2w scans were not collected in the longitudinal sample. Further, scanner software was upgraded from VD13D to VE11E in March 2021, after longitudinal data collection commenced. Children were first familiarised with a mock scanner and played an interactive game designed to minimise head motion. Across 4 min and 32 s, 192 slices (1 mm isotropic resolution with 256 mm field of view [FOV]) of T1w structural data were collected using a 3D magnetisation prepared rapid-echo sequence (MP-RAGE), with repetition time (TR) of 2500 ms, echo time (TE) of 3.02 s, inversion time (TI) of 900ms, 9° flip angle, $\frac{7}{8}$ partial Fourier, and generalised auto-calibrating partial parallel acquisition (GRAPPA) acceleration factor 4. At baseline, T2w data was collected using a 3D sampling perfection with application optimised contrast using different flip angle evolution turbo spin-echo sequence. 29 slices (.6875 mm x.6875 mm x 5.2 mm voxel resolution) were collected for 1 min and 38 s, with TR of 5060 ms, TE of 102.9 ms, and GRAPPA acceleration factor 4. DTI data of resolution 2 mm isotropic voxels and 192 mm FOV was acquired using 64 gradient diffusion directions of 1000 s/mm$^2$ and 4 interleaved reference b-value=0 s/mm$^2$ images, with TR of 8500 ms, TE of 90 ms, GRAPPA acceleration factor 2, 192 mm FOV read, and $\frac{7}{8}$ partial Fourier. Whilst participants lay with their eyes closed, 270 volumes with 32 interleaved axial slices of T2* gradient-echo echo planar imaging (EPI) was collected across 9 min and 6 swith TR of 2000 ms, TE of 30 ms, 78° flip angle, 3 mm isotropic resolution, and 192 mm FOV.

## NKI-Rockland sample longitudinal discovery of brain development trajectories

All modalities were collected using a Siemens Magnetom Trio-Tim (syngo MRB17) 3T scanner with 32-channel head coil (see *Tobe et al., 2022*) at the Nathan Kline Institute for Psychiatric Research, New York, United States. The neuroimaging protocol consisted of a T1w MP-RAGE, DTI, and 6 fMRI sequences of varying TRs, three of which were resting state. Across 4 min and 18 s, 176 slices of T1w structural data were collected using a single-shot 3D MP-RAGE sequence and GRAPPA acceleration factor of 2 with TR of 1900 ms, TE of 2.52 ms, TI of 900 ms, 9° flip angle, and 250 mm FOV. Whilst participants listened to music, 64 slices of interleaved DTI data of 2 mm isotropic resolution were collected across 5 min and 43 s using 128 b=1500 s/mm$^2$ diffusion directions and 9 b=0 s/mm$^2$ images, with TR of 2400 ms, TE of 85 ms, 90° flip angle, multi-band (MB) acceleration factor of 4, $\frac{6}{8}$ partial Fourier, and 212 mm FOV. Whilst participants lay still in silence, across 9 min and 45 s, 64 slices of an interleaved multi-slice EPI sequence with 2 mm isotropic resolution were acquired with TE of 30 ms, 65° flip angle, $\frac{6}{8}$ partial Fourier, and MB acceleration factor 4. To maximise spatial, rather than temporal, resolution, we selected the 1400 ms TR sequence for our analyses, instead of the 2500 ms TR sequence.

## MRI data processing

Across modalities and data sets, processing pipelines were identical. The following sections contain information from QSIprep (*Cieslak et al., 2021*) and fMRIprep (*Esteban et al., 2019*) boilerplates, respectively.

### Structural MRI

We processed structural and diffusion-weighted data using QSIprep 0.14.2 (*Cieslak et al., 2021*), implemented through Nipype 1.6.1 (*Gorgolewski et al., 2011*), Nilearn 0.8.0 (*Abraham et al., 2014*), and Dipy 1.4.1 (*Garyfallidis et al., 2014*). In brief, using Advanced Normalization Tools 2.3.1 (ANTs; *Avants et al., 2008*), the T1w image was corrected for intensity non-uniformity using the N4 algorithm, skull-stripped with a target from the Open Access Series of Imaging Studies, and non-linearly registered to the ICBM 152 Nonlinear Asymmetrical 2009c template (*Fonov et al., 2011*). Finally, the image was segmented into white matter (WM), gray matter (GM), and cerebrospinal fluid (CSF) using FMRIB's Automated Segmentation Tool (FAST; *Zhang et al., 2001*).

### DWI

Using MRtrix3 (*Tournier et al., 2019*), DWI scans were denoised using Marchenko-Pastur Principal Component Analysis, corrected for field inhomogeneity using the N4 algorithm (*Tournier et al., 2019*), and adjusted to the mean intensity of b=0 s/mm$^2$ images. Corrections for head motion and Eddy currents were implemented using FSL 6.0.3 (*Andersson and Sotiropoulos, 2016*), and motion-associated outliers replaced (*Andersson et al., 2016*). A rigid ANTs registration created transformations from the b=0 s/mm$^2$ to the T1w image. In preparation for constructing affinity matrices, which requires fully connected networks, we conducted whole-brain probabilistic tractography with 10 million streamlines ranging between 30 and 250 mm in length, and fibre orientation density (FOD) power of 0.33. Whilst probabilistic tractograms are considerably denser than deterministic, they suffer from high false-positive rates and inaccurate reconstruction for crossing and fanning fibres (*Maier-Hein et al., 2017*). Therefore, we implemented anatomically constrained tractography (*Smith et al., 2012*), which uses biologically informed priors to terminate streamlines, such as when entering cortical or subcortical GM or CSF. To further improve the estimate accuracy of the contribution of WM, GM, and CSF FODs to the DWI signal, single-shell multi-tissue constrained spherical deconvolution was implemented in the MRtrix3Tissue (https://3Tissue.github.io) fork of MRtrix3 (*Tournier et al., 2019*), and tissue components normalised against inhomogeneity using a log-domain version of an algorithm developed by *Raffelt et al., 2017*. Finally, spherical-deconvolution informed filtering of tractograms (SIFT-2; *Smith et al., 2015*) ensured that the density of the reconstructed streamlines more accurately mirrored that of the underlying fibre density.

We constructed structural connectomes using the SIFT2-weighted fibre bundle capacities (FBC), parcellated into 200 cortical regions, each assigned to one of seven intrinsic functional connectivity

networks (*Schaefer et al., 2018*). We used this parcellation because it matches a previous similar study assessing manifold expansion in adolescents (*Park et al., 2021*), and provides sufficient spatial resolution to clearly visualise the separation of the extremes in each DME-derived axis. For each participant, we retained the strongest 10% of connections per row, thus creating fully connected networks required for building affinity matrices. We excluded any connectomes in which such thresholding was not possible due to insufficient non-zero row values. To further ensure accuracy in connectome reconstruction, we excluded any participants whose connectomes failed thresholding in two alternative parcellations: the 100-node Schaefer 7-network (*Schaefer et al., 2018*) and Brainnetome 246-node (*Fan et al., 2016*) parcellations, respectively. To generate a group-representative connectome, we concatenated connectomes across time points, and conducted distance-dependent consensus-thresholding designed to retain the distribution of short-range connections (*Betzel et al., 2019*). This produced a binary mask, from which we extracted mean FBC. Streamlines in the brain provide tracts through which information is transferred between regions through synaptic transmission. One measure of information transfer is *communicability*, where the communicability of node $i$ is the weighted sum of all possible paths between a pair of nodes passing through it, with shortest paths weighted most strongly (*Estrada and Hatano, 2008*). Therefore, to better reflect the utility of these tracts in terms of network functioning, and to increase the predictive validity of SC for FC when examining structure-function relationships (*Seguin et al., 2020*), we converted the thresholded connectomes into weighted communicability matrices (*Crofts and Higham, 2009*).

## Resting-state functional image processing

We processed resting-state functional MRI data using fMRIprep 21.0.1 (*Esteban et al., 2019*) through Nilearn 0.8.1 (*Abraham et al., 2014*). In brief, each BOLD run was slice-time corrected using the 3dtshift tool from the Analysis of Functional Images software, head motion parameters estimated using FSL's mcflirt, resampled to original space, and non-linearly co-registered to the anatomical reference using boundary-based registration with 6 degrees of freedom. The resultant signal was smoothed with an isotropic Gaussian kernel of 6 mm full-width half-maximum, and then non-aggressively denoised using independent component analysis-based Automatic Removal of Motion Artifacts (ICA-AROMA; *Pruim et al., 2015*), during which non-steady states were automatically removed. Total WM, GM, and CSF signals were extracted. Much debate exists around fMRI denoising strategies. However, we selected ICA-AROMA due to prior evidence (*Parkes et al., 2018*) that it performs well against other denoising strategies, such as regressing head motion parameters, anatomical or temporal derivatives of noise components, in terms of minimising the dependence between distance and FC, the difference in mean signal between high- and low-motion participants, temporal degrees of freedom, and test-retest reliability. Note that whilst these researchers demonstrated that censoring marginally outperformed ICA-AROMA, this decreased temporal degrees of freedom and required at least 4 min of uncensored data for each participant. Based on this evidence, we simultaneously regressed out global signal, WM and CSF signals from ICA-AROMA denoised data using Nilearn 0.6.0 (*Abraham et al., 2014*).

To construct functional connectomes, we averaged across the BOLD time-series, conducted pair-wise Pearson correlations between each pair of ROIs from the Schaefer 200-node 7-network parcellation, and $z$-transformed the edge weights. Whilst prior DME work retained the top 10% of connections per row (*Margulies et al., 2016*), this was in vertex-wise data, rather than parcellated. Therefore, to ensure that retained connections were all positive and non-zero, as required for an affinity matrix for diffusion-map embedding, we retained the 10% absolute strongest connections per row. The individual thresholded connectomes were averaged to produce a group-representative connectome.

## DWI sample construction

Across all modalities, we included participants with low in-scanner motion (described below) and whose connectomes were successfully thresholded across three parcellations: the Brainnetome 246-node parcellation (*Fan et al., 2016*), alongside 100-node and 200-node Schaefer 7-network parcellations (*Schaefer et al., 2018*).

## CALM

At baseline, 411 participants had DWI scans and anatomical data (T1w/T2w) for registration, from which we reconstructed structural connectomes for 408. One participant was removed due to high in-scanner motion (mean FD >3 mm; *Power et al., 2012*), producing a final sample of 407 participants (Mean$_{FD}$ = 0.47 mm, SD$_{FD}$ = 0.40 mm). At follow-up, 131 participants had DWI and T1w scans, from which we reconstructed structural connectomes for 129, all with low in-scanner motion (Mean$_{FD}$ = 0.33 mm, SD$_{FD}$ = 0.20 mm).

## NKI

Out of 268 participants with a single scanning session, we reconstructed structural connectomes for 239 and retained 223 across all parcellations (Mean$_{FD}$ = 0.74, SD$_{FD}$ = 0.34). For those with two scanning sessions, we reconstructed structural connectomes for 141 of 155, all with low in-scanner motion (Mean$_{FD}$ = 0.84 mm, SD$_{FD}$ = 0.40 mm). For those with three scanning sessions, we reconstructed connectomes for 90 of 104, 2 of which were excluded due to motion, resulting in 84 participants (Mean$_{FD}$ = 0.98 mm, SD$_{FD}$ = 0.42 mm).

## rsfMRI sample construction

Across both datasets, we retained functional connectomes for participants who exhibited low in-scanner motion (mean FD < 0.5 mm), and less than 20% of spikes exceeding 0.5 mm FD.

## CALM

Out of 373 baseline participants with rsfMRI BOLD scans and anatomical data, we reconstructed functional connectomes for 372 and retained 259 after motion quality-control (Mean$_{FD}$ = 0.20, SD$_{FD}$ = 0.09). At follow-up, from 148 participants with rsfMRI BOLD and anatomical scans, we reconstructed 148 functional connectomes and retained 127 after motion quality-control (Mean$_{FD}$ = 0.19, SD$_{FD}$ = 0.08).

## NKI

Out of 301 participants with a single scanning session of rsfMRI and T1w data, we reconstructed functional connectomes for 292 and retained 214 following motion quality-control (Mean$_{FD}$ = 0.24 mm, SD$_{FD}$ = 0.08 mm). From 176 participants with two scanning sessions, we reconstructed functional connectomes for 169 and retained 133 (Mean$_{FD}$ = 0.23 mm, SD$_{FD}$ = 0.08 mm). From 112 participants with three scanning sessions, we reconstructed functional connectomes for 107 and retained 78 (Mean$_{FD}$ = 0.25 mm, SD$_{FD}$ = 0.09 mm).

## Harmonising CALM neuroimaging data

To correct for the scanner software update in March 2021 following baseline collection, which affected 22.83% of functional and 25.58% of DWI scans included in this study, respectively, we harmonised structural and functional connectomes at the edge-level using ComBat with parametric adjustments (*Fortin et al., 2017*; *Fortin et al., 2018*; *Johnson et al., 2007*). For each node, ComBat estimates the statistical variance of additive and multiplicative site effects using empirical Bayes. This, alongside a covariance matrix, is regressed out from each node. Our covariates were age at scan, sex, and age-standardised scores for matrix reasoning from the Weschler Abbreviated Scale of Intelligence II (*Wechsler, 2011*), PhAB (*Frederickson et al., 1997*) alliteration and object naming, alongside AWMA (*Alloway, 2007*) digit recall, backward digit recall, and dot matrix tests. Harmonisation aims to retain key characteristics of interest whilst removing scanner or site effects. However, the site effects in the current study are confounded with neurodivergence, and it is unlikely that neurodivergence may be captured *fully* using common covariates across CALM and NKI. Therefore, to preserve variation in neurodivergence, whilst reducing scanner effects, we harmonised within the CALM dataset only.

To assess the impact of the scanner software update, we conducted a series of generalised linear models predicting single global graph theory metrics, computed using the Brain Connectivity Toolbox (*Rubinov and Sporns, 2010*) in MATLAB R2022a (*The MathWorks, Inc, 2022*). We included age at scan, sex, head motion, timepoint, and referral status as covariates. Before harmonisation, scanner type was not a statistically significant predictor for structural efficiency ($\beta$<0.001, p=0.921), assortativity

($\beta$<0.001, p=0.911), density ($\beta$<0.001, p=0.584), transitivity ($\beta$=–0.003, p=0.566), and modularity ($\beta$=0.001, p=0.855). For functional connectomes, before harmonisation scanner type was not a statistically significant predictor for efficiency ($\beta$=0.003, p=0.849), assortativity ($\beta$<0.001, p=0.955), density ($\beta$<–0.001, p=0.352), transitivity ($\beta$=0.004, p=0.740), or modularity ($\beta$=–0.002, p=0.869). This suggests that the scanner software had minimal effects on connectome properties, even before harmonisation.

## Statistical analyses and data processing

### Manifold generation

Using the Brain Space package (*Vos de Wael et al., 2020*) in Python 3.7.3, we derived group-level and individual-level gradients for structural and functional connectivity, respectively. For each modality and hemisphere, we first constructed an affinity matrix from the group-level connectome, using a normalised angle kernel without additional sparsity, producing a *nroi/2 x nroi/2* matrix. This affinity matrix represents statistical similarities between each pair of nodes, normalised between 0.5 (least similarity) and 1 (greatest similarity), and represents the data's local geometry. We created affinity matrices in each hemisphere to avoid capturing a left-right hemisphere split. We then used diffusion-map embedding as a non-linear dimensionality reduction technique (*Coifman and Lafon, 2006*) to extract the underlying latent structure of these similarities. The algorithm uses a Markov chain to model transition probabilities between different nodes, using a random walker. The eigenfunctions of this Markov matrix produce a low-dimensional embedding (*Coifman et al., 2005*), where nodes with similar properties are organised along axes represented by eigenvectors, ordered by decreasing amount of total variance explained. Inter-nodal distances are diffusion distances, where larger distances represent less similarity for a feature, such as connectivity strength. It is controlled by two parameters. The first is diffusion time *t*, which controls the scale of the embedding. That is, the diffusion operator sums all paths of length *t* connecting each pair of nodes. The second is anisotropic diffusion $\alpha$, which controls the contribution of the data distribution to the manifold. The data distribution exerts minimal influence on the manifold when $\alpha$=1 (scale-free), and maximum influence when $\alpha$=0. In line with prior work (*Margulies et al., 2016*; *Park et al., 2021*), we set *t*=0, meaning that transitions between nodes *i* and *j* along all walk lengths are considered, and $\alpha$=0.5. Compared to principal component analysis, diffusion-map embedding does not assume a linear underlying data structure, is robust to noise, and preserves local structure (*Coifman and Lafon, 2006*; *Margulies et al., 2016*). Crucially, the pairwise diffusion distances produced are analogous to Euclidean distances and hence can be mapped back onto the cortex.

### Procrustes rotation

For group-level analysis, for each hemisphere we constructed an affinity matrix using a normalised angle kernel and applied diffusion-map embedding. The left hemisphere was then aligned to the right using a Procrustes rotation. For individual-level analysis, eigenvectors for the left hemisphere were aligned with the corresponding group-level rotated eigenvectors. No alignment was applied across datasets. The only exception to this was for structural gradients derived from the referred CALM cohort. Specifically, we aligned the principal gradient of the left hemisphere to the secondary gradient of the right hemisphere: this was due to the first and second gradients explaining a very similar amount of variance, and hence their order was switched.

### Manifold eccentricity

To examine individual differences in the orientation of structural and functional gradients, we calculated manifold eccentricity, originally conceptualised by *Park et al., 2021*. Consider a 3D structural manifold space, where each participant's structural organisation can be represented by a nodal 3D coordinate. Manifold eccentricity is the Euclidean distance between the group centroid, defined as the intersection between the 3 group-level templates, and each node's 3D coordinate. For each participant and modality, we calculated nodal eccentricity, producing a *nroi x ngradient* vector.

Gradient-based measures hold value in developmental contexts, above and beyond traditional graph theory metrics: within a sample of over 600 cognitively healthy adults aged between 18 and 88 years old, sensitivity of gradient-based within-network functional dispersion to age were stronger and more consistent across networks compared to segregation (*Bethlehem et al., 2020*). In the context of microstructural profile covariance, modules resolved by Louvain community detection

occupied distinct positions across the principal two gradients, suggesting that gradients offer a way to meaningfully order discrete graph theory analyses (*Paquola et al., 2019*).

Note that we used a dataset-specific approach when we computed manifold eccentricity for each of the three groups relative to their group-level origin: neurotypical CALM (SC origin = $-7.698 \times 10^{-7}$, FC origin = $6.724 \times 10^{-7}$), neurodivergent CALM (SC origin = $-6.422 \times 10^{-7}$, FC origin = $1.363 \times 10^{-7}$), and NKI (SC origin = $-7.434 \times 10^{-7}$, FC origin = $4.308 \times 10^{-6}$). Eccentricity is a relative measure and thus normalised relative to the origin. Because of this normalisation, each time gradients are constructed the manifold origin is necessarily near-zero, meaning that differences in manifold eccentricity of individual nodes, either between groups or individuals, stem from the eccentricity of that node rather than a difference in origin point.

## Null models to assess inter-gradient relationships

To assess the significance of Spearman-rank correlations between gradients, we used a permutation test preserving spatial autocorrelation, developed for parcellated data by *Váša et al., 2018*. In brief, the parcellated map is projected onto a sphere. FreeSurfer vertices within each parcellated region on the sphere, excluding the medial wall, are averaged to create a centroid, and randomly rotated 10,000 times in each hemisphere. The correlation between the rotated and original coordinates in each hemisphere produces a null distribution.

## Measures of psychopathology and cognition

Using k-nearest neighbours imputation (k=5), with uniform weights, we imputed missing values for the following measures across datasets: forward (5.56%) and backward (5.56%) digit span tasks from the Automated Working Memory Assessment (*Alloway, 2007*) Tower Task total achievement scores (17.94%), visual scanning speed (13.81%), number-letter switching accuracy (16.35%), and motor speed (14.13%) from the Delis-Kaplan assessment suite (*Delis et al., 2001*) inattention (2.22%), hyperactivity (2.22%), peer relations (2.22%), learning problems (2.06%), executive function (2.06%), and aggression (2.22%) from the Conners III scale (*Conners et al., 2011*). Only raw cognitive scores were provided, and hence we age-standardised the scores by regressing out age and age$^2$.

As we removed participants with high motion, this may have overlapped with those with higher psychopathology scores, and thus incomplete coverage. To examine coverage and sensitivity to broad-range psychopathology following quality control, we calculated the Fisher-Pearson skewness statistic $g_1$ for each of the six Conners $t$-statistic measures and the proportion of youth with a $t$-statistic equal to or greater than 65, indicating an elevated or very elevated score. Measures of inattention ($g_1$=0.11, 44.20% elevated), hyperactivity/impulsivity ($g_1$=0.48, 36.41% elevated), learning problems ($g_1$=0.45, 37.36% elevated), executive functioning ($g_1$=0.27, 38.16% elevated), aggression ($g_1$=1.65, 15.58% elevated), and peer relations ($g_1$=0.49, 38% elevated) were positively skewed and comprised of at least 15% of children with elevated or very elevated scores, suggesting sufficient coverage of those with extreme scores.

### Missing data

To avoid a loss of statistical power, we imputed missing data. 27.50% of the sample had one or more missing psychopathology or cognitive measures (equal to 7% of all values), and the data was not missing at random: using a Welch's $t$-test, we observed a significant effect of missingness on age [$t$(264.479)=3.029, p=0.003, Cohen's $d$=0.296], whereby children with missing data ($M$=12.055 years, SD = 3.272) were younger than those with complete data ($M$=12.902 years, SD = 2.685). Using a subset with complete data (N=456), we randomly sampled 10% of the values in each column with replacement and assigned those as missing, thereby mimicking the proportion of missingness in the entire dataset. We conducted KNN imputation (uniform weights) on the subset with complete data and calculated the imputation accuracy as the root mean squared error normalised by the observed range of each measure. Thus, each measure was assigned a percentage which described the imputation margin of error. Across cognitive measures, imputation was within a 5.40% mean margin of error, with the lowest imputation error in the Trail motor speed task (4.43%) and highest in the Trails number-letter switching task (7.19%). Across psychopathology measures, imputation exhibited a mean 7.81% error margin, with the lowest imputation error in the Conners executive function scale (5.75%) and

the highest in the Conners peer relations scale (11.04%). Together, this suggests that imputation was accurate.

## GAMMs

Using R 4.2.2, we modelled linear and non-linear relationships between age and manifold eccentricity using generalised additive mixed modelling (GAMMs). Like generalised additive models, GAMMs model covariates as a smooth penalised spline, the weighted sum of $k$ basis functions. However, GAMMs additionally include random effects to model correlations between observations, for example in data sets with cross-sectional and longitudinal data. The smooth is decomposed into a combination of fixed unpenalised linear components and random penalised non-linear components (*Sørensen et al., 2021*). We used the gamm4 R package to specify smooths and parametric coefficients (*Lin and Zhang, 1999*), and the mgcv R package to specify tensors (*Wood, 2017*). Mean framewise displacement, sex, and data set were parametric covariates, whilst age and an interaction between age and data set were smooths. Subject-specific random intercepts were included as random effects. To prevent overfitting, we set a maximum of three basis functions. When comparing manifold eccentricity across seven intrinsic functional connectivity networks (*Schaefer et al., 2018*), we corrected for multiple comparisons by controlling the false discovery rate.

To visualise the interaction between age and data set as a smooth spline, we predicted manifold eccentricity using $n$ increments of age from youngest to oldest for each data set, where $n$ was the number of participants in that data set. Median mean framewise displacement for each data set and a single factor of sex were used as covariates to generate predictions. Following previous work (*Sydnor et al., 2023*; *Tervo-Clemmens et al., 2023*), to generate 95% credible intervals, we took 10,000 draws from a multivariate gaussian distribution with a covariance matrix of the fitted GAMM and generated a critical value by calculating the maximum absolute standardised deviation of the draws from the gaussian distribution to the true GAMM distribution. We calculated the first derivative of the age smooth for each data set and defined sensitive periods of development as those with non-zero simultaneous confidence intervals ($p<0.05$, two-tailed).

When using our GAMMs to test for the relationship between cognition and psychopathology and our coupling metrics, we opted to predict structure-function coupling using cognitive or psychopathological dimensions, rather than vice versa, to minimise multiple comparisons. In the current framework, we corrected for eight multiple comparisons within each domain. This would have increased to 16 multiple comparison corrections for predicting two cognitive dimensions using network-level coupling and 24 multiple comparison corrections for predicting three psychopathology dimensions. Incorporating multiple networks as predictors within the same regression framework introduces collinearity, whilst the behavioural dimensions were orthogonal: for example, coupling is strongly correlated between the somato-motor and ventral attention networks ($r_s = 0.721$), between the default-mode and frontoparietal networks ($r_s = 0.670$), and between the dorsal attention and frontoparietal networks ($r_s = 0.650$).

## GAMM cross-validation

We implemented a fivefold cross validation procedure, stratified by dataset (two levels: CALM or NKI). All observations from any given participant were assigned to either the testing or training fold, to prevent data leakage, and the cross-validation procedure was repeated 100 times, to account for randomness in data splits. The outcome was predicted global or network-level structure-function coupling across all test splits, operationalised as the Spearman rank correlation coefficient. To assess whether prediction accuracy exceeded chance, we compared empirical prediction accuracy with that of GAMMs trained and tested on null data in which cognitive factor 2 scores were permuted across subjects. The number of observations formed three exchangeability blocks (N=320 with one observation, N=105 with two observations, and N=33 with three observations), whereby scores from a participant with two observations were replaced by scores from another participant with two observations, with participant-level scores kept together, and so on for all numbers of observations. We compared empirical and null prediction accuracies using independent sample t-tests as, although the same participants were examined, the shuffling meant that the relative ordering of participants within both distributions was not preserved. For parallelisation and better stability when estimating models fit on permuted data, we used the bam function from the mgcv R package (*Wood, 2017*).

## Code/data availability

Detailed documentation about how to implement these analyses is available here (copy archived at *Monaghan, 2026*). CALM and NKI are both managed-access datasets. However, to qualified researchers, we provide harmonised, longitudinal, multi-modal connectomes spanning the full neurotypical-neurodivergent range, covering common dimensions of cognition and psychopathology. This builds upon existing data sharing initiatives for primarily cross-sectional data and psychopathology (*Shafiei et al., 2025*).

## Acknowledgements

We thank the families of NKI and CALM for their participation, alongside Theodore Satterthwaite and Valerie Sydnor for valuable discussions. AM acknowledges support of the Cambridge ESRC Doctoral Training Programme and the UK Medical Research Council (MRC). DEA is supported by the Gnodde Goldman Sachs endowed Professorship in Neuroinformatics awarded to the University of Cambridge. Both DA and DEA are supported by the James S McDonnell Foundation Opportunity Award and the Templeton World Charity Foundation (TCWF-2022-30510). AM and DEA are supported by the MRC Programme Grant MC-A0606-5PQ41. RAIB is supported by the HDR UK Molecular to Health Records Program and an Academy of Medical Sciences Springboard Award. DA was also supported by the Imperial College Research Fellowship with support from Schmidt Sciences and a Nature Computes Better Opportunity Seed with the Advanced Research + Invention Agency. DM received funding from the European Research Council under the European Union's Horizon 2020 research and innovation programme (grant agreement No. 866533-CORTIGRAD), the Wellcome Trust Core Award and the NIH Oxford BRC. All research at the Department of Psychiatry at the University of Cambridge is supported by the National Institute for Health and Care Research Cambridge Biomedical Research Centre (NIHR203312) and the NIHR Applied Research Collaboration East of England.

## Additional information

### Competing interests

Richard AI Bethlehem: R.A.I.B. serves as a director of and holds equity in Centile Bioscience Inc. The other authors declare that no competing interests exist.

### Funding

| Funder | Grant reference number | Author |
|---|---|---|
| Templeton World Charity Foundation | TCWF-2022-30510 | Duncan E Astle<br>Danyal Akarca |
| Health Data Research UK | Molecular to Health Records Program | Richard AI Bethlehem |
| Academy of Medical Sciences | Springboard Award | Richard AI Bethlehem |
| Cambridge ESRC Doctoral Training Programme | | Alicja Monaghan |
| European Union's Horizon 2020 research and innovation programme | No. 866533-CORTIGRAD | Daniel S Margulies |
| Schmidt Sciences | Imperial College Research Fellowship | Danyal Akarca |
| Wellcome Trust | Core Award | Daniel S Margulies |
| National Institute of Health | Oxford BRC | Daniel S Margulies |
| James S. McDonnell Foundation | Opportunity Award | Danyal Akarca<br>Duncan E Astle |

| Funder | Grant reference number | Author |
|---|---|---|
| UK Medical Research Council | MC-A0606-5PQ41 | Alicja Monaghan<br>Duncan E Astle |
| National Institute for Health and Care Research Cambridge Biomedical Research Centre | NIHR203312 | Duncan E Astle |

The funders had no role in study design, data collection and interpretation, or the decision to submit the work for publication. For the purpose of Open Access, the authors have applied a CC BY public copyright license to any Author Accepted Manuscript version arising from this submission

## Author contributions

Alicja Monaghan, Conceptualization, Data curation, Formal analysis, Investigation, Visualization, Methodology, Writing – original draft, Project administration; Richard AI Bethlehem, Supervision, Investigation, Writing – review and editing; Danyal Akarca, Formal analysis, Visualization, Writing – review and editing; Daniel S Margulies, Investigation, Writing – review and editing; the CALM Team, Data curation, Funding acquisition, Project administration, Resources; Duncan E Astle, Formal analysis, Supervision, Investigation, Methodology, Writing – original draft, Writing – review and editing

## Author ORCIDs

Alicja Monaghan ⓘ https://orcid.org/0000-0002-1283-9845
Richard AI Bethlehem ⓘ https://orcid.org/0000-0002-0714-0685
Danyal Akarca ⓘ https://orcid.org/0000-0002-5931-0295
Daniel S Margulies ⓘ https://orcid.org/0000-0002-8880-9204
Duncan E Astle ⓘ https://orcid.org/0000-0002-7042-5392

## Ethics

Human subjects: Ethical approval was obtained for using CALM and NKI data as secondary data sources.

Reviewer #2 (Public review): https://doi.org/10.7554/eLife.103097.3.sa1
Author response https://doi.org/10.7554/eLife.103097.3.sa2

# Additional files

## Supplementary files

MDAR checklist

## Data availability

Detailed documentation about how to implement these analyses, including de-identified derivatives from diffusion-map embedding, are available at: https://github.com/AlicjaMonaghan/neurodevelopmental_gradients (copy archived at *Monaghan, 2026*).

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
