## [Editor Report · eLife Assessment]

This **important** study provides insights into the neurodevelopmental trajectories of structural and functional connectivity gradients in the human brain and their potential associations with behaviour and psychopathology. The evidence supporting the findings is **solid**. This study will be of interest to neuroscientists interested in understanding functional connectivity across development.

---

## [Referee Report · Reviewer #2 (Public review)]

Summary:

This study aims to show how structural and functional brain organization develops during childhood and adolescence using two large neuroimaging datasets. It addresses whether core principles of brain organization are stable across development, how they change over time, and how these changes relate to cognition and psychopathology. The study finds that brain organization is established early and remains stable but undergoes gradual refinement, particularly in higher-order networks. Structural-functional coupling is linked to better working memory but shows no clear relationship with psychopathology.

Comments on revisions:

Follow-up: I would like to thank the authors for their thoughtful and comprehensive revisions. The additional analyses addressing developmental differences in structure-function coupling between CALM and NKI are valuable and clearly strengthen the manuscript. I particularly appreciate the inclusion of the neurotypical subgroup within CALM to disentangle neurotypicality from potential site-related effects, as well as the expanded discussion of these findings in the context of individual variability and equifinality.

Regarding my earlier comment on the use of COMBAT, I realize that "exclusion" may have been a poor choice of wording. What I meant was that harmonization procedures like COMBAT can, in some cases, weaken extremes or reduce variability by shrinking values toward the mean, rather than literally excluding participants from the analysis. Nevertheless, I appreciate the authors' careful consideration of this point and their additional analysis examining sample coverage following motion-based exclusions.

Overall, I am satisfied with the revisions, and I believe the manuscript has been substantially improved.

---

## [Author Response]

The following is the authors’ response to the original reviews.

**Reviewer #1 (Public review):**
Lack of Sensitivity Analyses for some Key Methodological Decisions: Certain methodological choices in this manuscript diverge from approaches used in previous works. In these cases, I recommend the following: (i) The authors could provide a clear and detailed justification for these deviations from established methods, and (ii) supplementary sensitivity analyses could be included to ensure the robustness of the findings, demonstrating that the results are not driven primarily by these methodological changes. Below, I outline the main areas where such evaluations are needed:

This detailed guidance is incredibly valuable, and we are grateful. Work of this kind is in its relative infancy, and there are so many design choices depending on the data available, questions being addressed, and so on. Help us navigate that has been extremely useful. In our revised manuscript we are very happy to add additional justification for design choices made, and wherever possible test the impact of those choices. It is certainly the case that different approaches have been used across the handful of papers published in this space, and, unlike in other areas of systems neuroscience, we have yet to reach the point where any of these approaches are established. We agree with the reviewer that wherever possible these design choices should be tested.

Use of Communicability Matrices for Structural Connectivity Gradients: The authors chose to construct structural connectivity gradients using communicability matrices, arguing that diffusion map embedding "requires a smooth, fully connected matrix." However, by definition, the creation of the affinity matrix already involves smoothing and ensures full connectedness. I recommend that the authors include an analysis of what happens when the communicability matrix step is omitted. This sensitivity test is crucial, as it would help determine whether the main findings hold under a simpler construction of the affinity matrix. If the results significantly change, it could indicate that the observations are sensitive to this design choice, thereby raising concerns about the robustness of the conclusions. Additionally, if the concern is related to the large range of weights in the raw structural connectivity (SC) matrix, a more conventional approach is to apply a log-transformation to the SC weights (e.g., log(1+𝑆𝐶_𝑖𝑗_)), which may yield a more reliable affinity matrix without the need for communicability measures.

The reason we used communicability is indeed partly because we wanted to guarantee a smooth fully connected matrix, but also because our end goal for this project was to explore structure-function coupling in these low-dimensional manifolds. Structural communicability – like standard metrics of functional connectivity – includes both direct and indirect pathways, whereas streamline counts only capture direct communication. In essence we wanted to capture not only how information might be routed from one location to another, but also the more likely situation in which information propagates through the system.

In the revised manuscript we have given a clearer justification for why we wanted to use communicability as our structural measure (Page 4, Line 179):

“To capture both direct and indirect paths of connectivity and communication, we generated weighted communicability matrices using SIFT2-weighted fibre bundle capacity (FBC). These communicability matrices reflect a graph theory measure of information transfer previously shown to maximally predict functional connectivity (Esfahlani et al., 2022; Seguin et al., 2022). This also foreshadowed our structure-function coupling analyses, whereby network communication models have been shown to increase coupling strength relative to streamline counts (Seguin et al., 2020)”.

We have also referred the reader to a new section of the Results that includes the structural gradients based on the streamline counts (Page 7, line 316):

“Finally, as a sensitivity analysis, to determine the effect of communicability on the gradients, we derived affinity matrices for both datasets using a simpler measure: the log of raw streamline counts. The first 3 components derived from streamline counts compared to communicability were highly consistent across both NKI (r_s_ = 0.791, r_s_ = 0.866, r_s_ = 0.761) and the referred subset of CALM (r_s_ = 0.951, r_s_ = 0.809, r_s_ = 0.861), suggesting that in practice the organisational gradients are highly similar regardless of the SC metric used to construct the affinity matrices”.

Methodological ambiguity/lack of clarity in the description of certain evaluation steps: Some aspects of the manuscript’s methodological description are ambiguous, making it challenging for future readers to fully reproduce the analyses based on the information provided. I believe the following sections would benefit from additional detail and clarification:Computation of Manifold Eccentricity: The description of how eccentricity was computed (both in the results and methods sections) is unclear and may be problematic. The main ambiguity lies in how the group manifold origin was defined or computed. (1) In the results section, it appears that separate manifold origins were calculated for the NKI and CALM groups, suggesting a dataset-specific approach. (2) Conversely, the methods section implies that a single manifold origin was obtained by somehow combining the group origins across the three datasets, which seems contradictory. Moreover, including neurodivergent individuals in defining the central group manifold origin in conceptually problematic. Given that neurodivergent participants might exhibit atypical brain organization, as suggested by Figure 1, this inclusion could skew the definition of what should represent a typical or normative brain manifold. A more appropriate approach might involve constructing the group manifold origin using only the neurotypical participants from both the NKI and CALM datasets. Given the reported similarity between group-level manifolds of neurotypical individuals in CALM and NKI, it would be reasonable to expect that this combined origin should be close to the origin computed within neurotypical samples of either NKI or CALM. As a sanity check, I recommend reporting the distance of the combined neurotypical manifold origin to the centres of the neurotypical manifolds in each dataset. Moreover, if the manifold origin was constructed while utilizing all samples (including neurodivergent samples) I think this needs to be reconsidered.

This is a great point, and we are very happy to clarify. Separate manifolds were calculated for the NKI and CALM participants, hence a dataset-specific approach. Indeed, in the long-run our goal was to explore individual differences in these manifolds, relative to the respective group-level origins, and their intersection across modalities, so manifold eccentricity was calculated at an individual level for subsequent analyses. At the group level, for each modality, we computed 3 manifold origins: one for NKI, one for the referred subset of CALM, and another for the neurotypical portion of CALM. Crucially, because the manifolds are always normal, in each case the manifold origin point is near-zero (extremely near-zero, to the 6^th^ or 7^th^ decimal place). In other words, we do indeed calculate the origin separately each time we calculate the gradients, but the origin is zero in every case. As a result, differences in the origin point cannot be the source of any differences we observe in manifold eccentricity between groups or individuals. We have updated the Methods section with the manifold origin points for each dataset and clarified our rationale (Page 16, Line 1296):

“Note that we used a dataset-specific approach when we computed manifold eccentricity for each of the three groups relative to their group-level origin: neurotypical CALM (SC origin = -7.698 x 10^-7^, FC origin = 6.724 x 10^-7^), neurodivergent CALM (SC origin = -6.422 x 10 , FC origin = 1.363 x 10), and NKI (SC origin = -7.434 x 10 , FC origin = 4.308 x 10^-6^). Eccentricity is a relative measure and thus normalised relative to the origin. Because of this normalisation, each time gradients are constructed the manifold origin is necessarily near-zero, meaning that differences in manifold eccentricity of individual nodes, either between groups or individuals, are stem from the eccentricity of that node rather than a difference in origin point”.

We clarified the computation of the respective manifold origins within the Results section, and referred the reader to the relevant Methods section (Page 9, line 446):

“For each modality (2 levels: SC and FC) and dataset (3 levels: neurotypical CALM, neurodivergent CALM, and NKI), we computed the group manifold origin as the mean of their respective first three gradients. Because of the normal nature of the manifolds this necessarily means that these origin points will be very near-zero, but we include the exact values in the ‘Manifold Eccentricity’ methodology sub-section”.

Individual-Level Gradients vs. Group-Level Gradients: Unlike previous studies that examined alterations in principal gradients (e.g., Xia et al., 2022; Dong et al., 2021), this manuscript focuses on gradients derived directly from individual-level data. In contrast, earlier works have typically computed gradients based on grouped data, such as using a moving window of individuals based on age (Xia et al.) or evaluating two distinct age groups (Dong et al.). I believe it is essential to assess the sensitivity of the findings to this methodological choice. Such an evaluation could clarify whether the observed discrepancies with previous reports are due to true biological differences or simply a result of different analytical strategies.

This is a brilliant point. The central purpose of our project was to test how individual differences in these gradients, and their intersection across modalities, related to differences in phenotype (e.g. cognitive difficulties). These necessitated calculating gradients at the level of individuals and building a pipeline to do so, given that we could find no other examples. Nonetheless, despite this different goal and thus approach, we had expected to replicate a couple of other key findings, most prominently the ‘swapping’ of gradients shown by Dong et al. (2021). We were also surprised that we did not find this changing in order. The reviewer is right and there could be several design features that produce the difference, and in the revised manuscript we test several of them. We have added the following text to the manuscript as a sensitivity analysis for the Results sub-section titled “Stability of individual-level gradients across developmental time” (Page 7, Line 344 onwards):

“One possibility is that our observation of gradient stability – rather than a swapping of the order for the first two gradients (Dong et al., 2021) – is because we calculated them at an individual level. To test this, we created subgroups and contrasted the first two group-level structural and functional gradients derived from children (younger than 12 years old) versus those from adolescents (12 years old and above), using the same age groupings as prior work (Dong et al., 2021). If our use of individually calculated gradients produces the stability, then we should observe the swapping of gradients in this sensitivity analysis. Using baseline scans from NKI, the primary structural gradient in childhood (N = 99) as shown in Figure 1f, this was highly correlated (r_s_ = 0.995) with those derived from adolescents (N = 123). Likewise, the secondary structural gradient in childhood was highly consistent in adolescence (r_s_ = 0.988). In terms of functional connectivity, the principal gradient in childhood (N = 88) was highly consistent in adolescence (r_s_ = 0.990, N = 125). The secondary gradient in childhood was again highly similar in adolescence (r_s_ = 0.984). The same result occurred in the CALM dataset: In the baseline referred subset of CALM, the primary and secondary communicability gradients derived from children (N = 258) and adolescents (N = 53) were near-identical (r_s_ = 0.991 and r_s_ = 0.967, respectively). Alignment for the primary and secondary functional gradients derived from children (N = 130) and adolescents (N = 43) were also near-identical (r_s_ = 0.972 and r_s_ = 0.983, respectively). These consistencies across development suggest that gradients of communicability and functional connectivity established in childhood are the same as those in adolescence, irrespective of group-level or individual-level analysis. Put simply, our failure to replicate the swapping of gradient order in Dong et al. (2021) is not the result of calculating gradients at the level of individual participants.”

Procrustes Transformation: It is unclear why the authors opted to include a Procrustes transformation in this analysis, especially given that previous related studies (e.g., Dong et al.) did not apply this step. I believe it is crucial to evaluate whether this methodological choice influences the results, particularly in the context of developmental changes in organizational gradients. Specifically, the Procrustes transformation may maximize alignment to the group-level gradients, potentially masking individual-level differences. This could result in a reordering of the gradients (e.g., swapping the first and second gradients), which might obscure true developmental alterations. It would be informative to include an analysis showing the impact of performing vs. omitting the Procrustes transformation, as this could help clarify whether the observed effects are robust or an artifact of the alignment procedure. (Please also refer to my comment on adding a subplot to Figure 1). Additionally, clarifying how exactly the transformation was applied to align gradients across hemispheres, individuals, and/or datasets would help resolve ambiguity.

The current study investigated individual differences in connectome organisation, rather than group-level trends (Dong et al., 2021). This necessitates aligning individual gradients to the corresponding group-level template using a Procrustes rotation. Without a rotation, there is no way of knowing if you are comparing ‘like with like’: the manifold eccentricity of a given node may appear to change across individuals simply due to subtle differences in the arbitrary orientation of the underlying manifolds. We also note that prior work examining individual differences in principal alignment have used Procrustes (Xia et al., 2022), who demonstrated emergence of the principal gradient across development, albeit with much smaller effects than Dong and colleagues (2021). Nonetheless, we agree, the Procrustes rotation could be another source of the differences we observed with the previous paper (Dong et al. 2021). We explored the impact of the Procrustes rotation on individual gradients as our next sensitivity analysis. We recalculated everyone’s gradients without Procrustes rotation. We then tested the alignment of each participant with the group-level gradients using Spearman’s correlations, followed by a series of generalised linear models to predict principal gradient alignment using head motion, age, and sex. The expected swapping of the first and second functional gradient (Dong et al., 2021) would be represented by a decrease in the spatial similarity of each child’s principal functional gradient to the principal childhood group-level gradient, at the onset of adolescence (~age 12). However, there is no age effect on this unrotated alignment, suggesting that the lack of gradient swapping in our data does not appear to be the result of the Procrustes rotation. When you use unrotated individual gradients the alignment is remarkably consistent across childhood and adolescence. Alignment is, however, related to head motion, which is often related to age. To emphasise the importance of motion, particularly in relation to development, we conducted a mediation analysis between the relationship between age and principal alignment (without correcting for motion), with motion as a mediator, within the NKI dataset. Before accounting for motion, the relationship between age and principal alignment is significant, but this can be entirely accounted for by motion. In our revised manuscript we have included this additional analysis in the Results sub-section titled “Stability of individual-level gradients across developmental time”, following on from the above point about the effect of group-level versus individual-level analysis (Page 8, Line 400):

“A second possible discrepancy between our results and that of prior work examining developmental change in group-level functional gradients (Dong et al., 2021) was the use of Procrustes alignment. Such alignment of individual-level gradients to group-level templates is a necessary step to ensure valid comparisons between corresponding gradients across individuals, and has been implemented in sliding-window developmental work tracking functional gradient development (Xia et al., 2022). Nonetheless, we tested whether our observation of stable principal functional and communicability gradients may be an artefact of the Procrustes rotation. We did this by modelling how individual-level alignment without Procrustes rotation to the group-level templates varies with age, head motion, and sex, as a series of generalised linear models. We included head motion as the magnitude of the Procrustes rotation has been shown to be positively correlated with mean framewise displacement (Sasse et al., 2024), and prior group-level work (Dong et al., 2021) included an absolute motion threshold rather than continuous motion estimates. Using the baseline referred CALM sample, there was no significant relationship between alignment and age (β = -0.044, 95% CI = [-0.154, 0.066], p = 0.432) after accounting for head motion and sex. Interestingly, however head motion was significantly associated with alignment (β = -0.318, 95% CI = [-0.428, -.207], p = 1.731 x 10^-8^), such that greater head motion was linked to weaker alignment. Note that older children tended to have exhibit less motion for their structural scans (r_s_ = 0.335, p < 0.001). We observed similar trends in functional alignment, whereby tighter alignment was significantly predicted by lower head motion (β = -0.370, 95% CI = [-0.509, -0.231], p = 1.857 x 10^-7^), but not by age (β = 0.049, 95% CI = [-0.090, 0.187], p = 0.490). Note that age and head motion for functional scans were not significantly related (r_s_ = -0.112, p = 0.137). When repeated for the baseline scans of NKI, alignment with the principal structural gradient was not significantly predicted by either scan age (β = 0.019, 95% CI = [-0.124, 0.163], p = 0.792) or head motion (β = -0.133, 95% CI = [-0.175, 0.009], p = 0.067) together in a single model, where age and motion were negatively correlated (r_s_ = -0.355, p < 0.001). Alignment with the principal functional gradient was significantly predicted by head motion (β = -0.183, 95% CI = [-0.329, -0.036], p = 0.014) but not by age (β = 0.066, 95% CI = [-0.081, 0.213], p = 0.377), where age and motion were also negatively correlated (r_s_ = -0.412, p < 0.001). Across modalities and datasets, alignment with the principal functional gradient in NKI was the only example in which there was a significant correlation between alignment and age (r_s_ = 0.164, p = 0.017) before accounting for head motion and sex. This suggests that apparent developmental effects on alignment are minimal, and where they do exist they are removed after accounting for head motion. Put together this suggests that the lack of order swapping for the first two gradients is not the result of the Procrustes rotation – even without the rotation there is no evidence for swapping”.

“To emphasise the importance of head motion in the appearance of developmental change in alignment, we examined whether accounting for head motion removes any apparent developmental change within NKI. Specifically, we tested whether head motion mediates the relationship between age and alignment (Figure 1X), controlling for sex, given that higher motion is associated with younger children (β = -0.429, 95% CI = [0.552, -0.305], p = 7.957 x 10^-11^), and stronger alignment is associated with reduced motion (β = -0.211, 95% CI = [-0.344, -0.078], p = 2.017 x 10^-3^). Motion mediated the relationship between age and alignment (β = 0.078, 95% CI = [0.006, 0.146], p = 1.200 x 10^-2^), accounting for 38.5% variance in the age-alignment relationship, such that the link between age and alignment became non-significant after accounting for motion (β = 0.066, 95% CI = [-0.081, 0.214], p = 0.378). This firstly confirms our GLM analyses, where we control for motion and find no age associations. Moreover, this suggests that caution is required when associations between age and gradients are observed. In our analyses, because we calculate individual gradients, we can correct for individual differences in head motion in all our analyses. However, other than using an absolute motion threshold and motion-matched child and adolescent groups, individual differences in motion were not accounted for by prior work which demonstrated a flipping of the principal functional gradients with age (Dong et al., 2021)”.

We further clarify the use of Procrustes rotation as a separate sub-section within the Methods (Page 25, Line 1273):

“Procrustes Rotation

For group-level analysis, for each hemisphere we constructed an affinity matrix using a normalized angle kernel and applied diffusion-map embedding. The left hemisphere was then aligned to the right using a Procrustes rotation. For individual-level analysis, eigenvectors for the left hemisphere were aligned with the corresponding group-level rotated eigenvectors. No alignment was applied across datasets. The only exception to this was for structural gradients derived from the referred CALM cohort. Specifically, we aligned the principal gradient of the left hemisphere to the secondary gradient of the right hemisphere: this was due to the first and second gradients explaining a very similar amount of variance, and hence their order was switched”.

SC-FC Coupling Metric: The approach used to quantify nodal SC-FC coupling in this study appears to deviate from previously established methods in the field. The manuscript describes coupling as the "Spearman-rank correlation between Euclidean distances between each node and all others within structural and functional manifolds," but this description is unclear and lacks sufficient detail. Furthermore, this differs from what is typically referred to as SC-FC coupling in the literature. For instance, the cited study by Park et al. (2022) utilizes a multiple linear regression framework, where communicability, Euclidean distance, and shortest path length are independent variables predicting functional connectivity (FC), with the adjusted R-squared score serving as the coupling index for each node. On the other hand, the Baum et al. (2020) study, also cited, uses Spearman correlation, but between raw structural connectivity (SC) and FC values. If the authors opt to introduce a novel coupling metric, it is essential to demonstrate its similarity to these previous indices. I recommend providing an analysis (supplementary) showing the correlation between their chosen metric and those used in previous studies (e.g., the adjusted R-squared scores from Park et al. or the SC-FC correlation from Baum et al.). Furthermore, if the metrics are not similar and results are sensitive to this alternative metric, it raises concerns about the robustness of the findings. A sensitivity analysis would therefore be helpful (in case the novel coupling metric is not like previous ones) to determine whether the reported effects hold true across different coupling indices.

This is a great point, and we are happy to take the reviewer’s recommendation. There are multiple different ways of calculating structure-function coupling. For our set of questions, it was important that our metric incorporated information about the structural and functional manifolds, rather than being a separate approach that is unrelated to these low-dimensional embeddings. Put simply, we wanted our coupling measure to be about the manifolds and gradients outlined in the early sections of the results. We note that the multiple linear regression framework was developed by Vázquez-Rodríguez and colleagues (2019), whilst the structure-function coupling computed in manifold space by Park and colleagues (2022) was operationalised as a linear correlation between z-transformed functional connectomes and structural differentiation eigenvectors. To clarify how this coupling was calculated, and to justify why we developed a new coupling method based on manifolds rather than borrow an existing approach from the literature, we have revised the manuscript to make this far clearer for readers (Page 13, line 604):

“To examine the relationship between each node’s relative position in structural and functional manifold space, we turned our attention to structure-function coupling. Whilst prior work typically computed coupling using raw streamline counts and functional connectivity matrices, either as a correlation (Baum et al., 2020) or through a multiple linear regression framework (Vázquez-Rodríguez et al., 2019), we opted to directly incorporate low-dimensional embeddings within our coupling framework. Specifically, as opposed to correlating row-wise raw functional connectivity with structural connectivity eigenvectors (Park et al., 2022), our metric directly incorporates the relative position of each node in low-dimensional structural and functional manifold spaces. Each node was situated in a low-dimensional 3D space, the axes of which were each participant’s gradients, specific to each modality. For each participant and each node, we computed the Euclidean distance with all other nodes within structural and functional manifolds separately, producing a vector of size 200 x 1 per modality. The nodal coupling coefficient was the Spearman correlation between each node’s Euclidean distance to all other nodes in structural manifold space, and that in functional manifold space. Put simply, a strong nodal coupling coefficient suggests that that node occupies a similar location in structural space, relative to all other nodes, as it does in functional space”.

We also agree with the reviewer’s recommendation to compare this to some of the more standard ways of calculating coupling. We compare our metric with 3 others (Baum et al., 2020; Park et al., 2022; VázquezRodríguez et al., 2019), and find that all metrics capture the core developmental sensorimotor-to-association axis (Sydnor et al., 2021). Interestingly, manifold-based coupling measures captured this axis more strongly than non-manifold measures. We have updated the Results accordingly (Page 14, Line 638):

“To evaluate our novel coupling metric, we compared its cortical spatial distribution to three others (Baum et al., 2020; Park et al., 2022; Vázquez-Rodríguez et al., 2019), using the group-level thresholded structural and functional connectomes from the referred CALM cohort. As shown in Figure 4c, our novel metric was moderately positively correlated to that of a multi-linear regression framework (r_s_ = 0.494, p_spin_ = 0.004; Vázquez-Rodríguez et al., 2019) and nodal correlations of streamline counts and functional connectivity (r_s_ = 0.470, p_spin_ = 0.005; Baum et al., 2020). As expected, our novel metric was strongly positively correlated to the manifold-derived coupling measure (r_s_ = 0.661, p_spin_ < 0.001; Park et al., 2022), more so than the first (Z(198) = 3.669, p < 0.001) and second measure (Z(198) = 4.012, p < 0.001). Structure-function coupling is thought to be patterned along a sensorimotor-association axis (Sydnor et al., 2021): all four metrics displayed weak-tomoderate alignment (Figure 4c). Interestingly, the manifold-based measures appeared most strongly aligned with the sensorimotor-association axis: the novel metric was more strongly aligned than the multi-linear regression framework (Z(198) = -11.564, p < 0.001) and the raw connectomic nodal correlation approach (Z(198) = -10.724, p < 0.001), but the previously-implemented structural manifold approach was more strongly aligned than the novel metric (Z(198) = -12.242, p < 0.001). This suggests that our novel metric exhibits the expected spatial distribution of structure-function coupling, and the manifold approach more accurately recapitulates the sensorimotor-association axis than approaches based on raw connectomic measures”.

We also added the following to the legend of Figure 4 on page 15:

“d. The inset Spearman correlation plot of the 4 coupling measures shows moderate-to-strong correlations (p_spin_ < 0.005 for all spatial correlations). The accompanying lollypop plot shows the alignment between the sensorimotor-to-association axis and each of the 4 coupling measures, with the novel measure coloured in light purple (p_spin_ < 0.007 for all spatial correlations)”.

Prediction vs. Association Analysis: The term “prediction” is used throughout the manuscript to describe what appear to be in-sample association tests. This terminology may be misleading, as prediction generally implies an out-of-sample evaluation where models trained on a subset of data are tested on a separate, unseen dataset. If the goal of the analyses is to assess associations rather than make true predictions, I recommend refraining from the term “prediction” and instead clarifying the nature of the analysis. Alternatively, if prediction is indeed the intended aim (which would be more compelling), I suggest conducting the evaluations using a k-fold cross-validation framework. This would involve training the Generalized Additive Mixed Models (GAMMs) on a portion of the data and training their predictive accuracy on a held-out sample (i.e. different individuals). Additionally, the current design appears to focus on predicting SC-FC coupling using cognitive or pathological dimensions. This is contrary to the more conventional approach of predicting behavioural or pathological outcomes from brain markers like coupling. Could the authors clarify why this reverse direction of analysis was chosen? Understanding this choice is crucial, as it impacts the interpretation and potential implications of the findings.

We have replaced “prediction” with “association” across the manuscript. However, for analyses corresponding to Figure 5, which we believe to be the most compelling, we conducted a stratified 5-fold cross-validation procedure, outlined below, repeated 100 times to account for random variation in the train-test splits. To assess whether prediction accuracy in the test splits was significantly greater than chance, we compared our results to those derived from a null dataset in which cognitive factor 2 scores had been permuted across participants. To account for the time-series element and block design of our data, in that some participants had 2 or more observations, we permuted entire participant blocks of cognitive factor 2 scores, keeping all other variables, including covariates, the same. Included in our manuscript are methodological details and results pertaining to this procedure. Specifically, the following has been added to the Results (Page 16, Line 758):

“To examine the predictive value of the second cognitive factor for global and network-level structure-function coupling, operationalised as a Spearman rank correlation coefficient, we implemented a stratified 5-fold crossvalidation framework, and predictive accuracy compared with that of a null data frame with cognitive factor 2 scores permuted across participant blocks (see ‘GAMM cross-validation’ in the Methods). This procedure was repeated 100 times to account for randomness in the train-test splits, using the same model specification as above. Therefore, for each of the 5 network partitions in which an interaction between the second cognitive factor and age was a significant predictor of structure-function coupling (global, visual, somato-motor, dorsal attention, and default-mode), we conducted a Welch’s independent-sample t-test to compare 500 empirical prediction accuracies with 500 null prediction accuracies. Across all 5 network partitions, predictive accuracy of coupling was significantly higher than that of models trained on permuted cognitive factor 2 scores (all p < 0.001). We observed the largest difference between empirical (M = 0.029, SD = 0.076) and null (M = -0.052, SD = 0.087) prediction accuracy in the somato-motor network [t (980.791) = 15.748, p < 0.001, Cohen’s d = 0.996], and the smallest difference between empirical (M = 0.080, SD = 0.082) and null (M = 0.047, SD = 0.081) prediction accuracy in the dorsal attention network [t (997.720) = 6.378, p < 0.001, Cohen’s d = 0.403]. To compare relative prediction accuracies, we ordered networks by descending mean accuracy and conducted a series of Welch’s independent sample t-tests, followed by FDR correction (Figure 5X). Prediction accuracy was highest in the default-mode network (M = 0.265, SD = 0.085), two-fold that of global coupling (t(992.824) = 25.777, p_FDR_ = 5.457 x 10^-112^, Cohen’s d = 1.630, M = 0.131, SD = 0.079). Global prediction accuracy was significantly higher than the visual network (t (992.644) = 9.273, p_FDR_ = 1.462 x 10^-19^, Cohen’s d = 0.586, M = 0.083, SD = 0.085), but visual prediction accuracy was not significantly higher than within the dorsal attention network (t (997.064) = 0.554, p_FDR_ = 0.580, Cohen’s d = 0.035, M = 0.080, SD = 0.082). Finally, prediction accuracy within the dorsal attention network was significantly stronger than that of the somato-motor network [t (991.566) = 10.158, p_FDR_ = 7.879 x 10^-23^, Cohen’s d = 0.642 M = 0.029, SD = 0.076]. Together, this suggests that out-of-sample developmental predictive accuracy for structure-function coupling, using the second cognitive factor, is strongest in the higher-order default-mode network, and lowest in the lower-order somatosensory network”.

We have added a separate section for GAMM cross-validation in the Methods (Page 27, Line 1361):

GAMM cross-validation

“We implemented a 5-fold cross validation procedure, stratified by dataset (2 levels: CALM or NKI). All observations from any given participant were assigned to either the testing or training fold, to prevent data leakage, and the cross-validation procedure was repeated 100 times, to account for randomness in data splits. The outcome was predicted global or network-level structure-function coupling across all test splits, operationalised as the Spearman rank correlation coefficient. To assess whether prediction accuracy exceeded chance, we compared empirical prediction accuracy with that of GAMMs trained and tested on null data in which cognitive factor 2 scores were permuted across subjects. The number of observations formed 3 exchangeability blocks (N = 320 with one observation, N = 105 with two observations, and N = 33 with three observations), whereby scores from a participant with two observations were replaced by scores from another participant with two observations, with participant-level scores kept together, and so on for all numbers of observations. We compared empirical and null prediction accuracies using independent sample t-tests as, although the same participants were examined, the shuffling meant that the relative ordering of participants within both distributions was not preserved. For parallelisation and better stability when estimating models fit on permuted data, we used the bam function from the mgcv R package (Wood, 2017)”.

We also added a justification for why we predicted coupling using behaviour or psychopathology, rather than vice versa (Page 27, Line 1349):

“When using our GAMMs to test for the relationship between cognition and psychopathology and our coupling metrics, we opted to predict structure-function coupling using cognitive or psychopathological dimensions, rather than vice versa, to minimise multiple comparisons. In the current framework, we corrected for 8 multiple comparisons within each domain. This would have increased to 16 multiple comparison corrections for predicting two cognitive dimensions using network-level coupling, and 24 multiple comparison corrections for predicting three psychopathology dimensions. Incorporating multiple networks as predictors within the same regression framework introduces collinearity, whilst the behavioural dimensions were orthogonal: for example, coupling is strongly correlated between the somato-motor and ventral attention networks (r_s_ = 0.721), between the default-mode and frontoparietal networks (r_s_ = 0.670), and between the dorsal attention and fronto-parietal networks (r_s_ = 0.650)”.

Finally, we noticed a rounding error in the ages of the data frame containing the structure-function coupling values and the cognitive/psychopathology dimensions. We rectified this and replaced the GAMM results, which largely remained the same.

In typical applications of diffusion map embedding, sparsification (e.g., retaining only the top 10 of the strongest connections) is often employed at the vertex-level resolution to ensure computational feasibility. However, since the present study performs the embedding at the level of 200 brain regions (a considerably coarser resolution), this step may not be necessary or justifiable. Specifically, for FC, it might be more appropriate to retain all positive connections rather than applying sparsification, which could inadvertently eliminate valuable information about lower-strength connections. Whereas for SC, as the values are strictly non-negative, retaining all connections should be feasible and would provide a more complete representation of the structural connectivity patterns. Given this, it would be helpful if the authors could clarify why they chose to include sparsification despite the coarser regional resolution, and whether they considered this alternative approach (using all available positive connections for FC and all non-zero values for SC). It would be interesting if the authors could provide their thoughts on whether the decision to run evaluations at the resolution of brain regions could itself impact the functional and structural manifolds, their alteration with age, and or their stability (in contrast to Dong et al. which tested alterations in highresolution gradients).

This is another great point. We could retain all connections, but we usually implement some form of sparsification to reduce noise, particularly in the case of functional connectivity. But we nonetheless agree with the reviewer’s point. We should check what impact this is having on the analysis. In brief, we found minimal effects of thresholding, suggesting that the strongest connections are driving the gradient (Page 7, Line 304):

“To assess the effect of sparsity on the derived gradients, we examined group-level structural (N = 222) and functional (N = 213) connectomes from the baseline session of NKI. The first three functional connectivity gradients derived using the full connectivity matrix (density = 92%) were highly consistent with those obtained from retaining the strongest 10% of connections in each row (r_1_ = 0.999, r_2_ = 0.998, r_3_ < 0.999, all p < 0.001). Likewise, the first three communicability gradients derived from retaining all streamline counts (density = 83%) were almost identical to those obtained from 10% row-wise thresholding (r_1_ = 0.994, r_2_ = 0.963, r_3_ = 0.955, all p < 0.001). This suggests that the reported gradients are driven by the strongest or most consistent connections within the connectomes, with minimal additional information provided by weaker connections. In terms of functional connectivity, such consistency reinforces past work demonstrating that the sensorimotor-toassociation axis, the major axis within the principal functional connectivity gradient, emerges across both the top- and bottom-ranked functional connections (Nenning et al., 2023)”.

Furthermore, we appreciate the nudge to share our thoughts on whether the difference between vertex versus nodal metrics could be important here, particularly regarding thresholds. To combine this point with R2’s recommendation to expand the Discussion, we have added the following paragraph (Page 19, Line 861):

“We consider the role of thresholding, cortical resolution, and head motion as avenues to reconcile the present results with select reports in the literature (Dong et al., 2021; Xia et al., 2022). We would suggest that thresholding has a greater effect on vertex-level data, rather than parcel-level. For example, a recent study revealed that the emergence of principal vertex-level functional connectivity gradients in childhood and adolescence are indeed threshold-dependent (Dong et al., 2024). Specifically, the characteristic unimodal organisation for children and transmodal organisation for adolescents only emerged at the 90% threshold: a 95% threshold produced a unimodal organisation in both groups, whilst an 85% threshold produced a transmodal organisation in both groups. Put simply, the ‘swapping’ of gradient orders only occurs at certain thresholds. Furthermore, our results are not necessarily contradictory to this prior report (Dong et al., 2021): developmental changes in high-resolution gradients may be supported by a stable low-dimensional coarse manifold. Indeed, our decision to use parcellated connectomes was partly driven by recent work which demonstrated that vertex-level functional gradients may be derived using biologically-plausible but random data with sufficient spatial smoothing, whilst this effect is minimal at coarser resolutions (Watson & Andrews, 2023). We observed a gradual increase in the variance of individual connectomes accounted for by the principal functional connectivity gradient in the referred subset of CALM, in line with prior vertex-level work demonstrating a gradual emergence of the sensorimotor-association axis as the principal axis of connectivity (Xia et al., 2022), as opposed to a sudden shift. It is also possible that vertex-level data is more prone to motion artefacts in the context of developmental work. Transitioning from vertex-level to parcel-level data involves smoothing over short-range connectivity, thus greater variability in short-range connectivity can be observed in vertex-level data. However, motion artefacts are known to increase short-range connectivity and decrease long-range connectivity, mimicking developmental changes (Satterthwaite et al., 2013). Thus, whilst vertexlevel data offers greater spatial resolution in representation of short-range connectivity relative to parcel-level data, it is possible that this may come at the cost of making our estimates of the gradients more prone to motion”.

Evaluating the consistency of gradients across development: the results shown in Figure 1e are used as evidence suggesting that gradients are consistent across ages. However, I believe additional analyses are required to identify potential sources of the observed inconsistency compared to previous works. The claim that the principal gradient explains a similar degree of variance across ages does not necessarily imply that the spatial structure remains the same. The observed variance explanation is hence not enough to ascertain inconsistency with findings from Dong et al., as the spatial configuration of gradients may still change over time. I suggest the following additional analyses to strengthen this claim. Alignment to group-level gradients: Assess how much of the variance in individual FC matrices is explained by each of the group-level gradients (G1, G2, and G3, for both FC and SC). This analysis could be visualized similarly to Figure 1e, with age on the x-axis and variance explained on the y-axis. If the explained variance varies as a function of age, it may indicate that the gradients are not as consistent as currently suggested.

This is another great suggestion. In the additional analyses above (new group-level analyses and unrotated gradient analyses) we rule-out a couple of the potential causes of the different developmental trends we observe in our data – namely the stability of the gradients over time. The suggested additional analysis is a great idea, and we have implemented it as follows (Page 8, Line 363):

“To evaluate the consistency of gradients across development, across baseline participants with functional connectomes from the referred CALM cohort (N = 177), we calculated the proportion of variance in individuallevel connectomes accounted for by group-level functional gradients. Specifically, we calculated the proportion of variance in an adjacency matrix A accounted for by the vector v_i_ as the fraction of the square of the scalar projection of v_i_ onto A, over the Frobenius norm of A. Using a generalised linear model, we then tested whether the proportion of variance explained varies systematically with age, controlling for sex and headmotion. The variance in individual-level functional connectomes accounted for by the group-level principal functional gradient gradually increased with development (β = 0.111, 95% CI = [0.022, 0.199], p = 1.452 x 10^-2^, Cohen’s d = 0.367), as shown in Figure 1g, and decreased with higher head motion (β = -10.041, 95% CI = [12.379, -7.702], p = 3.900 x 10^-17^), with no effect of sex (β = 0.071, 95% CI = [-0.380, 0.523], p = 0.757). We observed no developmental effects on the variance explained by the second (r_s_ = 0.112, p = 0.139) or third (r_s_ = 0.053, p = 0.482) group-level functional gradient. When repeated with the baseline functional connectivity for NKI (N = 213), we observed no developmental effects (β = 0.097, 95% CI = [-0.035, 0.228], p = 0.150) on the variance explained by the principal functional gradient after accounting for motion (β = -3.376, 95% CI = [8.281, 1.528], p = 0.177) and sex (β = -0.368, 95% CI = [-1.078, 0.342], p = 0.309). However, we observed significant developmental correlations between age and variance (r_s_ = 0.137, p = 0.046) explained before accounting for head motion and sex. We observed no developmental effects on the variance explained by the second functional gradient (r_s_ = -0.066, p = 0.338), but a weak negative developmental effect on the variance explained by the third functional gradient (r_s_ = -0.189, p = 0.006). Note, however, the magnitude of the variance accounted for by the third functional gradient was very small (all < 1%). When applied to communicability matrices in CALM, the proportion of variance accounted for by the group-level communicability gradient was negligible (all < 1%), precluding analysis of developmental change”.

“To further probe the consistency of gradients across development, we examined developmental changes in the standard deviation of gradient values, corresponding to heterogeneity, following prior work examining morphological (He et al., 2025) and functional connectivity gradients (Xia et al., 2022). Using a series of generalised linear models within the baseline referred subset of CALM, correcting for head motion and sex, we found that gradient variation for the principal functional gradient increased across development (=0.219, 95% CI = [0.091, 0.347], p = 0.001, Cohen’s d = 0.504), indicating greater heterogeneity (Figure 1h), whilst gradient variation for the principal communicability gradient decreased across development (β = -0.154, 95% CI = [-0.267, -0.040], p = 0.008, Cohen’s d = -0.301), indicating greater homogeneity (Figure 1h). Note, a paired t-test on the 173 common participants demonstrated a significant effect of modality on gradient variability (t(172) = -56.639, p = 3.663 x 10^-113^), such that the mean variability of communicability gradients (M = 0.033, SD = 0.001) was less than half that of functional connectivity (M = 0.076, SD = 0.010). Together, this suggests that principal functional connectivity and communicability gradients are established early in childhood and display age-related refinement, but not replacement”.

The Issue of Abstraction and Benefits of the Gradient-Based View: The manuscript interprets the eccentricity findings as reflecting changes along the segregation-integration spectrum. Given this, it is unclear why a more straightforward analysis using established graph-theory metrics of segregationintegration was not pursued instead. Mapping gradients and computing eccentricity adds layers of abstraction and complexity. If similar interpretations can be derived directly from simpler graph metrics, what additional insights does the gradient-based framework offer? While the manuscript argues that this approach provides “a more unifying account of cortical reorganization”, it is not evident why this abstraction is necessary or advantageous over traditional graph metrics. Clarifying these benefits would strengthen the rationale for using this method.

This is a great point, and something we spent quite a bit of time considering when designing the analysis. The central goal of our project was to identify gradients of brain organisation across different datasets and modalities and then test how the organisational principles of those modalities align. In other words, how do structural and functional ‘spaces’ intersect, and does this vary across the cortex? That for us was the primary motivation for operationalising organisation as nodal location within a low-dimensional manifold space (Bethlehem et al., 2020; Gale et al., 2022; Park et al., 2021), using a simple composite measure to achieve compression, rather than as a series of graph metrics. The reason we subsequently calculated those graph metrics and tested for their association was simply to help us interpret what eccentricity within that lowdimensional space means. Manifold eccentricity was moderately positively correlated to graph-theory metrics of integration, leaving a substantial portion of variance unaccounted for, but that association we think is nonetheless helpful for readers trying to interpret eccentricity. However, since ME tells us about the relative position of a node in that low-dimensional space, it is also likely capturing elements of multiple graph theory measures. Following the Reviewer’s question, this is something we decided to test. Specifically, using 4 measures of segregation, including two new metrics requested by the Reviewer in a minor point (weighted clustering coefficient and normalized degree centrality), we conducted a dominance analysis (Budescu, 1993) with normalized manifold eccentricity of the group-level referred CALM structural connectome. We also detail the use of gradient measures in developmental contexts, and how they can be complementary to traditional graph theory metrics.

We have added the following to the Results section (Page 10, Lines 472 onwards):

“To further contextualise manifold eccentricity in terms of integration and segregation beyond simple correlations, we conducted a multivariate dominance analysis (Budescu, 1993) of four graph theory metrics of segregation as predictors of nodal normalised manifold eccentricity within the group-level referred CALM structural and functional connectomes (Figure 2c). A dominance analysis assesses the relative importance of each predictor in a multilinear regression framework by fitting 2^n^ – 1 models (where n is the number of predictors) and calculating the relative increase in adjusted R2 caused by adding each predictor to the model across both main effects and interactions. A multilinear regression model including weighted clustering coefficient, within-module degree Z-score, participation coefficient and normalized degree centrality accounted for 59% of variance in nodal manifold eccentricity in the group-level CALM structural connectome. Withinmodule degree Z score was the most important predictor (40.31% dominance), almost twice that of the participation coefficient (24.03% dominance) and normalized degree centrality (24.05% dominance) which made roughly equal contributions. The least important predictor was the weighted clustering coefficient (11.62% dominance). When the same approach was applied for the group-level referred CALM functional connectome, the 4 predictors accounted for 52% variability. However, in contrast to the structural connectome, functional manifold eccentricity seemed to incorporate the same graph theory metrics in different proportions. Normalized degree centrality was the most important predictor (47.41% dominance), followed by withinmodule degree Z-score (24.27%), and then the participation coefficient (15.57%) and weighted clustering coefficient (12.76%) which made approximately equal contributions. Thus, whilst structural manifold eccentricity was dominated most by within-module degree Z-score and least by the weighted clustering coefficient, functional manifold eccentricity was dominated most by normalized degree centrality and least by the weighted clustering coefficient. This suggests that manifold mapping techniques incorporate different aspects of integration dependent on modality. Together, manifold eccentricity acts as a composite measure of segregation, being differentially sensitive to different aspects of segregation, without necessitating a priori specification of graph theory metrics. Further discussion of the value of gradient-based metrics in developmental contexts and as a supplement to traditional graph theory analyses is provided in the ‘Manifold Eccentricity’ methodology sub-section”.

We added further justification to the manifold eccentricity Methods subsection (Page 26, line 1283):

“Gradient-based measures hold value in developmental contexts, above and beyond traditional graph theory metrics: within a sample of over 600 cognitively-healthy adults aged between 18 and 88 years old, sensitivity of gradient-based within-network functional dispersion to age were stronger and more consistent across networks compared to segregation (Bethlehem et al., 2020). In the context of microstructural profile covariance, modules resolved by Louvain community detection occupied distinct positions across the principal two gradients, suggesting that gradients offer a way to meaningfully order discrete graph theory analyses (Paquola et al., 2019)”.

We added the following to the Introduction section outlining the application of gradients as cortex-wide coordinate systems (Page 3, Line 121):

“Using the gradient-based approach as a compression tool, thus forgoing the need to specify singular graph theory metrics a priori, we operationalised individual variability in low-dimensional manifolds as eccentricity (Gale et al., 2022; Park et al., 2021). Crucially, such gradients appear to be useful predictors of phenotypic variation, exceeding edge-level connectomics. For example, in the case of functional connectivity gradients, their predictive ability for externalizing symptoms and general cognition in neurotypical adults surpassed that of edge-level connectome-based predictive modelling (Hong et al., 2020), suggesting that capturing lowdimensional manifolds may be particularly powerful biomarkers of psychopathology and cognition”.

We also added the following to the Discussion section (Page 18, Line 839):

“By capitalising on manifold eccentricity as a composite measure of segregation across development, we build upon an emerging literature pioneering gradients as a method to establish underlying principles of structural (Paquola et al., 2020; Park et al., 2021) and functional (Dong et al., 2021; Margulies et al., 2016; Xia et al., 2022) brain development without a priori specification of specific graph theory metrics of interest”.

It is unclear whether the statistical tests finding significant dataset effects are capturing effects of neurotypical vs. Neurodivergent, or simply different scanners/sites. Could the neurotypical portion of CALM also be added to distinguish between these two sources of variability affecting dataset effects (i.e. ideally separating this to the effect of site vs. neurotypicality would better distinguish the effect of neurodivergence).

At a group-level, differences in the gradients between the two cohorts are very minor. Indeed, in the manuscript we describe these gradients as being seemingly ‘universal’. But we agree that we should test whether we can directly attribute any simple main effects of ‘dataset’ are resulting from the different site or the phenotype of the participants. The neurotypical portion of CALM (collected at the same site on the same scanner) helped us show that any minor differences in the gradient alignments is likely due to the site/scanner differences rather than the phenotype of the participants. We took the same approach for testing the simple main effects of dataset on manifold eccentricity. To better parse neurotypicality and site effects at an individual-level, we conducted a series of sensitivity analyses. First, in response to the reviewer’s earlier comment, we conducted a series of nodal generalized linear models for communicability and FC gradients derived from neurotypical and neurodivergent portions of CALM, alongside NKI, and tested for an effect of neurotypicality above and beyond scanner. As at the group level, having those additional scans on a ‘comparison’ sample for CALM is very helpful in teasing apart these effects. We find that neurotypicality affects communicability gradient expression to a greater degree than functional connectivity. We visualised these results and added them to Figure 1. Second, we used the same approach but for manifold eccentricity. Again, we demonstrate greater sensitivity of neurotypicality to communicability at a global-level, but we cannot pin these effects down to specific networks because the effects do not survive the necessary multiple comparison correction. We have added these analyses to the manuscript (Page 13, Line 583):

“Much as with the gradients themselves, we suspected that much of the simple main effect of dataset could reflect the scanner / site, rather than the difference in phenotype. Again, we drew upon the CALM comparison children to help us disentangle these two explanations. As a sensitivity analysis to parse effects of neurotypicality and dataset on manifold eccentricity, we conducted a series of generalized linear models predicting mean global and network-level manifold eccentricity, for each modality. We did this across all the baseline data (i.e. including the neurotypical comparison sample for CALM) using neurotypicality (2 levels: neurodivergent or neurotypical), site (2 levels: CALM or NKI), sex, head motion, and age at scan (Figure 3X). We restricted our analysis to baseline scans to create more equally-balanced groups. In terms of structural manifold eccentricity (N = 313 neurotypical, N = 311 neurodivergent), we observed higher manifold eccentricity in the neurodivergent participants at a global level (β = 0.090, p = 0.019, Cohen’s d = 0.188) but the individual network level effects did not survive the multiple comparison correction necessary for looking across all seven networks, with the default-mode network being the strongest (β = 0.135, p = 0.027, p_FDR_ = 0.109, Cohen’s d = 0.177). There was no significant effect of neurodiversity on functional manifold eccentricity (N = 292 neurotypical and N = 177 neurodivergent). This suggests that neurodiversity is significantly associated with structural manifold eccentricity, over and above differences in site, but we cannot distinguish these effects reliably in the functional manifold data”.

Third, we removed the Scheirer-Ray-Hare test from the results for two reasons. First, its initial implementation did not account for repeated measures, and therefore non-independence between observations, as the same participants may have contributed both structural and functional data. Second, if we wanted to repeat this analysis in CALM using the referred and control portions, a significant difference in group size existed, which may affect the measures of variability. Specifically, for baseline CALM, 311 referred and 91 control participants contributed SC data, whilst 177 referred and 79 control participants contributed FC data. We believe that the ‘cleanest’ parsing of dataset and site for effects of eccentricity is achieved using the GLMs in Figure 3.

We observed no significant effect of neurodivergence on the magnitude of structure-function coupling across development, and have added the following text (Page 14, Line 632):

“To parse effects of neurotypicality and dataset on structure-function coupling, we conducted a series of generalized linear models predicting mean global and network-level coupling using neurotypicality, site, sex, head motion, and age at scan, at baseline (N = 77 CALM neurotypical, N = 173 CALM neurodivergent, and N = 170 NKI). However, we found no significant effects of neurotypicality on structure-function coupling across development”.

Since we demonstrated no significant effects of neurotypicality on structure-function coupling magnitude across development, but found differential dataset-specific effects of age on coupling development, we added the following sentence at the end of the coupling trajectory results sub-section (Page 14, line 664):

“Together, these effects demonstrate that whilst the magnitude of structure-function coupling appears not to be sensitive to neurodevelopmental phenotype, its development with age is, particularly in higher-order association networks, with developmental change being reduced in the neurodivergent sample”.

Figure 1.c: A non-parametric permutation test (e.g. Mann-Whitney U test) could quantitatively identify regions with significant group differences in nodal gradient values, providing additional support for the qualitative findings.

This is a great idea. To examine the effect of referral status on nodal gradient values, whilst controlling for covariates (head motion and sex), we conducted a series of generalised linear models. We opted for this instead of a Mann-Whitney U test, as the former tests for differences in distributions, whilst the direction of the t-statistic for referral status from the GLM would allow us to specify the magnitude and direction of differences in nodal gradient values between the two groups. Again, we conducted this in CALM (referred vs control), at an individual-level, as downstream analyses suggested a main effect of dataset (which is reflected in the highly-similar group-level referred and control CALM gradients). We have updated the Results section with the following text (Page 6, Line 283):

“To examine the effect of referral status on participant-level nodal gradient values in CALM, we conducted a series of generalized linear models controlling for head motion, sex and age at scan (Figure 1d). We restricted our analyses to baseline scans to reduce the difference in sample size for the referred (311 communicability and 177 functional gradients, respectively) and control participants (91 communicability and 79 functional gradients, respectively), and to the principal gradients. For communicability, 42 regions showed a significant effect (p < 0.05) of neurodivergence before FDR correction, with 9 post FDR correction. 8 of these 9 regions had negative t-statistics, suggesting a reduced nodal gradient value and representation in the neurodivergent children, encompassing both lower-order somatosensory cortices alongside higher-order fronto-parietal and default-mode networks. The largest reductions were observed within the prefrontal cortices of the defaultmode network (t = -3.992, p = 6.600 x 10^-5^, p_FDR_ = 0.013, Cohen’s d = -0.476), the left orbitofrontal cortex of the limbic network (t = -3.710, p = 2.070 x 10^-4^, p_FDR_ = 0.020, Cohen’s d = -0.442) and right somato-motor cortex (t = -3.612, p = 3.040 x 10^-4^, p_FDR_ = 0.020, Cohen’s d = -0.431). The right visual cortex was the only exception, with stronger gradient representation within the neurotypical cohort (t = 3.071, p = 0.002, p_FDR_ = 0.048, Cohen’s d = 0.366). For functional connectivity, comparatively fewer regions exhibited a significant effect (p < 0.05) of neurotypicality, with 34 regions prior to FDR correction and 1 post. Significantly stronger gradient representation was observed in neurotypical children within the right precentral ventral division of the defaultmode network (t = 3.930, p = 8.500 x 10^-5^, p_FDR_ = 0.017, Cohen’s d = 0.532). Together, this suggests that the strongest and most robust effects of neurodivergence are observed within gradients of communicability, rather than functional connectivity, where alterations in both affect higher-order associative regions”.

In the harmonization methodology, it is mentioned that “if harmonisation was successful, we’d expect any significant effects of scanner type before harmonisation to be non-significant after harmonisation”. However, given that there were no significant effects before harmonization, the results reported do not help in evaluating the quality of harmonization.

We agree with the Reviewer, and have removed the post-harmonisation GLMs, and instead stating that there were no significant effects of scanner type before harmonization.

Figure 3: It would be helpful to include a plot showing the GAMM predictions versus real observations of eccentricity (x-axis: predictions, y-axis: actual values).

To plot the GAMM-predicted smooth effects of age, which we used for visualisation purposes only, we used the get_predictions function from the itsadug R package. This creates model predictions using the median value of nuisance covariates. Thus, whilst we specified the entire age range, the function automatically chooses the median of head motion, alongside controlling for sex (default level: male) and, for each dataset-specific trajectory. Since the gamm4 package separates the fitted model into a gam and linear mixed effects model (which accounts for participant ID as a random effect), and the get_predictions function only uses gam, random effects are not modelled in the predicted smooths. Therefore, any discrepancy between the observed and predicted manifold eccentricity values is likely due to sensitivity to default choices of covariates other than age, or random effects. To prevent Figure 3 being too over-crowded, we opted to not include the predictions: these were strongly correlated with real structural manifold data, but less for functional manifold data especially where significant developmental change was absent.

The 30mm threshold for filtering short streamlines in tractography is uncommon. What is the rationale for using such a large threshold, given the potential exclusion of many short-range association fibres?

A minimum length of 30mm was the default for the MRtrix3 reconstruction workflow, and something we have previously used. In a previous project, we systematically varied the minimum fibre length and found that this had minimal impact on network organisation (e.g. Mousley et al. 2025). However, we accept that short-range association fibres may have been excluded and have included this in the Discussion as a methodological limitation, alongside our predictions for how the gradients and structure-function coupling may’ve been altered had we included such fibres (Page 20, Line 955):

“A potential methodological limitation in the construction of structural connectomes was the 30mm tract length threshold which, despite being the QSIprep reconstruction default (Cieslak et al., 2021), may have potentially excluded short-range association fibres. This is pertinent as tracts of different lengths exhibit unique distributions across the cortex and functional roles (Bajada et al., 2019) : short-range connections occur throughout the cortex but peak within primary areas, including the primary visual, somato-motor, auditory, and para-hippocampal cortices, and are thought to dominate lower-order sensorimotor functional resting-state networks, whilst long-range connections are most abundant in tertiary association areas and are recruited alongside tracts of varying lengths within higher-order functional resting-state networks. Therefore, inclusion of short-range association fibres may have resulted in a relative increase in representation of lower-order primary areas and functional networks. On the other hand, we also note the potential misinterpretation of short-range fibres: they may be unreliably distinguished from null models in which tractography is restricted by cortical gyri only (Bajada et al., 2019). Further, prior (neonatal) work has demonstrated that the order of connectivity of regions and topological fingerprints are consistent across varying streamline thresholds (Mousley et al., 2025), suggesting minimal impact”.

Given the spatial smoothing of fMRI data (6mm FWHM), it would be beneficial to apply connectome spatial smoothing to structural connectivity measures for consistent spatial smoothness.

This is an interesting suggestion but given we are looking at structural communicability within a parcellated network, we are not sure that it would make any difference. The data structural data are already very smooth. Nonetheless we have added the following text to the Discussion (Page 20, Line 968):

“Given the spatial smoothing applied to the functional connectivity data, and examining its correspondence to streamline-count connectomes through structure-function coupling, applying the equivalent smoothing to structural connectomes may improve the reliability of inference, and subsequent sensitivity to cognition and psychopathology. Connectome spatial smoothing involves applying a smoothing kernel to the two streamline endpoints, whereby variations in smoothing kernels are selected to optimise the trade-off between subjectlevel reliability and identifiability, thus increasing the signal-to-noise ratio and the reliability of statistical inferences of brain-behaviour relationships (Mansour et al., 2022). However, we note that such smoothing is more effective for high-resolution connectomes, rather than parcel-level, and so have only made a modest improvement (Mansour et al., 2022)”.

Why was harmonization performed only within the CALM dataset and not across both CALM and NKI datasets? What was the rationale for this decision?

We thought about this very carefully. Harmonization aims to remove scanner or site effects, whilst retaining the crucial characteristics of interest. Our capacity to retain those characteristics is entirely dependent on them being *fully* captured by covariates, which are then incorporated into the harmonization process. Even with the best set of measures, the idea that we can fully capture ‘neurodivergence’ and thus preserve it in the harmonisation process is dubious. Indeed, across CALM and NKI there are limited number of common measures (i.e. not the best set of common measures), and thus we are limited in our ability to fully capture the neurodivergence with covariates. So, we worried that if we put these two very different datasets into the harmonisation process we would essentially eliminate the interesting differences between the datasets. We have added this text to the harmonization section of the Methods (Page 24, Line 1225):

“Harmonization aims to retain key characteristics of interest whilst removing scanner or site effects. However, the site effects in the current study are confounded with neurodivergence, and it is unlikely that neurodivergence may be captured fully using common covariates across CALM and NKI. Therefore, to preserve variation in neurodivergence, whilst reducing scanner effects, we harmonized within the CALM dataset only”.

The exclusion of subcortical areas from connectivity analyses is not justified.

This is a good point. We used the Schaefer atlas because we had previously used this to derive both functional and structural connectomes, but we agree that it would have been good to include subcortical areas (Page 20, Line 977).

“A potential limitation of our study was the exclusion of subcortical regions. However, prior work has shed light on the role of subcortical connectivity in structural and functional gradients, respectively, of neurotypical populations of children and adolescents (Park et al., 2021; Xia et al., 2022). For example, in the context of the primary-to-transmodal and sensorimotor-to-visual functional connectivity gradients, the mean gradient scores within subcortical networks were demonstrated to be relatively stable across childhood and adolescence (Xia et al., 2022). In the context of structural connectivity gradients derived from streamline counts, which we demonstrated were highly consistent with those derived from communicability, subcortical structural manifolds weighted by their cortical connectivity were anchored by the caudate and thalamus at one pole, and by the hippocampus and nucleus accumbens at the opposite pole, with significant age-related manifold expansion within the caudate and thalamus (Park et al., 2021)”.

In the KNN imputation method, were uniform weights used, or was an inverse distance weighting applied?

Uniform weights were used, and we have updated the manuscript appropriately.

The manuscript should clarify from the outset that the reported sample size (N) includes multiple longitudinal observations from the same individuals and does not reflect the number of unique participants.

We have rectified the Abstract (Page 2, Line 64) and Introduction (Page 3, Line 138):

“We charted the organisational variability of structural (610 participants, N = 390 with one observation, N = 163 with two observations, and N = 57 with three) and functional (512 participants, N = 340 with one observation, N = 128 with two observations, and N = 44 with three)”.

The term “structural gradients” is ambiguous in the introduction. Clarify that these gradients were computed from structural and functional connectivity matrices, not from other structural features (e.g. cortical thickness).

We have clarified this in the Introduction (Page 3, Line 134):

“Applying diffusion-map embedding as an unsupervised machine-learning technique onto matrices of communicability (from streamline SIFT2-weighted fibre bundle capacity) and functional connectivity, we derived gradients of structural and functional brain organisation in children and adolescents…”

Page 5: The sentence, “we calculated the normalized angle of each structural and functional connectome to derive symmetric affinity matrices” is unclear and needs clarification.

We have clarified this within the second paragraph of the Results section (Page 4, Line 185):

“To capture inter-nodal similarity in connectivity, using a normalised angle kernel, we derived individual symmetric affinity matrices from the left and right hemispheres of each communicability and functional connectivity matrix. Varying kernels capture different but highly-related aspects of inter-nodal similarity, such as correlation coefficients, Gaussian kernels, and cosine similarity. Diffusion-map embedding is then applied on the affinity matrices to derive gradients of cortical organisation”.

Figure 1.a: “Affine A” likely refers to the affinity matrix. The term “affine” may be confusing; consider using a clearer label. It would also help to add descriptive labels for rows and columns (e.g. region x region).

Thank you for this suggestion! We have replaced each of the labels with “pairwise similarity”. We also labelled the rows and columns as regions.

Figure 1.d: Are the cross-group differences statistically significant? If so, please indicate this in the figure.

We have added the results of a series of linear mixed effects models to the legend of Figure 1 (Page 6, line 252):

“indicates a significant effect of dataset (p < 0.05) on variance explained within a linear mixed effects model controlling for head motion, sex, and age at scan”.

The sentence “whose connectomes were successfully thresholded” in the methods is unclear. What does “successfully thresholded” mean? Additionally, this seems to be the first mention of the Schaefer 100 and Brainnetome atlas; clarify where these parcellations are used.

We have amended the Methodology section (Page 23, Line 1138):

“For each participant, we retained the strongest 10% of connections per row, thus creating fully connected networks required for building affinity matrices. We excluded any connectomes in which such thresholding was not possible due to insufficient non-zero row values. To further ensure accuracy in connectome reconstruction, we excluded any participants whose connectomes failed thresholding in two alternative parcellations: the 100node Schaefer 7-network (Schaefer et al., 2018) and Brainnetome 246-node (Fan et al., 2016) parcellations, respectively”.

We have also specified the use of the Schaefer 200-node parcellation in the first sentence on the second Results paragraph.

The use of “streamline counts” is misleading, as the method uses SIFT2-weighted fibre bundle capacity rather than raw streamline counts. It would be better to refer to this measure as “SIFT2-weighted fibre bundle capacity” or “FBC”.

We replaced all instances of “streamline counts” with “SIFT2-weighted fibre bundle capacity” as appropriate.

Figure 2.c: Consider adding plots showing changes in eccentricity against (1) degree centrality, and (2) weighted local clustering coefficient. Additionally, a plot showing the relationship between age and mean eccentricity (averaged across nodes) at the individual level would be informative.

We added the correlation between eccentricity and both degree centrality and the weighted local clustering coefficient and included them in our dominance analysis in Figure 2. In terms of the relationship between age and mean (global) eccentricity, these are plotted in Figure 3.

Figure 2.b: Considering the results of the following sections, it would be interesting to include additional KDE/violin plots to show group differences in the distribution of eccentricity within 7 different functional networks.

As part of our analysis to parse neurotypicality and dataset effects, we tested for group differences in the distribution of structural and functional manifold eccentricity within each of the 7 functional networks in the referred and control portions of CALM and have included instances of significant differences with a coloured arrow to represent the direction of the difference within Figure 3.

Figure 3: Several panels lack axis labels for x and y axes. Adding these would improve clarity.

To minimise the amount of text in Figure 3, we opted to include labels only for the global-level structural and functional results. However, to aid interpretation, we added a small schematic at the bottom of Figure 3 to represent all axis labels.

The statement that “differences between datasets only emerged when taking development into account” seems inaccurate. Differences in eccentricity are evident across datasets even before accounting for development (see Fig 2.b and the significance in the Scheirer-Ray-Hare test).

We agree – differences in eccentricity across development and datasets are evident in structural and functional manifold eccentricity, as well as within structure-function coupling. However, effects of neurotypicality were particularly strong for the maturation of structure-function coupling, rather than magnitude. Therefore, we have rephrased this sentence in the Discussion (page 18, line 832):

“Furthermore, group-level structural and functional gradients were highly consistent across datasets, whilst differences between datasets were emphasised when taking development into account, through differing rates of structural and functional manifold expansion, respectively, alongside maturation of structure-function coupling”.

The handling of longitudinal data by adding a random effect for individuals is not clear in the main text. Mentioning this earlier could be helpful.

We have included this detail in the second sentence of the “developmental trajectories of structural manifold contraction and functional manifold expansion” results sub-section (page 11, line 503):

“We included a random effect for each participant to account for longitudinal data”.

Figure 4.b: Why were ranks shown instead of actual coefficient of variation values? Consider including a cortical map visualization of the coefficients in the supplementary material.

We visualised the ranks, instead of the actual coefficient of variation (CV) values, due to considerable variability and skew in the magnitude of the CV, ranging from 28.54 (in the right visual network) to 12865.68 (in the parietal portion of the left default-mode network), with a mean of 306.15. If we had visualised the raw CV values, these larger values would’ve been over-represented. We’ve also noticed and rectified an error in the labelling of the colour bar for Figure 4b: the minimum should be most variable (i.e. a rank of 1). To aid contextualisation of the ranks, we have added the following to the Results (page 14, line 626):

“The distribution of cortical coefficients of variation (CV) varied considerably, with the largest CV (in the parietal division of the left default-mode network) being over 400 times that of the smallest (in the right visual network). The distribution of absolute CVs was positively skewed, with a Fisher skewness coefficient g_1_ of 7.172, meaning relatively few regions had particularly high inter-individual variability, and highly peaked, with a kurtosis of 54.883, where a normal distribution has a skewness coefficient of 0 and a kurtosis of 3”.

**Reviewer #2 (Public review):**

**Some differences in developmental trajectories between CALM and NKI (e.g. Figure 4d) are not explained. Are these differences expected, or do they suggest underlying factors that require further investigation?**

This is a great point, and we appreciate the push to give a fuller explanation. It is very hard to know whether these effects are expected or not. We certainly don’t know of any other papers that have taken this approach. In response to the reviewer’s point, we decided to run some more analyses to better understand the differences. Having observed stronger age effects on structure-function coupling within the neurotypical NKI dataset, compared to the absent effects in the neurodivergent portion of CALM, we wanted to follow up and test that it really is that coupling is more sensitive to the neurodivergent versus neurotypical difference between CALM and NKI (rather than say, scanner or site effects). In short, we find stronger developmental effects of coupling within the neurotypical portion of CALM, rather than neurodivergent, and have added this to the Results (page 15, line 701):

“To further examine whether a closer correspondence of structure-function coupling with age is associated with neurotypicality, we conducted a follow-up analysis using the additional age-matched neurotypical portion of CALM (N = 77). Given the widespread developmental effects on coupling within the neurotypical NKI sample, compared to the absent effects in the neurodivergent portion of CALM, we would expect strong relationships between age and structure-function coupling with the neurotypical portion of CALM. This is indeed what we found: structure-function coupling showed a linear negative relationship with age globally (F = 16.76, p_FDR_ < 0.001, adjusted R^2^ = 26.44%), alongside fronto-parietal (F = 9.24, p_FDR_ = 0.004, adjusted R^2^ = 19.24%), dorsalattention (F = 13.162, p_FDR_ = 0.001, adjusted R^2^ = 18.14%), ventral attention (F = 11.47, p_FDR_ = 0.002, adjusted R^2^ = 22.78), somato-motor (F = 17.37, p_FDR_ < 0.001, adjusted R^2^ = 21.92%) and visual (F = 11.79, p_FDR_ = 0.002, adjusted R^2^ = 20.81%) networks. Together, this supports our hypothesis that within neurotypical children and adolescents, structure-function coupling decreases with age, showing a stronger effect compared to their neurodivergent counterparts, in tandem with the emergence of higher-order cognition. Thus, whilst the magnitude of structure-function coupling across development appeared insensitive to neurotypicality, its maturation is sensitive. Tentatively, this suggests that neurotypicality is linked to stronger and more consistent maturational development of structure-function coupling, whereby the tethering of functional connectivity to structure across development is adaptive”.

In conjunction with the Reviewer’s later request to deepen the Discussion, we have included an additional paragraph attempting to explain the differences in neurodevelopmental trajectories of structure-function coupling (Page 19, Line 924):

“Whilst the spatial patterning of structure-function coupling across the cortex has been extensively documented, as explained above, less is known about developmental trajectories of structure-function coupling, or how such trajectories may be altered in those with neurodevelopmental conditions. To our knowledge, only one prior study has examined differences in developmental trajectories of (non-manifold) structure-function coupling in typically-developing children and those with attention-deficit hyperactivity disorder (Soman et al., 2023), one of the most common conditions in the neurodivergent portion of CALM. Namely, using cross-sectional and longitudinal data from children aged between 9 and 14 years old, they demonstrated increased coupling across development in higher-order regions overlapping with the defaultmode, salience, and dorsal attention networks, in children with ADHD, with no significant developmental change in controls, thus encompassing an ectopic developmental trajectory (Di Martino et al., 2014; Soman et al., 2023). Whilst the current work does not focus on any condition, rather the broad mixed population of young people with neurodevelopmental symptoms (including those with and without diagnoses), there are meaningful individual and developmental differences in structure-coupling. Crucially, it is not the case that simply having stronger coupling is desirable. The current work reveals that there are important developmental trajectories in structure-function coupling, suggesting that it undergoes considerable refinement with age. Note that whilst the magnitude of structure-function coupling across development did not differ significantly as a function of neurodivergence, its relationship to age did. Our working hypothesis is that structural connections allow for the ordered integration of functional areas, and the gradual functional modularisation of the developing brain. For instance, those with higher cognitive ability show a stronger refinement of structurefunction coupling across development. Future work in this space needs to better understand not just how structural or functional organisation change with time, but rather how one supports the other”.

The use of COMBAT may have excluded extreme participants from both datasets, which could explain the lack of correlations found with psychopathology.

COMBAT does not exclude participants from datasets but simply adjusts connectivity estimates. So, the use of COMBAT will not be impacting the links with psychopathology by removing participants. But this did get us thinking. Excluding participants based on high motion may have systematically removed those with high psychopathology scores, meaning incomplete coverage. In other words, we may be under-representing those at the more extreme end of the range, simply because their head-motion levels are higher and thus are more likely to be excluded. We found that despite certain high-motion participants being removed, we still had good coverage of those with high scores and were therefore sensitive within this range. We have added the following to the revised Methods section (Page 26, Line 1338):

“As we removed participants with high motion, this may have overlapped with those with higher psychopathology scores, and thus incomplete coverage. To examine coverage and sensitivity to broad-range psychopathology following quality control, we calculated the Fisher-Pearson skewness statistic g_1_ for each of the 6 Conners t-statistic measures and the proportion of youth with a t-statistic equal to or greater than 65, indicating an elevated or very elevated score. Measures of inattention (g_1_ = 0.11, 44.20% elevated), hyperactivity/impulsivity (g_1_ = 0.48, 36.41% elevated), learning problems (g_1_ = 0.45, 37.36% elevated), executive functioning (g_1_ = 0.27, 38.16% elevated), aggression (g_1_ = 1.65, 15.58% elevated), and peer relations (g_1_ = 0.49, 38% elevated) were positively skewed and comprised of at least 15% of children with elevated or very elevated scores, suggesting sufficient coverage of those with extreme scores”.

There is no discussion of whether the stable patterns of brain organization could result from preprocessing choices or summarizing data to the mean. This should be addressed to rule out methodological artifacts.

This is a brilliant point. We are necessarily using a very lengthy pipeline, with many design choices to explore structural and functional gradients and their intersection. In conjunction with the Reviewer’s later suggestion to deepen the Discussion, we have added the following paragraph which details the sensitivity analyses we carried out to confirm the observed stable patterns of brain organization (Page 18, Line 863):

“That is, whilst we observed developmental refinement of gradients, in terms of manifold eccentricity, standard deviation, and variance explained, we did not observe replacement. Note, as opposed to calculating gradients based on group data, such as a sliding window approach, which may artificially smooth developmental trends and summarise them to the mean, we used participant-level data throughout. Given the growing application of gradient-based analyses in modelling structural (He et al., 2025; Li et al., 2024) and functional (Dong et al., 2021; Xia et al., 2022) brain development, we hope to provide a blueprint of factors which may affect developmental conclusions drawn from gradient-based frameworks”.

Although imputing missing data was necessary, it would be useful to compare results without imputed data to assess the impact of imputation on findings.

It is very hard to know the impact of imputation without simply removing those participants with some imputed data. Using a simulation experiment, we expressed the imputation accuracy as the root mean squared error normalized by the range of observable data in each scale. This produced a percentage error margin. We demonstrate that imputation accuracy across all measures is at worst within approximately 11% of the observed data, and at best within approximately 4% of the observed data, and have included the following in the revised Methods section (Page 27, Line 1348):

“Missing data

To avoid a loss of statistical power, we imputed missing data. 27.50% of the sample had one or more missing psychopathology or cognitive measures (equal to 7% of all values), and the data was not missing at random: using a Welch’s t-test, we observed a significant effect of missingness on age [t (264.479) = 3.029, p = 0.003, Cohen’s d = 0.296], whereby children with missing data (M = 12.055 years, SD = 3.272) were younger than those with complete data (M = 12.902 years, SD = 2.685). Using a subset with complete data (N = 456), we randomly sampled 10% of the values in each column with replacement and assigned those as missing, thereby mimicking the proportion of missingness in the entire dataset. We conducted KNN imputation (uniform weights) on the subset with complete data and calculated the imputation accuracy as the root mean squared error normalized by the observed range of each measure. Thus, each measure was assigned a percentage which described the imputation margin of error. Across cognitive measures, imputation was within a 5.40% mean margin of error, with the lowest imputation error in the Trail motor speed task (4.43%) and highest in the Trails number-letter switching task (7.19%). Across psychopathology measures, imputation exhibited a mean 7.81% error margin, with the lowest imputation error in the Conners executive function scale (5.75%) and the highest in the Conners peer relations scale (11.04%). Together, this suggests that imputation was accurate”.

The results section is extensive, with many reports, while the discussion is relatively short and lacks indepth analysis of the findings. Moving some results into the discussion could help balance the sections and provide a deeper interpretation.

We agree with the Reviewer and appreciate the nudge to expand the Discussion section. We have added 4 sections to the Discussion. The first explores the importance of the default-mode network as a region whose coupling is most consistently predicted by working memory across development and phenotypes, in terms of its underlying anatomy (Paquola et al., 2025) (Page 20, Line 977):

“An emerging theme from our work is the importance of the default-mode network as a region in which structure-function coupling is reliably predicted by working memory across neurodevelopmental phenotypes and datasets during childhood and adolescence. Recent neurotypical adult investigations combining highresolution post-mortem histology, in vivo neuroimaging, and graph-theory analyses have revealed how the underlying neuroanatomy of the default-mode network may support diverse functions (Paquola et al., 2025), and thus exhibit lower structure-function coupling compared to unimodal regions. The default-mode network has distinct neuroanatomy compared to the remaining 6 intrinsic resting-state functional networks (Yeo et al., 2011), containing a distinctive combination of 5 of the 6 von Economo and Koskinas cell types (von Economo & Koskinas, 1925), with an over-representation of heteromodal cortex, and uniquely balancing output across all cortical types. A primary cytoarchitectural axis emerges, beyond which are mosaic-like spatial topographies. The duality of the default-mode network, in terms of its ability to both integrate and be insulated from sensory information, is facilitated by two microarchitecturally distinct subunits anchored at either end of the cytoarchitectural axis (Paquola et al., 2025). Whilst beyond the scope of the current work, structure-function coupling and their predictive value for cognition may also differ across divisions within the default-mode network, particularly given variability in the smoothness and compressibility of cytoarchitectural landscapes across subregions (Paquola et al., 2025)”.

The second provides a deeper interpretation and contextualisation of greater sensitivity of communicability, rather than functional connectivity, to neurodivergence (Page 19, Lines 907):

“We consider two possible factors to explain the greater sensitivity of neurodivergence to gradients of communicability, rather than functional connectivity. First, functional connectivity is likely more sensitive to head motion than structural-based communicability and suffers from reduced statistical power due to stricter head motion thresholds, alongside greater inter-individual variability. Second, whilst prior work contrasting functional connectivity gradients from neurotypical adults with those with confirmed ASD diagnoses demonstrated vertex-level reductions in the default-mode network in ASD and marginal increases in sensorymotor communities (Hong et al., 2019), indicating a sensitivity of functional connectivity to neurodivergence, important differences remain. Specifically, whilst the vertex-level group-level differences were modest, in line with our work, greater differences emerged when considering step-wise functional connectivity (SFC); in other words, when considering the dynamic transitions of or information flow through the functional hierarchy underlying the static functional connectomes, such that ASD was characterised by initial faster SFC within the unimodal cortices followed by a lack of convergence within the default-mode network (Hong et al., 2019). This emphasis on information flow and dynamic underlying states may point towards greater sensitivity of neurodivergence to structural communicability – a measure directly capturing information flow – than static functional connectivity”.

The third paragraph situates our work within a broader landscape of reliable brain-behaviour relationships, focusing on the strengths of combining clinical and normative samples to refine our interpretation of the relationship between gradients and cognition, as well as the importance of equifinality in developmental predictive work (Page 20, line 994):

“In an effort to establish more reliable brain-behaviour relationships despite not having the statistical power afforded by large-scale, typically normative, consortia (Rosenberg & Finn, 2022), we demonstrated the development-dependent link between default-mode structure-function coupling and working memory generalised across clinical (CALM) and normative (NKI) samples, across varying MRI acquisition parameters, and harnessing within- and across-participant variation. Such multivariate associations are likely more reliable than their univariate counterparts (Marek et al., 2022), but can be further optimised using task-related fMRI (Rosenberg & Finn, 2022). The consistency, or lack of, of developmental effects across datasets emphasises the importance of validating brain-behaviour relationships in highly diverse samples. Particularly evident in the case of structure-function coupling development, through our use of contrasting samples, is equifinality (Cicchetti & Rogosch, 1996), a key concept in developmental neuroscience: namely, similar ‘endpoints’ of structure-function coupling may be achieved through different initialisations dependent on working memory.

The fourth paragraph details methodological limitations in response to Reviewer 1’s suggestions to justify the exclusion of subcortical regions and consider the role of spatial smoothing in structural connectome construction as well as the threshold for filtering short streamlines”.

While the methods are thorough, it is not always clear whether the optimal approaches were chosen for each step, considering the available data.

In response to Reviewer 1’s concerns, we conducted several sensitivity analyses to evaluate the robustness of our results in terms of procedure. Specifically, we evaluated the impact of thresholding (full or sparse), level of analysis (individual or group gradients), construction of the structural connectome (communicability or fibre bundle capacity), Procrustes rotation (alignment to group-level gradients before Procrustes), tracking the variance explained in individual connectomes by group-level gradients, impact of head motion, and distinguishing between site and neurotypicality effects. All these analyses converged on the same conclusion: whilst we observe some developmental refinement in gradients, we do not observe replacement. We refer the reviewer to their third point, about whether stable patterns of brain organization were artefactual.

The introduction is overly long and includes numerous examples that can distract readers unfamiliar with the topic from the main research questions.

We have removed the following from the Introduction, reducing it to just under 900 words:

“At a molecular level, early developmental patterning of the cortex arises through interacting gradients of morphogens and transcription factors (see Cadwell et al., 2019). The resultant areal and progenitor specialisation produces a diverse pool of neurones, glia, and astrocytes (Hawrylycz et al., 2015). Across childhood, an initial burst in neuronal proliferation is met with later protracted synaptic pruning (Bethlehem et al., 2022), the dynamics of which are governed by an interplay between experience-dependent synaptic plasticity and genomic control (Gottlieb, 2007)”.

“The trends described above reflect group-level developmental trends, but how do we capture these broad anatomical and functional organisational principles at the level of an individual?”

We’ve also trimmed the second Introduction paragraph so that it includes fewer examples, such as removal of the wiring-cost optimisation that underlies structural brain development, as well as removing specific instances of network segregation and integration that occur throughout childhood.